# In utero adenine base editing corrects multi-organ pathology in a lethal lysosomal storage disease

Sourav K. Bose[1,2,9], Brandon M. White [1,2,9], Meghana V. Kashyap [1], Apeksha Dave[1,2], Felix R. De Bie [1,2], Haiying Li[1,2], Kshitiz Singh[1,2], Pallavi Menon[1,2], Tiankun Wang[1,2], Shiva Teerdhala[1,2], Vishal Swaminathan [1,2], Heather A. Hartman [1,2], Sowmya Jayachandran[3,4], Prashant Chandrasekaran[3,4], Kiran Musunuru[5,6,7], Rajan Jain[6,8], David B. Frank [3,4,5], Philip Zoltick[1,2] & William H. Peranteau[1,2✉]

In utero base editing has the potential to correct disease-causing mutations before the onset of pathology. Mucopolysaccharidosis type I (MPS-IH, Hurler syndrome) is a lysosomal storage disease (LSD) affecting multiple organs, often leading to early postnatal cardio-pulmonary demise. We assessed in utero adeno-associated virus serotype 9 (AAV9) delivery of an adenine base editor (ABE) targeting the *Idua* G→A (W392X) mutation in the MPS-IH mouse, corresponding to the common *IDUA* G→A (W402X) mutation in MPS-IH patients. Here we show efficient long-term W392X correction in hepatocytes and cardiomyocytes and low-level editing in the brain. In utero editing was associated with improved survival and amelioration of metabolic, musculoskeletal, and cardiac disease. This proof-of-concept study demonstrates the possibility of efficiently performing therapeutic base editing in multiple organs before birth via a clinically relevant delivery mechanism, highlighting the potential of this approach for MPS-IH and other genetic diseases.

[1] Center for Fetal Research, Children's Hospital of Philadelphia, Philadelphia, PA, USA. [2] Division of General, Thoracic and Fetal Surgery, Children's Hospital of Philadelphia, Philadelphia, PA, USA. [3] Division of Pediatric Cardiology, Children's Hospital of Philadelphia, Perelman School of Medicine at the University of Pennsylvania, Philadelphia, PA, USA. [4] Center for Pulmonary Biology, Perelman School of Medicine at the University of Pennsylvania, Philadelphia, PA, USA. [5] Cardiovascular Institute, Department of Medicine, Perelman School of Medicine at the University of Pennsylvania, Philadelphia, PA, USA. [6] Department of Medicine, Perelman School of Medicine at the University of Pennsylvania, Philadelphia, PA, USA. [7] Department of Genetics, Perelman School of Medicine at the University of Pennsylvania, Philadelphia, PA, USA. [8] Department of Cell and Developmental Biology, Institute for Regenerative Medicine, Perelman School of Medicine at the University of Pennsylvania, Philadelphia, PA, USA. [9] These authors contributed equally: Sourav K. Bose, Brandon M. White. ✉email: peranteauw@chop.edu

Lysosomal storage disorders affect multiple organs, have limited treatments, and have pathology that begins before birth[1]. In MPS-IH, *IDUA* gene mutations cause α-L-iduronidase (IDUA) deficiency and lysosomal accumulation of glycosaminoglycans (GAGs). The incidence of MPS-IH is 1:100,000 in Western society and one of the most common mutations (G→A; tryptophan→stop; W402X) accounts for over 40% of patients, results in undetectable IDUA in the homozygous state, and has a strong genotype–phenotype correlation[2]. Children present by 6 months of age with hepatosplenomegaly, abdominal wall hernias, musculoskeletal abnormalities, retinal and neurocognitive degeneration, and cardiac disease and die of cardiorespiratory complications by 5–10 years of age without treatment[3–5].

Although MPS-IH typically presents with symptoms by 6 months of age, it can be prenatally diagnosed via biochemical and genetic assays and associated pathology begins before birth[1,6–9]. On histopathologic examination, mid-gestation MPS-IH fetuses have demonstrated evidence of disease in multiple organs including the liver, heart, and brain[6–8,10]. Studies of severe MPS-IH cases demonstrate tissue deposition of GAGs leading to neurologic and bone pathology as early as 18 weeks gestation[11,12]. Finally, prenatal cardiac dysfunction has led to myocardial hypertrophy and early postnatal death in MPS-IH[9].

Current postnatal treatments include costly, lifelong, immunogenic enzyme replacement therapy (ERT), and hematopoietic stem cell transplantation (HSCT), which is limited by donor availability, graft failure, graft-versus-host disease, and complications of myeloablation/immunosuppression[3]. Both human and mouse studies have demonstrated improved outcomes following early initiation of ERT or HSCT compared to late treatment[12–16]. Importantly, in humans, neither treatment resolves preexisting musculoskeletal and cardiac pathologies[3,4,13], which significantly contribute to MPS-IH clinical grade[17]. Nonetheless, these findings suggest that there are benefits to early diagnosis and treatment in MPS-IH, potentially even before birth. Moreover, current therapies have a limited ability to correct the global disease phenotype, especially with delayed initiation.

Gene therapy and editing may address current treatment limitations in MPS-IH by augmenting IDUA expression in diseased organs or by enhancing liver IDUA secretion for systemic uptake. Postnatal systemic gene therapy and editing studies in the *Idua*[−/−] mouse model are encouraging. Studies involving the intravascular AAV and retroviral delivery of the *Idua* transgene have demonstrated mitigation of the skeletal, metabolic, neurologic, cardiac, ear, and eye disease phenotypes[18–20]; however, these approaches are respectively limited by potential loss of an episomal transgene and insertional mutagenesis. Similarly, AAV-mediated zinc-finger nuclease editing to express *Idua* in the hepatocyte *Albumin* locus of adult mice caused enhanced IDUA secretion, decreased tissue GAGs, and improved neurobehaviour[21]. Finally, neonatal hydrodynamic intravascular liposomal delivery of CRISPR-Cas9 targeting the hepatocyte *Rosa26* locus for homology-directed repair (HDR) with *Idua* integration partially improved GAGs, serum IDUA, and skeletal and cardiac disease[22]. Although encouraging, postnatal CRISPR-HDR is inefficient and requires double-stranded DNA breaks (DSBs) that are associated with unwanted mutagenesis, large deletions, and complex rearrangements at on- and off-target sites[23,24].

In contrast, base editing is a CRISPR editing approach that can convert adenine to guanine in a site-specific fashion without the need for DSBs or HDR templates. The ABE comprises a catalytically-impaired *Streptococcus pyogenes* Cas9 (SpCas9) and a modified tRNA adenine deaminase[25]. The SpCas9 guide RNA (gRNA) tethers the ABE to the target site, and the adenine deaminase converts a nearby adenine to hypoxanthine and, ultimately, guanine. Unlike HDR, adenine base editing does not require cells to be proliferating to efficiently introduce mutations and has infrequent unwanted on- and off-target mutagenesis[26,27], offering a potentially safer, more efficient correction of the MPS-IH G→A mutation.

Thus, the prenatal onset of pathology, feasibility of prenatal diagnosis, and early progressive postnatal morbidity responsive to early therapy suggest a potential benefit to MPS-IH prenatal therapy. The developing fetus has many properties that make it ideal for in vivo base editing. Small fetal size allows delivery of high-dose gene editing technology per weight; somatic and progenitor cells of multiple organs are accessible for efficient viral transduction[28–30]; and the nascent and permissive blood–brain barrier (BBB) facilitates systemic access to the central nervous system (CNS)[31]. In addition, the fetal immune system is tolerant. For example, multiple studies have demonstrated the lack of an immune response to the viral vector and transgene product, including SpCas9, following in utero gene therapy/editing[32–35]. Furthermore, Cas9-specific immunity has been demonstrated in humans after birth[36] and has eliminated edited hepatocytes following AAV-mediated Cas9 delivery in mouse models[37]. Finally, in utero base editing offers the potential to treat MPS-IH prior to the onset of pathology.

We previously demonstrated that adenovirus-mediated in utero base editing takes advantage of these properties and efficiently introduces a nonsense mutation in the *Hpd* gene in hepatocytes to rescue the lethal phenotype in the hereditary tyrosinemia type 1 mouse model[34]. However, adenoviral delivery is not clinically relevant, and there is a tremendous survival advantage of corrected hepatocytes in that model.

We now evaluate AAV9-mediated in utero intravascular delivery of an ABE to correct the G→A mutation and rescue the disease phenotype in the *Idua*-W392X MPS-IH mouse model which recapitulates W402X MPS-IH disease in humans[38].

## Results

### In utero split-intein AAV gene editing in the *R26*[mTmG/+] mouse model results in liver and heart editing.

Due to the limited AAV9 packaging capacity, we sought to deliver the ABE-gRNA transgene in split AAVs with reconstitution of the ABE protein in vivo via inteins[39]. This approach has previously demonstrated the ability to produce a functioning adenine base editor in vivo[39]. To initially evaluate this approach in utero, standard gene editing was performed in the tractable *R26*[mTmG/+] mouse model wherein deletion of a *loxP*-flanked membrane tomato (*mT*) cassette and subsequent nonhomologous end-joining (NHEJ) switches native red fluorescence to green. Split-intein AAV9s containing SpCas9 and the *loxP*-targeting gRNA were injected via the vitelline vein, which drains directly into the fetal liver, into embryonic day 15.5 (E15.5) *R26*[mTmG/+] fetuses. Wide-field microscopy and flow cytometry in the heart and liver on days of life 1 and 7 demonstrated GFP expression consistent with editing, whereas minimal GFP expression was noted in the brain (Supplementary Fig. 1a–s). Furthermore, immunohistochemistry (IHC) to investigate discrete cell populations revealed editing in cardiomyocytes and LGR5[+] liver progenitor cells (Supplementary Fig. 1t-aa).

### In utero split-intein AAV base editing in the *Idua*-W392X MPS-IH mouse corrects the G→A mutation.

Having demonstrated successful in utero split-intein AAV CRISPR-mediated gene editing in the heart and liver, two affected organs in MPS-IH, we next sought to use an ABE to prenatally correct the G→A mutation in the *Idua*-W392X mouse model. The

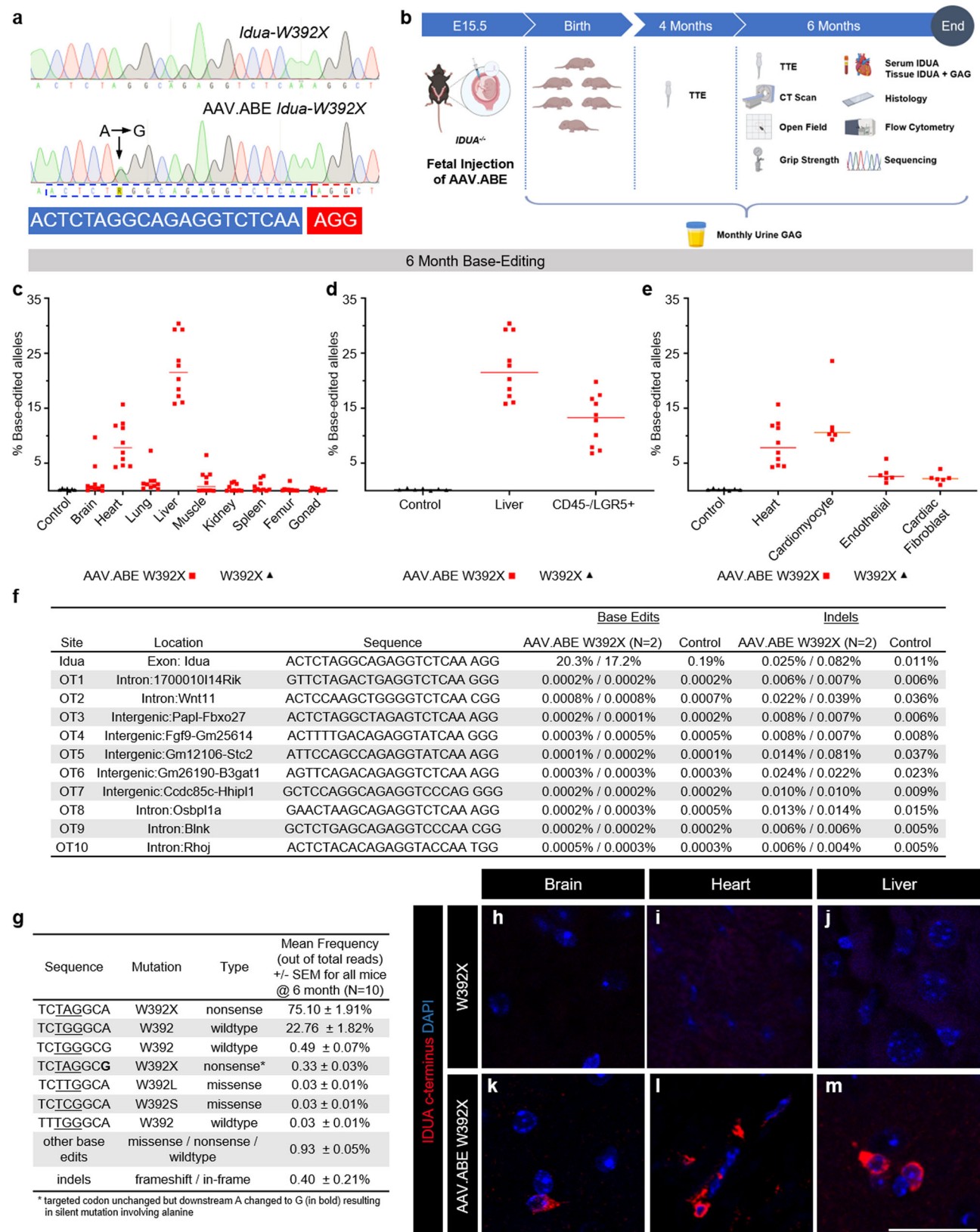

**f**

| Site | Location | Sequence | Base Edits | | Indels | |
|---|---|---|---|---|---|---|
| | | | AAV.ABE W392X (N=2) | Control | AAV.ABE W392X (N=2) | Control |
| Idua | Exon: Idua | ACTCTAGGCAGAGGTCTCAA AGG | 20.3% / 17.2% | 0.19% | 0.025% / 0.082% | 0.011% |
| OT1 | Intron:1700010I14Rik | GTTCTAGACTGAGGTCTCAA GGG | 0.0002% / 0.0002% | 0.0002% | 0.006% / 0.007% | 0.006% |
| OT2 | Intron:Wnt11 | ACTCCAAGCTGGGGTCTCAA CGG | 0.0008% / 0.0008% | 0.0007% | 0.022% / 0.039% | 0.036% |
| OT3 | Intergenic:Papl-Fbxo27 | ACTCTAGGCTAGAGTCTCAA AGG | 0.0002% / 0.0001% | 0.0002% | 0.008% / 0.007% | 0.006% |
| OT4 | Intergenic:Fgf9-Gm25614 | ACTTTTGACAGAGGTATCAA GGG | 0.0003% / 0.0005% | 0.0005% | 0.008% / 0.007% | 0.008% |
| OT5 | Intergenic:Gm12106-Stc2 | ATTCCAGCCAGAGGTATCAA AGG | 0.0001% / 0.0002% | 0.0001% | 0.014% / 0.081% | 0.037% |
| OT6 | Intergenic:Gm26190-B3gat1 | AGTTCAGACAGAGGTCTCAA AGG | 0.0003% / 0.0003% | 0.0003% | 0.024% / 0.022% | 0.023% |
| OT7 | Intergenic:Ccdc85c-Hhipl1 | GCTCCAGGCAGAGGTCCCAG GGG | 0.0002% / 0.0002% | 0.0002% | 0.010% / 0.010% | 0.009% |
| OT8 | Intron:Osbpl1a | GAACTAAGCAGAGGTCTCAA AGG | 0.0002% / 0.0003% | 0.0005% | 0.013% / 0.014% | 0.015% |
| OT9 | Intron:Blnk | GCTCTGAGCAGAGGTCCCAA CGG | 0.0002% / 0.0002% | 0.0002% | 0.006% / 0.006% | 0.005% |
| OT10 | Intron:Rhoj | ACTCTACACAGAGGTACCAA TGG | 0.0005% / 0.0003% | 0.0003% | 0.006% / 0.004% | 0.005% |

**g**

| Sequence | Mutation | Type | Mean Frequency (out of total reads) +/- SEM for all mice @ 6 month (N=10) |
|---|---|---|---|
| TCTAGGCA | W392X | nonsense | 75.10 ± 1.91% |
| TCTGGGCA | W392 | wildtype | 22.76 ± 1.82% |
| TCTGGGCG | W392 | wildtype | 0.49 ± 0.07% |
| TCTAGGCG | W392X | nonsense* | 0.33 ± 0.03% |
| TCTTGGCA | W392L | missense | 0.03 ± 0.01% |
| TCTCGGCA | W392S | missense | 0.03 ± 0.01% |
| TTTGGGCA | W392 | wildtype | 0.03 ± 0.01% |
| other base edits | missense / nonsense / wildtype | | 0.93 ± 0.05% |
| indels | frameshift / in-frame | | 0.40 ± 0.21% |

\* targeted codon unchanged but downstream A changed to G (in bold) resulting in silent mutation involving alanine

optimized ABE7.10 (ABEmax)[40] and gRNA with the target adenine mutation at position 6 within the 20-base protospacer sequence upstream of an NGG protospacer-adjacent motif (PAM) were packaged in two split-intein AAV9s (AAV.ABE. Idua) (Fig. 1a). In an initial screening experiment, two E15.5 Idua-W392X fetuses were injected via the vitelline vein with AAV.ABE.Idua. Sanger sequencing and next-generation sequencing (NGS) at 1 month of age demonstrated efficient heart (14.7, 13.1%) and liver (31.1, 22.4%) editing and low-level brain editing (1.9, 0.66%) (Fig. 1a).

**Fig. 1 In utero base editing in the Idua-W392X MPS-IH mouse model. a** Sanger sequencing of liver genomic DNA for E15.5 AAV.ABE.Idua injected and uninjected *Idua*-W392X fetuses at 1 month of age. PAM and protospacer in red and blue, respectively. On-target A→G edit noted. $N = 2$ biological replicates each. **b** Scheme for long-term genetic, biochemical, and phenotypic studies. **c** NGS, genomic DNA from indicated organs at 6 months of age. **d** NGS, genomic DNA from whole liver and sorted CD45−/LGR5+ liver cells at 6 months of age. **e** NGS, genomic DNA from whole heart and isolated cardiac myocytes, endothelial cells, and fibroblasts at 6 months of age. **f** NGS of the *Idua* on-target site and the top ten predicted off-target sites in liver genomic DNA at 6 months of age from two AAV.ABE.Idua recipients and one control. **g** Frequencies of base-edited and indel-bearing alleles assessed at 6 months of age via NGS of liver genomic DNA of in utero AAV.ABE.Idua recipients ($N = 10$). Underlined bases indicate the target codon. **h–m** Representative IHC of brain, heart, and liver of uninjected (**h–j**) and prenatal AAV.ABE.Idua (**k–m**) injected 6-month-old *Idua*-W392X mice stained with antibody specific for the C-terminus of IDUA (red) and DAPI (blue). Scale bar = 25 μm. **c–f** Control: uninjected 6-month-old *Idua*-W392X mice. **c–e** Experimental (red marks) $N = 10$ except for cardiomyocyte, endothelial, and cardiac fibroblast in which experimental $N = 6$; Control $N = 7$. AAV.ABE, represents mice injected with AAV.ABE.Idua. FACS gating strategy for **d** and **e** presented in Supplementary Fig. 6.

We next sought to understand the relationship between dual AAV transduction and editing levels. AAV9s containing GFP or mCherry transgenes (AAV.GFP.mCherry) were delivered to C57BL/6 (B6) mice at a 1:1 ratio via the vitelline vein at E15.5. Flow cytometry at 7 days post injection demonstrated GFP + mCherry+ populations in the brain, heart, and liver, which were compared to the concentration of AAV genomes in the same tissues yielding a significant relationship between dual viral delivery and fluorescent expression ($R^2 = 0.18$, $p = 0.034$) (Supplementary Fig. 2). This predictive regression model was then used to estimate the transduction efficiency of AAV.ABE.Idua based on AAV genomes and correlate transduction with editing efficiency at 30 days. The predicted dual transduction efficiency was 0.7% (95% CI 0.3, 1.2) in the heart and 0.4% (95% CI 0.2, 0.7) in the liver, suggesting substantially higher editing levels than viral transgene expression. Finally, the brains, hearts, muscles, livers, and uteri of two mothers of injected fetuses were evaluated by flow cytometry 7 days post injection and demonstrated no evidence of GFP or mCherry expression suggesting that no maternal transduction occurred.

Next, an experimental cohort of E15.5 *Idua*-W392X fetuses was intravascularly injected with AAV.ABE.Idua for long-term genetic, biochemical, and phenotypic analyses (Fig. 1b). Uninjected age- and sex-matched *Idua*-W392X mice and B6 mice served as positive and negative controls, respectively. Survival to birth after in utero injection of AAV.ABE.Idua (60.9%) was comparable to survival after in utero injection of phosphate-buffered saline (PBS) in *Idua*-W392X fetuses (60.0%). NGS at 6 months of age demonstrated efficient heart (~8.6%) and liver (~22.8%) editing in all experimental mice and low-level brain, lung, kidney, and spleen editing in some experimental mice (Fig. 1c–e). No editing was noted in genomic DNA from the ovaries ($N = 7$ mice; NGS: 0–0.55%) or sperm-containing epididymis ($N = 3$ mice; NGS: 0–0.39%) in experimental mice (control gonad; $N = 2$ mice; NGS: 0.16–0.34%) (Fig. 1c). NGS of liver genomic DNA from mice with high on-target editing did not demonstrate off-target editing above background and demonstrated low rates of nucleotide insertions or deletions (indels) and unwanted base changes (Fig. 1f, g). On-target editing was consistent with IDUA expression observed with IHC in these organs (Fig. 1h–m). Given the significant heart and liver editing, the importance of MPS-IH cardiac pathology, and the rationale that in utero editing can target progenitor cells, NGS editing efficiencies were evaluated in liver LGR5+ progenitor cells (~12.8%) and cardiac cell subpopulations including myocytes (~12.6%), endothelial cells (~3.0%), and fibroblasts (~2.3%) (Fig. 1d, e). Finally, genomic DNA from the uterus, heart, and liver of two dams in whom fetuses underwent editing following in utero AAV.ABE.Idua injection was assessed by NGS. Editing efficiencies in experimental dams (heart: 0.04%, 0.01%; liver: 0.07%, 0.21%; uteri: 0.02%, 0.02%) were comparable to uninjected

control livers and hearts (0.19–0.49%) suggesting no maternal editing occurred.

**In utero AAV.ABE.Idua treatment results in durable biochemical rescue in MPS-IH.** We next explored if in utero base editing caused durable biochemical corrections in *Idua*-W392X mice. Similar to humans with W402X MPS-IH, *Idua*-W392X mice have undetectable IDUA activity and increased urine and tissue GAGs[38]. In utero AAV.ABE.Idua treated mice demonstrated reduced urine GAGs as measured by colorimetric assay at all time points compared to untreated mice (Fig. 2a). At sacrifice (6 months of age), IDUA activity was increased in the serum, heart, and liver, and GAG levels were decreased in the heart and liver in AAV.ABE.Idua treated mice (Fig. 2b–f). Alcian blue staining of GAGs was decreased in the heart and liver, corroborating these biochemical findings (Fig. 3c–f, h–l, o–r, s).

**In utero AAV.ABE.Idua treatment improves the multi-organ MPS-IH disease phenotype.** Many MPS-IH patients succumb to cardiac pathology, characterized by increased cardiac GAGs, collagen deposition, and cardiac complications[3,4]. We performed detailed histopathologic and echocardiographic analyses of untreated *Idua*-W392X mice for comparison to B6 and in utero AAV.ABE.Idua treated *Idua*-W392X mice. Compared to untreated mice, treated mice demonstrated improved echocardiographic function and decreased aortic outlet collagen deposition on Masson's trichrome stain (Fig. 4a–s). This further substantiates our findings of cardiomyocyte editing and improved cardiac IDUA and GAG levels (Fig. 1).

Similar to MPS-IH patients, *Idua*-W392X mice have a shortened lifespan[38]. In our cohort of untreated *Idua*-W392X mice followed for long-term studies, we observed a 6-month mortality of 40%, with non-survivors noted to have increased ascending aortic dilation and reduced left ventricular ejection fraction on 4-month echocardiography (Fig. 4a–d, i–l). Survival of in utero treated mice was 100% and comparable to B6 mice (Fig. 4t).

MPS-IH causes significant progressive musculoskeletal morbidity in patients requiring multiple orthopedic surgeries despite successful HSCT[3,41,42]. Improved musculoskeletal outcomes following early versus late HSCT suggest that this irreversible pathology may benefit from in utero treatment[13]. Compared to untreated *Idua*-W392X mice, micro-computed tomography scans demonstrated reduced skull and femur cortical bone deposition and normalized facies (Fig. 5a–l and Supplementary Fig. 3) in in utero AAV.ABE.Idua treated *Idua*-W392X mice. In addition, treated mice subjectively had decreased lordosis and snout broadness at 6 months of age (Fig. 6). Finally, grip strength demonstrated significant improvement following in utero base editing (Fig. 5m). Given this finding, bicep muscles were assessed

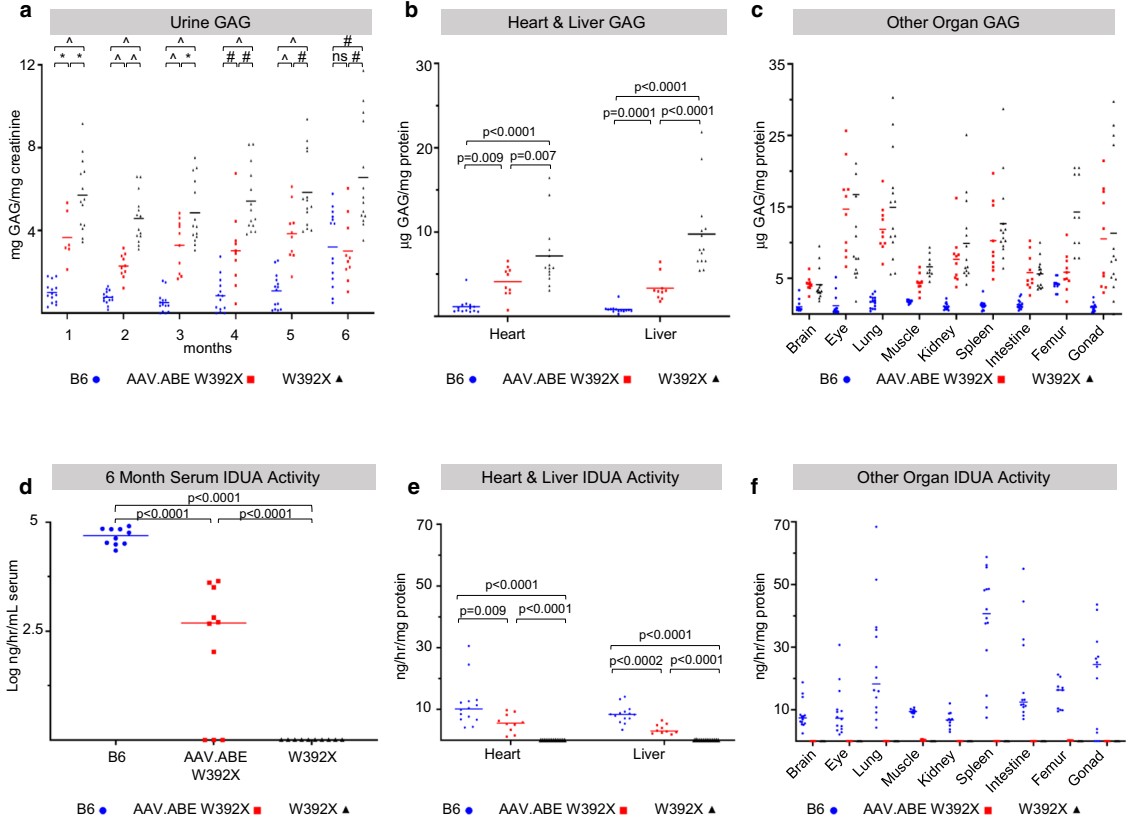

**Fig. 2 Durable improvement in biochemical parameters in Idua-W392X mice following in utero base editing. a** Urine GAGs were measured monthly in B6 (N = 14), *Idua*-W392X mice prenatally injected with AAV.ABE.Idua (N = 10 except for month 1, N = 6), and uninjected *Idua*-W392X mice (N = 14). **b, c** Tissue GAGs were measured at 6 months of age in the heart and liver (**b**) and other indicated organs (**c**) in B6 (N = 14, except for muscle/femur N = 10), prenatally injected mice (N = 10), and uninjected mice (N = 14, except eye, N = 13; and muscle/femur N = 10). **d** Serum IDUA activity was measured at 6 months of age in B6 (N = 10), prenatally injected mice (N = 10), and uninjected mice (N = 10). **e, f** Tissue IDUA activity was measured at 6 months of age in the heart and liver (**e**) and other indicated organs (**f**) in B6 (N = 14, except for muscle/femur N = 10), prenatally injected mice (N = 10), and uninjected mice (N = 14, except for muscle/femur N = 10). ^p < 0.0001; #p < 0.001; *p < 0.05. Exact p values outlined in Supplementary Table 1. Wilcoxon test for multiple comparisons used to assess urine GAG months 1–3, liver GAG, and heart and liver IDUA activity. All remaining statistical analyses used Student's t-test with α = 0.05. All tests two-sided. GAG glycosaminoglycans, IDUA α-L-iduronidase. AAV.ABE, represents mice injected with AAV.ABE. Idua. Source data are provided as a Source Data file.

for on-target editing efficiency in genomic DNA, IDUA enzyme activity, and GAG level. NGS demonstrated an on-target correction rate of ~1.65% (Fig. 1c) which was associated with detectable bicep IDUA enzyme activity (~3.2% of the normal IDUA activity in bicep muscles in B6 controls) compared to undetectable levels in W392X controls. This was consistent with a significant reduction in bicep GAG levels in prenatally AAV.ABE. Idua injected mice (~4.3 μg/mg protein) compared to W392X controls (6.6 μg/mg protein; p = 0.0016); however, the levels in treated mice were not equivalent to those in B6 controls (~1.7 μg/mg protein; p < 0.0001).

MPS-IH patients also have severe neurocognitive deficits. The hippocampi of 6-month-old prenatal AAV.ABE.Idua injected *Idua*-W392X mice were evaluated by Toluidine Blue histopathology and demonstrated nominal improvement in cellular vacuolization and no difference in cell size compared to uninjected *Idua*-W392X mice (Supplementary Fig. 4). However, Alcian Blue histology of the occipital cortex demonstrated decreased overall GAG staining (Fig. 3a, b, g, h, m, n, s). Mice were also subjected to open field testing to assess any effect of in utero base editing on this phenotype. Treated mice demonstrated reduced center entries with repetition suggesting appropriate habituation (Fig. 5n). However, lack of improvement in other parameters such as rearing is consistent with low-level editing in the brain

and emphasizes the need for strategies to enhance brain editing and/or brain IDUA activity (Fig. 5o).

**Postnatal treatment with AAV.ABE.Idua results in attenuation of the MPS-IH cardiac disease phenotype but induces an anti-Cas9 antibody response.** Although postnatal base editing would not prevent prenatal or early postnatal pathology, it could treat postnatally diagnosed patients. Therefore, five 10-week-old immunologically mature adult *Idua*-W392X mice were injected via the retroorbital vein with AAV.ABE.Idua and assessed for phenotypic and biochemical changes. At 4 months of age, NGS demonstrated editing of ~10.8% in liver genomic DNA (Fig. 7a) which was associated with a predicted dual AAV transduction efficiency of 0.9% (95% CI 0.32, 1.87). In addition, liver IDUA and GAG levels were significantly improved compared to untreated *Idua*-W392X mice (Fig. 7c, d) and by 5 months of age, urine GAGs were reduced compared to untreated *Idua*-W392X mice (Fig. 7b). At six months of age, NGS of genomic DNA demonstrated editing in the liver (mean% ± SD; 10.5% ± 5.2), the heart (5.0% ± 4.0), and in cardiac cell subpopulations including myocytes (4.0% ± 3.0), endothelial cells (2.1% ± 2.8), and fibroblasts (0.54% ± 1.1). Heart samples were also assessed for IDUA enzyme activity (3.9 ng/mg/hr ± 3.5) and GAG levels (3.3 μg/mg protein ± 1.2). The observed enzyme activity in postnatally

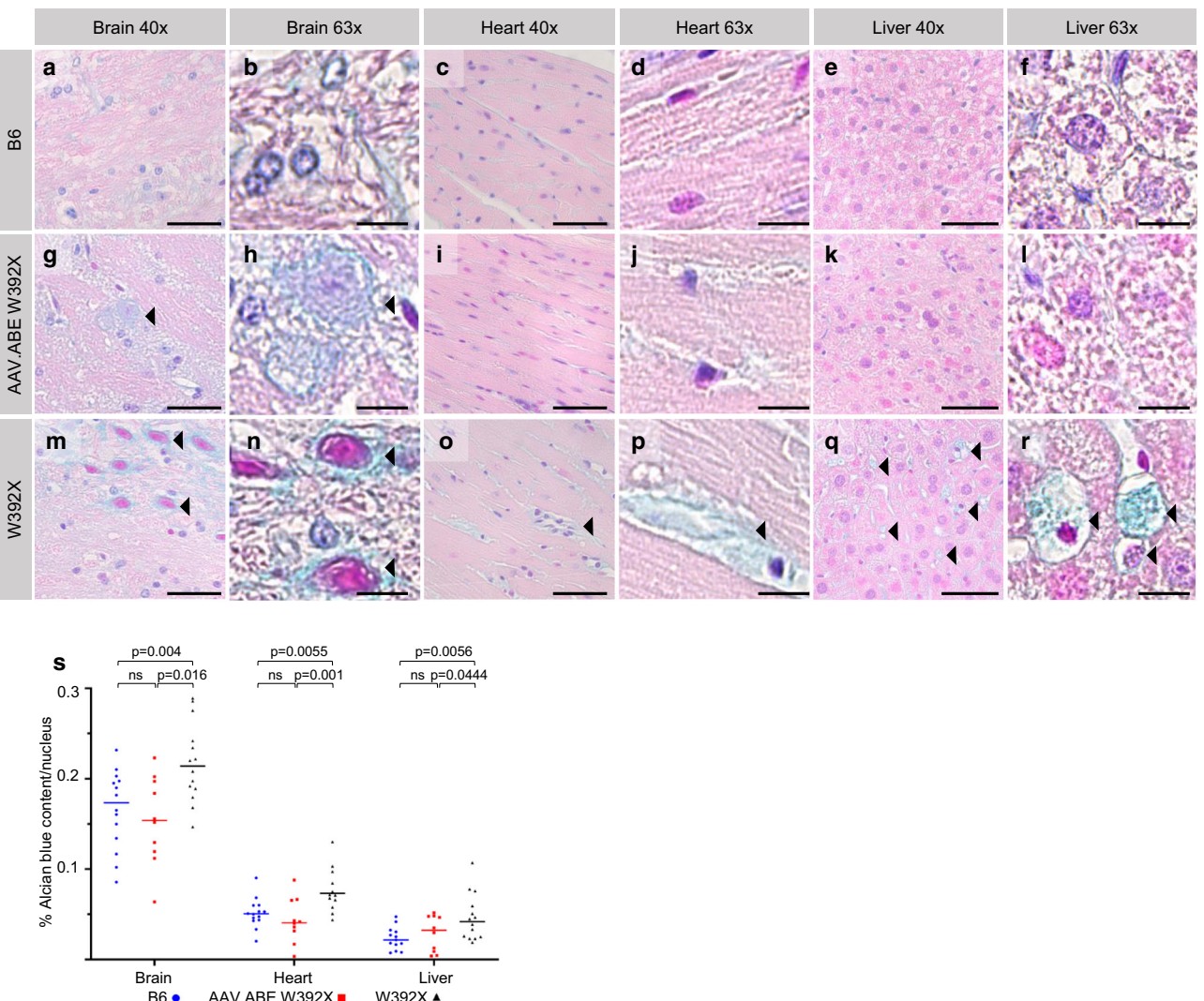

**Fig. 3 In utero base editing diminishes aberrant tissue GAG deposition in brain, heart, and liver. a–s** IHC with Alcian blue staining, which highlights accumulated GAGs, was performed in the brain, heart, and liver (three primary MPS-IH affected organs) in B6 ($N = 14$), *Idua*-W392X mice prenatally injected with AAV.ABE.Idua ($N = 10$), and uninjected *Idua*-W392X mice ($N = 14$, except $N = 12$ for heart). Representative sections: 40x brain sections featuring axonal tracts and periaxonal cells near cerebellar peduncles (**a, g, m**); 63x brain sections featuring axonal tracts and periaxonal cells near cerebellar peduncles (**b, h, n**); 40x left ventricles (**c, i, o**); 63x left ventricles (**d, j, p**); 40x liver (**e, k, q**); and 63x liver (**f, l, r**). **s** Alcian blue staining was quantified and normalized to number of visualized nuclei per high power field. Arrowheads denote Alcian blue staining of GAGs in vacuolated cells in the brain, the interstitial space in the heart, and in hepatocytes. (**a, g, m, c, i, o, e, k, q**) Scale bar = 50 μm. (**b, h, n, d, j, p, f, l, r**) Scale bar = 10 μm. Statistical analyses used Student's *t*-test with α = 0.05. All tests two-sided. GAG glycosaminoglycans, AAV.ABE indicates AAV.ABE.Idua. Source data are provided as a Source Data file.

injected mice were significantly improved compared to *Idua*-W392X controls ($p < 0.0001$), equivalent compared to in utero injected AAV.ABE.Idua mice (5.6 ng/mg/h ± 3.0, $p = 0.36$), and not equivalent to B6 controls (11.6 ng/mg/h ± 7.4, $p = 0.014$). Similarly, heart GAG levels were significantly improved compared to *Idua*-W392X controls (7.2 μg/mg protein ± 3.9, $p = 0.0063$), equivalent compared to in utero injected AAV.ABE.Idua mice (4.1 μg/mg protein ± 1.8, $p = 0.24$), and not equivalent to B6 controls (1.1 μg/mg protein ± 1.0, $p = 0.0047$).

Echocardiography at 4 months of age in *Idua*-W392X mice postnatally injected with AAV.ABE.Idua revealed reduced ascending aorta and aortic valve diameters but similar left ventricle size, ejection fraction, and fractional shortening compared to untreated *Idua*-W392X mice (Fig. 7e–i). Between 4 and 6 months of age, ascending aorta diameter (4-to 6-month differences of means $\Delta\bar{x} = 0.47$, $p = 0.23$), aortic valve diameter

($\Delta\bar{x} = -0.03$, $p = 0.65$), left ventricle size ($\Delta\bar{x} = 0.14$, $p = 0.63$), ejection fraction ($\Delta\bar{x} = -2.90$, $p = 0.63$), and fractional shortening ($\Delta\bar{x} = -1.76$, $p = 0.62$) did not worsen and at 6 months of age were not significantly different than either prenatally treated mice or untreated *Idua*-W392X controls. Notably, the death of the most diseased *Idua*-W392X controls between 4 and 6 months diminishes the power of comparison at 6 months. Consequently, the relative improvement of postnatally treated mice is likely statistically underestimated at 6 months. In sum, these findings suggest an attenuation of disease progression in mice that underwent postnatal injection of AAV.ABE.Idua.

Finally, one rationale for in utero gene editing is the absence of an immune response to the transgene product. We previously demonstrated an SpCas9 immune response following postnatal but not in utero adenoviral delivery of an SpCas9-based cytosine base editor[34]. Adenoviral vectors are highly immunogenic, and

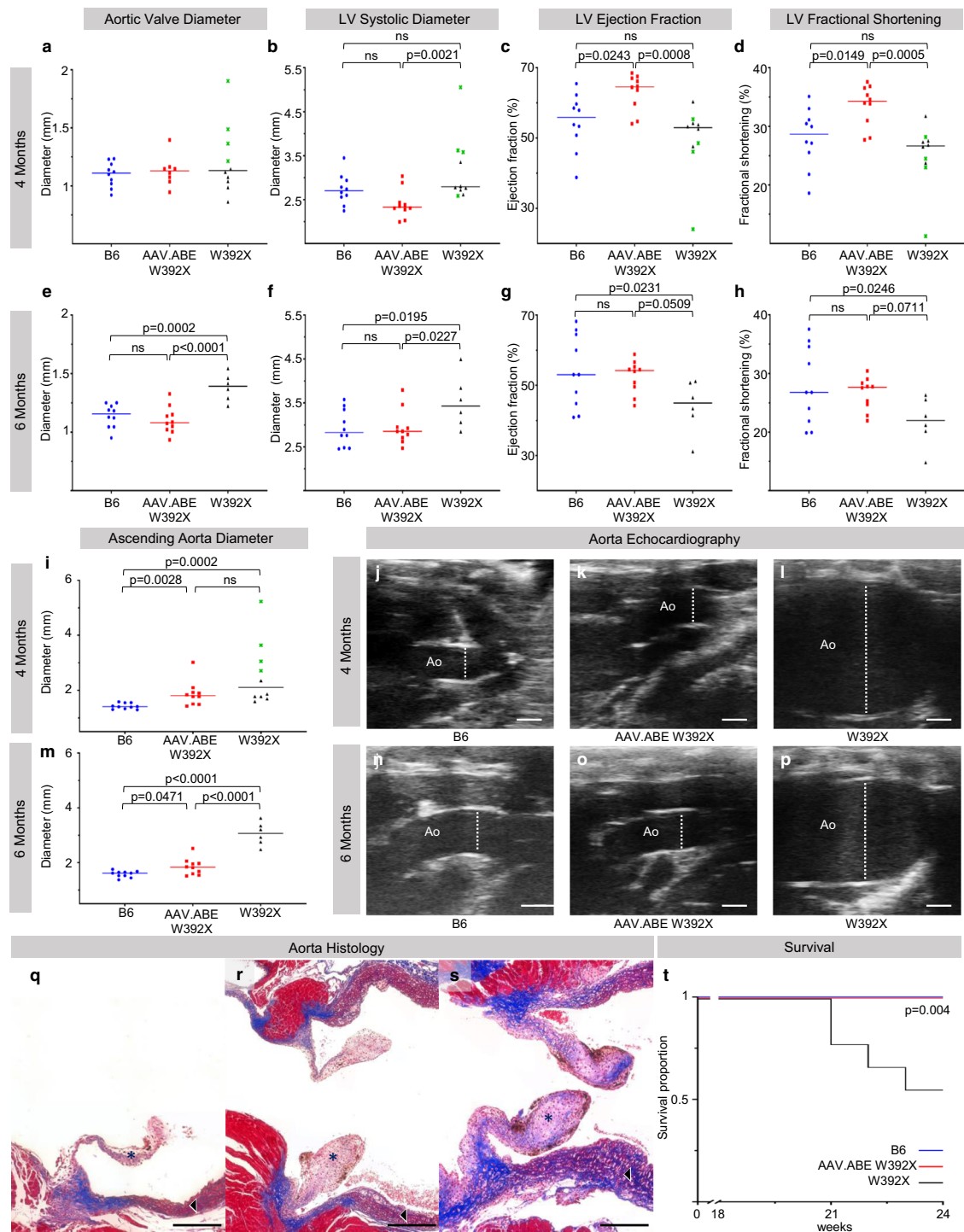

**Fig. 4 In utero base editing improves cardiac function and survival in Idua-W392X mice. a–d** Echocardiographic parameters were measured in 4-month-old B6 ($N=10$), *Idua*-W392X mice prenatally injected with AAV.ABE.Idua ($N=10$, except 4-month aortic valve diameter, $N=8$), and uninjected *Idua*-W392X mice ($N=10$). Mice that died prior to 6 months of age are denoted with a green X. **e–h** Echocardiographic parameters were measured in 6-month-old B6 ($N=10$), prenatally injected mice ($N=10$), and surviving uninjected mice ($N=6$). **i–l** Ascending aorta diameter and representative ultrasounds from 4-month-old B6 ($N=10$), prenatally injected mice ($N=10$), and uninjected mice ($N=10$). Mice that died prior to 6 months of age are denoted with a green X. Scale bar $=1\,mm$. **m–p** Ascending aorta diameter and representative ascending aorta ultrasounds from 6-month-old B6 ($N=10$), prenatally injected mice ($N=10$), and surviving uninjected mice ($N=6$). Scale bar $=1\,mm$. **q–s** Representative Masson's trichrome stain highlighting collagen deposition in aortic outlets including aortic valve (\*), and ascending aorta (black and white arrow) in 6-month-old B6, prenatally injected mice, and uninjected mice. Red stains muscle, pink stains cytoplasm, black stains nuclei, and blue stains collagen. Images constructed with 40x tile stitch. Scale bar $= 200\,\mu m$. Staining was repeated in five biological replicates per sample. **t** Six-month survival analysis comparing B6 ($N=10$), prenatally injected mice ($N=10$), and uninjected mice ($N=10$). Wilcoxon test for multiple comparisons used to assess 4 month ascending aorta diameter; Mantel–Cox method used to assess survival; Student's *t*-test used for all other comparisons; $\alpha=0.05$. All tests two-sided. LV left ventricle, Ao aorta. AAV.ABE indicates AAV.ABE.Idua. Source data are provided as a Source Data file.

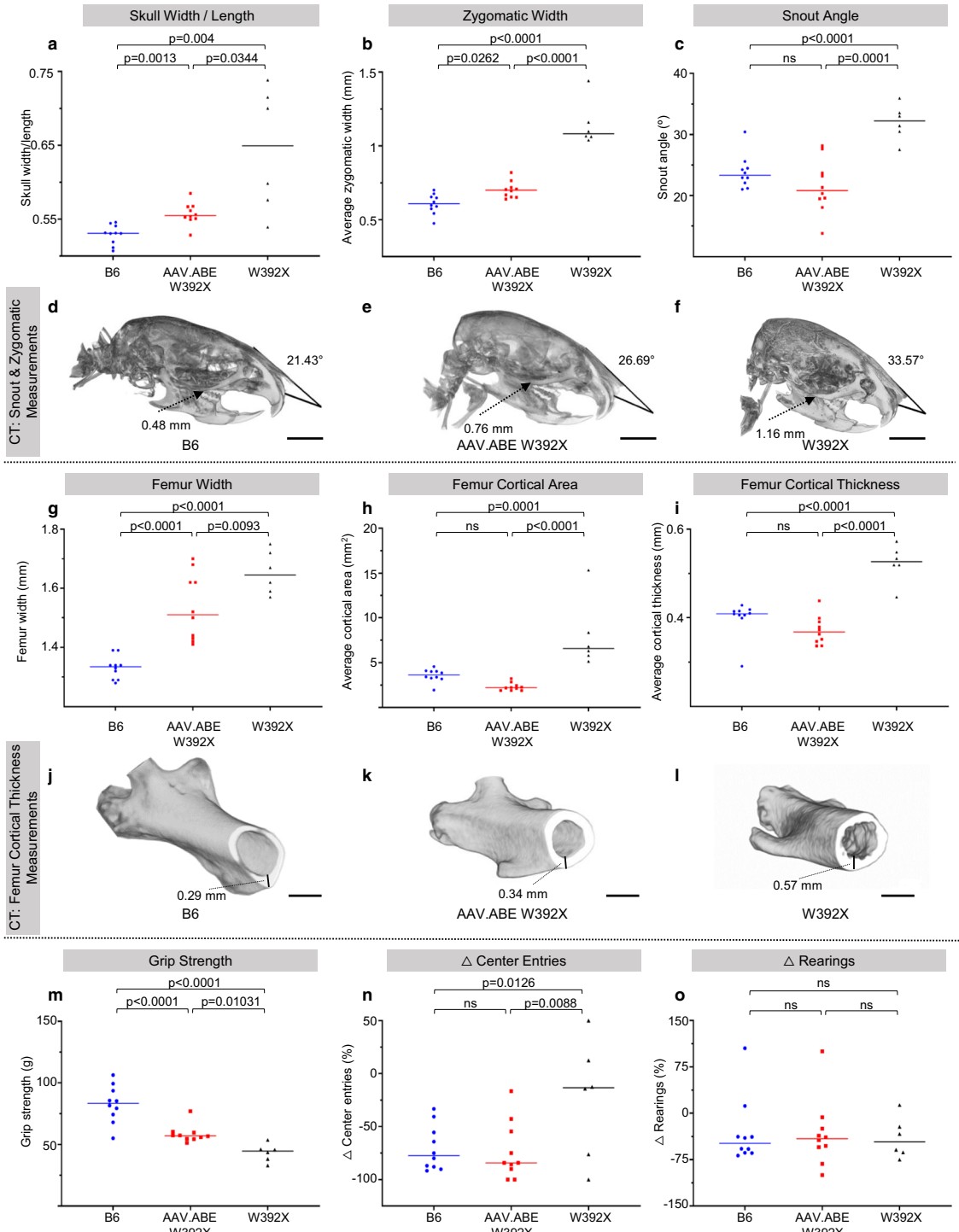

**Fig. 5 In utero base editing improves the musculoskeletal and neurobehavioural phenotype in Idua-W392X mice. a–c** CT face parameters were measured in 6-month-old B6 (N = 10), 6-month-old *Idua*-W392X mice prenatally injected with AAV.ABE.Idua (N = 10), and surviving uninjected *Idua*-W392X mice (N = 6). **d–f** 3D CT reconstructions of representative skulls from 6-month-old B6, prenatally injected mice, and surviving uninjected mice with snout angle and zygomatic arch measurements indicated. Scale bar = 5 mm. **g–i** CT femur parameters were measured in 6-month-old B6 (N = 10), prenatally injected mice (N = 10), and surviving uninjected mice (N = 6). **j–l** 3D CT reconstructions of representative femurs from 6-month-old B6, prenatally injected mice, and uninjected mice demonstrating cortical thickness measurements and surface contour. Scale bar = 1 mm. **m** Grip strength was measured in 6-month-old B6 (N = 10), prenatally injected mice (N = 10), and surviving uninjected mice (N = 6). **n, o** Open field test measurements of habituation between trials of testing in 6-month-old B6 (N = 10), prenatally injected mice (N = 10), and surviving uninjected mice (N = 6) Δ reflects change in parameter between the first and third trial. Wilcoxon test for multiple comparisons was used to assess skull width/thickness; Student's *t*-test used for all other comparisons; α = 0.05. All tests two-sided. CT computed tomography scan. AAV.ABE indicates AAV.ABE.Idua. Source data are provided as a Source Data file.

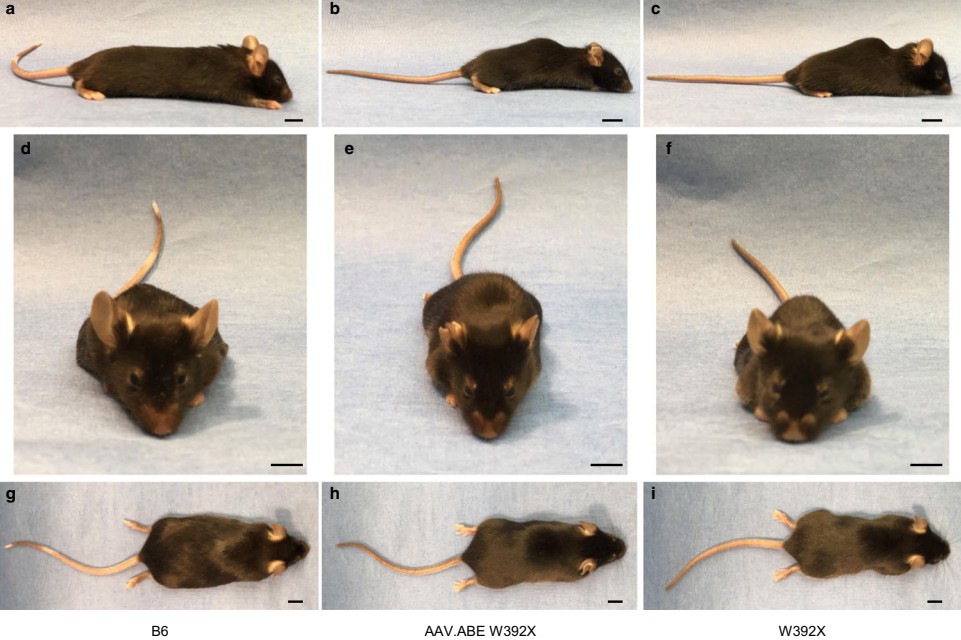

**Fig. 6 In utero base editing ameliorates the facial and skeletal abnormalities in Idua-W392X mice. a–i** Representative photographs of 6-month-old female B6 mice (**a**, **d**, **g**), *Idua*-W392X mice prenatally treated with AAV.ABE.Idua (**b**, **e**, **h**), and untreated *Idua*-W392X mice (**c**, **f**, **i**). Scale bar = 1 cm.

thus we sought to determine if a similar benefit to in utero base editing exists following delivery of ABEmax via an AAV. As in the previous study, anti-SpCas9 antibodies were noted in the serum of adult but not fetal AAV.ABE.Idua recipients 1 month following injection (Fig. 8).

## Discussion

CRISPR-mediated base editing has garnered excitement for the possibility of curing genetic diseases that result in significant morbidity and mortality and for which there exist no adequate treatments. Although many genetic diseases would best be treated with gene editing after birth, the natural characteristics of the developing fetus as well as the timing of onset of some diseases make prenatal therapeutic base editing an attractive approach. The best candidate diseases are those which can be diagnosed prenatally, which cause significant morbidity and/or mortality before or immediately after birth, and in which pathology begins before birth and irreversibly accumulates throughout the lifetime of the individual.

MPS-IH and other lysosomal storage diseases represent attractive targets for prenatal base editing as the multi-organ disease pathology is progressive and begins prior to birth. Furthermore, the bone and cardiac pathologies are not responsive to the limited current postnatal treatments including enzyme replacement and HSCT[3,41–43]. As an alternative, an ongoing trial of autologous transplantation of hematopoietic stem/progenitor cells treated with a lentivirus containing the *IDUA* transgene has had early encouraging results[44,45]. This approach, however, is limited to postnatal treatment and is likely to have the same phenotypic correction seen following allogenic HSCT. In our study, prenatal base editing was shown to correct the disease-causing mutation in multiple disease-affected organs in the *Idua*-W392X MPS-IH mouse model, with high-level correction in the liver and cardiomyocytes. Efficient prenatal base editing was associated with amelioration of the cardiac and musculoskeletal phenotypes at 6 months of age.

In addition to allowing treatment of disease before the onset of pathology, therapeutically favorable features of the developing fetus include small fetal size and access to progenitor cells of multiple organs[29]. We demonstrated efficient editing of LGR5[+] hepatic progenitor cells, which help populate the liver in fetal life and which can facilitate hepatic regeneration as a response to injury in the adult[46]. Small fetal size provides the ability to deliver high doses of gene editing technology per weight, which is of critical importance when a therapy requires multiple vectors—split base editors[26] or HDR with large template sequences[21]—as the same cell will need to be transduced twice. In this study, we predicted low but comparable levels of dual AAV transduction in both pre- and postnatally injected mice. Notably, our approximation of transduction is based on the delivery of fluorescent reporters in which kinetics may be different than that of AAV. ABE.Idua. In addition, despite similar viral dose and predicted transduction levels, we found that the level of liver editing in in utero injected mice was substantially higher than that in postnatally injected mice 1-month post injection. Given the difference in editing between pre- and postnatally injected mice not explained by viral delivery or predicted transduction, there is likely a potentiating mechanism for editing. Possible explanations include a competitive advantage for prenatally gene-edited cells, the accessibility of progenitor cells which are increasingly quiescent with age, and a higher rate of cellular replication and thus expansion of edited cells earlier in life. Collectively, these observations suggest that strategies to enhance transduction such as increasing the dose of AAV9 may improve editing efficiency. This, however, must be considered in the context of toxicity associated with high dose AAV wherein systemic delivery of greater than 1E14 vg/kg may result in dose limiting toxicities in humans[47,48]. Importantly, the toxic dose range of AAV has not been thoroughly evaluated in the prenatal setting. However, given the immaturity of the fetal immune system, it is conceivable that the maximal tolerated dose may be different than that observed in postnatal studies. Therefore, a thorough characterization of the in utero AAV dose-toxicity relationship should be conducted in large animal preclinical studies prior to clinical translation. Notably, in our experiments, the observed increased editing levels in prenatally treated mice were achieved using only ~6.5E13 vg/kg and 5% of the volume of viral vector delivered in adults. Given

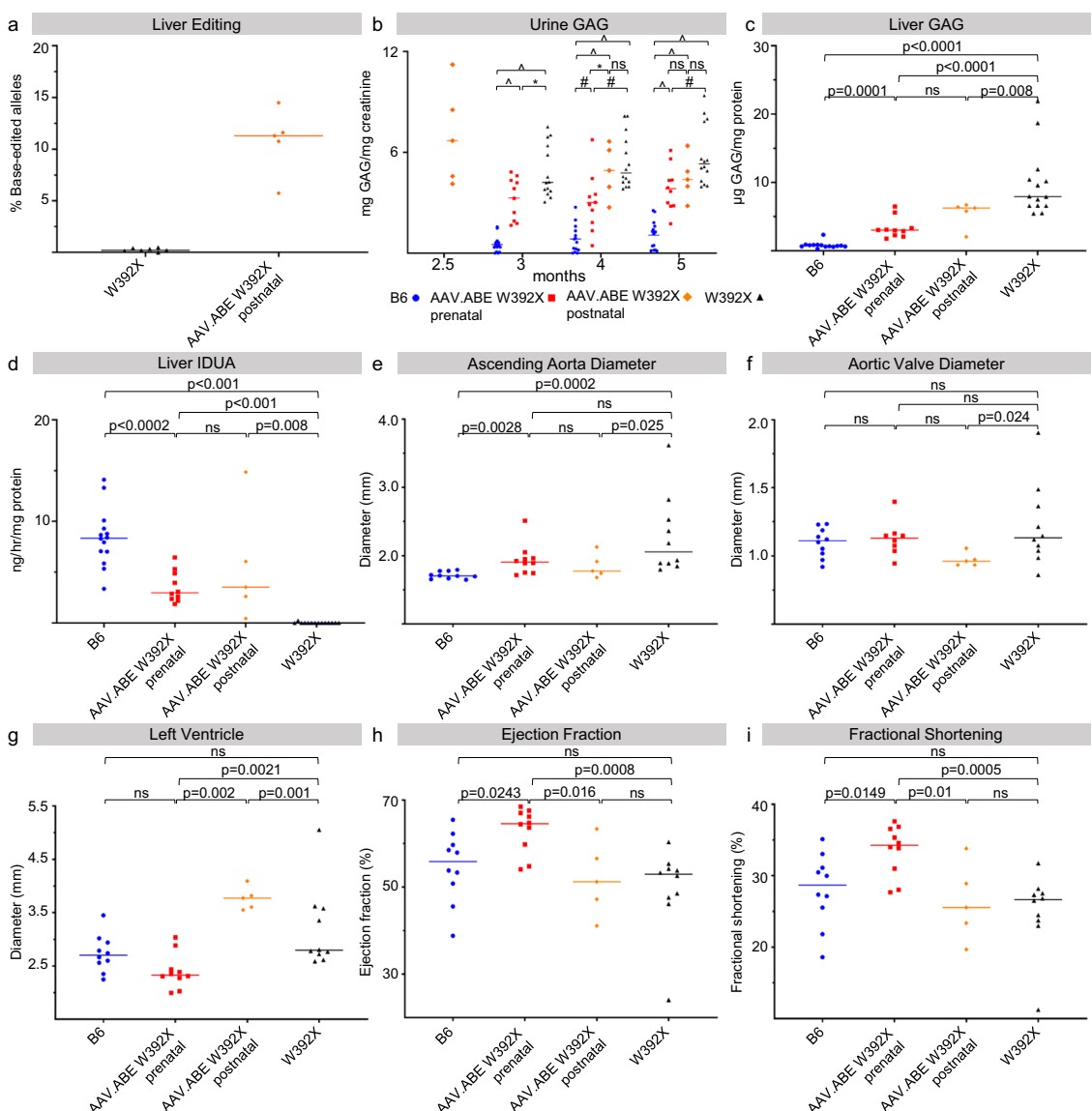

**Fig. 7 Base editing in adult Idua-W392X mice partially corrects biochemical and cardiac abnormalities.** Ten-week-old *Idua*-W392X mice were intravascularly injected with AAV.ABE.Idua. **a** A liver biopsy was performed at 4 months of age and genomic DNA was assessed by NGS for on-target *Idua* editing in postnatal *Idua*-W392X injected mice (N = 5) and *Idua*-W392X controls (N = 7). **b** Prior to postnatal injection, urine was collected for baseline urine GAG levels (2.5 month values) and urine GAGs were subsequently measured in B6 mice (N = 14), postnatal *Idua*-W392X injected mice (N = 5), prenatal *Idua*-W392X injected mice (N = 10), and uninjected *Idua*-W392X mice (N = 14) at 3, 4, and 5 months of age. **c, d** Liver GAGs (**c**) and IDUA activity (**d**) were measured from the liver biopsy specimen in postnatal *Idua*-W392X injected mice and compared to levels from B6 mice (N = 14), prenatal *Idua*-W392X injected mice (N = 10), and uninjected *Idua*-W392X mice (N = 14). **e–i** Postnatal injected *Idua*-W392X mice (N = 5) underwent echocardiogram at 4 months of age and the ascending aorta diameter (**e**), aortic valve diameter (**f**), left ventricle diameter (**g**), LV ejection fraction (**h**), and LV fractional shortening (**i**) were compared to those in 4-month-old B6 mice (N = 10), prenatal *Idua*-W392X injected mice (N = 10), except 4-month aortic valve diameter, (N = 8), and uninjected *Idua*-W392X mice (N = 10). ^p < 0.0001; #p < 0.001; *p < 0.05. Exact p values outlined in Table S1. Wilcoxon test for multiple comparisons used to assess 3-month urine GAGs and liver IDUA activity. All remaining statistical analyses used Student's t-test with α = 0.05. All tests were two-sided. GAG glycosaminoglycans, IDUA a-L-iduronidase. AAV.ABE indicates AAV.ABE.Idua. Source data are provided as a Source Data file.

the engineering and financial challenges related to gene therapy dosing, in utero gene editing may thus be an effective means to maximize dosing value.

In addition, prenatal gene editing has the potential to take advantage of BBB permeability—which progressively diminishes during fetal development—to target the CNS. We demonstrated variable low-level editing in the brain following prenatal base editing (highest level of brain editing = 9%), with a concomitant reduction in GAG staining on Alcian blue histology as well as improvement in some components of the open-field test.

However, the degree of hippocampal cellular vacuolization on Toluidine staining was not different between treated and untreated animals. Although low-level brain editing may contribute to some phenotypic improvement, we did not detect significant tissue IDUA activity in the brain following PBS perfusion. The nominal pathologic improvements therefore likely result from secretion of IDUA into serum, which may affect the brain and other low or unedited organs. Alternative approaches to optimizing brain editing such as delivering ABE earlier in gestation, modulating AAV serotypes, and/or directly injecting

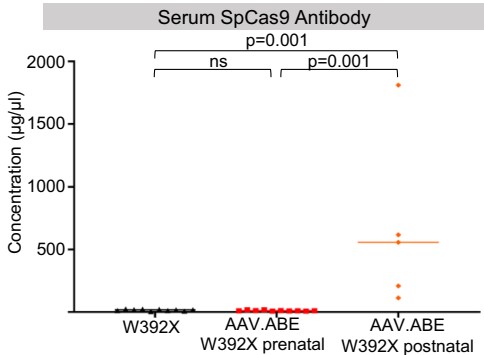

**Fig. 8 Postnatal base editing in Idua-W392X mice is associated with anti-SpCas9 specific antibodies.** Ten-week-old *Idua*-W392X mice were intravascularly injected with AAV.ABE.Idua. Serum from *Idua*-W392X mice prenatally (*N* = 10) or postnatally (*N* = 5) treated with AAV.ABE.Idua was harvested 1-month post injection and assessed for anti-SpCas9 antibodies. W392X = uninjected control serum (*N* = 10). Two-tailed Wilcoxon test for multiple comparisons was used with α = 0.05. AAV.ABE indicates AAV. ABE.Idua. Source data are provided as a Source Data file.

ABE intracranially as has been performed in studies of replacement gene therapy[49] are important considerations for future experiments in MPS-IH and other diseases with a neurologic phenotype.

Moreover, less stringent immunologic and physical barriers are present in the developing fetus compared to postnatal recipients. We demonstrated the presence of anti-SpCas9 antibodies in adult recipients of AAV.ABE.Idua that were not elicited following prenatal AAV.ABE.Idua injection. This finding substantiates our previous results in which an adenovirus was used to deliver a cytosine base editor[34] and highlights that even with a less immunogenic AAV delivery mechanism, a potential immunologic benefit exists for prenatal gene editing. Furthermore, the ability to avoid an immune response to therapy may not only protect edited cells from immune clearance and preserve long-term gene editing but also raises the possibility of delivering additional postnatal booster therapy[50]. Importantly our study does not address the direct causality between the presence of SpCas9 antibodies and the differential editing levels seen in postnatally versus prenatally treated mice. This difference is likely multifactorial due to, but not limited to, immune response, stem cell biology, and timing of treatment. Nonetheless, the potential benefit of a naïve fetal immune system may still be an important consideration for diseases in which a high level of correction and therefore additional booster treatments are necessary to rescue a phenotype.

Although ontological advantages for prenatal gene editing may exist, some patients may not be diagnosed before birth and the clinical safety of base editing technology initially needs to be demonstrated in a postnatal recipient. As such, we evaluated the efficiency of base editing and a limited phenotype assessment in postnatal recipients in the context of our prenatal experiments. Postnatally treated mice demonstrated efficient on-target editing in the heart and liver which was associated with improvement in cardiac parameters. Notably, although disease controls demonstrated rapid cardiac decline between 4 and 6 months even resulting in mortality, postnatally treated mice demonstrated cardiac disease at 4 months with attenuated progression between 4 and 6 months of age and no death. This difference at 4 months of age between prenatally and postnatally treated mice is likely due to prolonged IDUA exposure from the time of birth in those undergoing in utero base editing. This suggests that postnatal

treatment can have encouraging phenotypic benefits; though, maximal effect may be realized with earlier therapy.

In addition to demonstrating treatment efficacy, key elements for clinical translation of prenatal gene editing for MPS-IH include feasible prenatal diagnosis and ensuring maternal-fetal safety. In families with no known history of MPS-IH, commercial carrier screens can identify pregnancies at risk for MPS-IH (e.g., Progenity Preparent Carrier Test and Integrated Genetics 500 Plus Panel). Alternatively, in families at high-risk of MPS-IH or with known affected offspring, minimally invasive ultrasound guided amniocentesis and chorionic villus sampling (CVS) have been used to prenatally diagnose MPS-IH affected fetuses with high specificity and sensitivity based on elevated GAGs and reduced IDUA enzyme activity[51,52]. Moreover, amniocentesis in five pregnant women led to prenatal diagnosis in six fetuses on the basis of direct sequencing of the IDUA gene[53]. These studies demonstrate that prenatal diagnosis of MPS-IH, including the genetic diagnosis of the causative mutation, is currently feasible. Although invasive, these procedures are routinely performed (greater than 200,000/year in the United States[54]) and are associated with very low maternal and fetal complication rates (amniocentesis, 0.11% fetal loss; CVS, 0.22% fetal loss[55]). Recently, advances in noninvasive prenatal testing technology support the possibility that MPS-IH could be diagnosed via a noninvasive test. The optimization of techniques for collecting cell-free fetal DNA from maternal serum has allowed targeted sequencing of the fetal genome for disease-causing alleles and highlights the possibility of a noninvasive prenatal genetic test for MPS-IH[56]. This technique and other noninvasive prenatal genomic diagnostic technologies are likely to be increasingly accessible in the timeline during which prenatal therapies may be available for clinical use.

For all in utero procedures, the safety of both the fetus and mother is of utmost importance for potential clinical translation. Major risks include procedural morbidity, vector toxicity, and unintended mutagenesis. A limitation of prenatal intervention in the mouse is procedural risk related to injection. We demonstrated no maternal loss and a ~60% survival-to-birth for *Idua*-W392X fetuses injected with either PBS or AAV.ABE.Idua, which is comparable to previously published survival rates following injection of bone marrow cells in the mouse model[57]. The parity between the two groups in the current study suggests that fetal mortality is related to technical aspects of the procedure and not directly associated with AAV at the dose used. Similar technical challenges are not expected for clinical translation. Specifically, clinical translation would involve intravascular delivery of gene editing technology via the umbilical vein. This would be a minimally invasive procedure performed under ultrasound guidance and local anesthesia without a maternal laparotomy. The procedural risks for an umbilical vein injection would be similar to those documented for umbilical vein transfusions which are performed routinely, often multiple times during pregnancy, for select fetuses diagnosed with anemia. In a series of 937 fetal blood transfusions for fetal anemia, a 1.2% maternal/fetal complication rate (e.g., emergency cesarean section, fetal bradycardia, preterm prelabour rupture of membranes, and infection) and a 0.6% fetal loss rate were noted[58]. Importantly, these procedures were performed in anemic fetuses and complications may be lower in nonanaemic fetuses. In addition to procedure-related risks, maternal editing and off-target mutagenesis in the fetus are critical safety considerations. As such, we found no evidence of fluorescent expression in mothers of fetuses injected with AAV. GFP.mCherry and no evidence of base editing in the uteri, hearts, or livers of mothers of fetuses injected with AAV.ABE.Idua. These organs reflect both those in direct proximity to the fetal injection as well as those that align with AAV9 viral tropism.

Finally, an assessment of off-target mutagenesis at the top ten predicted sites in genomic DNA from the fetal liver revealed negligible unintended base edits and indels. Although we did not appreciate any off-target DNA editing activity in the current proof-of-concept study, ABEmax has also been shown to induce transcriptome-wide RNA off-target activities[59]. This potential limitation may be addressed with continued evolution of base editing enzymes, however, future studies in large animal models and human samples assessing additional off-target analyses including transcriptome-wide RNA off-target activities are essential prior to clinical translation. Nonetheless, these findings suggest that fetal injection of AAV.ABE.Idua portends limited procedural, genetic, and vector risks to the mother and fetuses in the mouse model.

Despite the findings demonstrated here, there are caveats to the study and critical steps that are required prior to clinical translation. First, the ability to assess the neurobehavioural phenotype and thus any improvement in this parameter is limited in the mouse model and calls for translation to clinically relevant large animals. In addition, although the objective of our study was to assess gross phenotypic correction including changes in total GAG levels, future translational work would benefit from GAG subtype analysis utilizing liquid chromatography/mass spectrometry to identify organ-specific saccharide changes following editing. Specifically, heparan sulfate and keratan sulfate levels have been shown to be related to CNS and bone pathology, respectively and this analysis will add value to urinary GAG levels for clinical monitoring[15,60–66]. Furthermore, although treated and B6 control mice had grossly comparable mobility and function, some bone parameters were reduced in treated compared to B6 mice. Whether these findings represent a clinically meaningful effect of the treatment requires investigation in additional models. Second, although our estimation of transduction efficiency as it relates to editing efficiency was determined by delivering $10^{10}$ vector copies of AAV.GFP.mCherry or AAV.ABE.Idua, the delivered dose did differ by ~2x (2.5E10 vs 5E10) due to experimental conditions. Notably, as prior work has demonstrated that AAV9 dose and tissue titers are linear in the $10^{10}$ dose range[67], the described correlations between transduction percentage and tissue viral titers and the conclusion that enhancing transduction has the potential to increase gene editing are unlikely to be altered by these differences. However, studies of the dose-editing relationship are merited in future investigations. Third, our experiments utilized uninjected and PBS-injected disease mice as controls due to limited availability of time-dated pregnancies. Future studies including those in large animals may benefit from the addition of a vector control with non-targeting gRNA in order to more thoroughly interrogate the on- and off-target efficiencies and safety of the selected base editor. Finally, the in utero mouse model does not fully recapitulate the corresponding techniques that would be employed in human prenatal interventions. This abrogates the extrapolation of model-specific safety data to clinical practice. As such, additional large animal studies that better approximate human anatomy and procedural technique are merited to assess maternal and fetal risk comprehensively. Thus, although we demonstrated the efficacy of a clinically relevant AAV9 delivery approach, future work to compare the efficiency, specificity, and safety of other delivery approaches including lipid nanoparticles, evolved AAVs, and aptamers is warranted.

In summary, in utero base editing was associated with improved survival and amelioration of metabolic, skeletal, and cardiac disease. This study advances our previous work, which was limited by its delivery approach and in which editing was restricted to the liver in a disease model that has a viable postnatal treatment option[34]. Specifically, we have now established the feasibility of in utero base editing, using a clinically relevant dual AAV, to correct a disease-causing mutation in multiple organs and ameliorate the multi-organ disease phenotype in MPS-IH. Due to the prenatal onset of pathology, potential for noninvasive prenatal diagnosis, and progressive and morbid nature of the disease, MPS-IH and other LSDs represent attractive targets for clinical in utero gene therapy/editing. Although the safety of these approaches for mothers and fetuses needs to be rigorously characterized prior to clinical translation, this proof-of-concept study offers hope for genetic diseases with limited postnatal treatments.

## Methods

**Selection of guide RNAs.** The gRNA targeting the loxP sites flanking the mT gene in $R26^{mTmG/+}$ mice was selected based on our previous publication[34] and its predicted high on-target efficiency and low off-target effects as determined by the online tool CRISPOR[68]. The loxP targeting protospacer and PAM was 5′-ATTA TACGAAGTTATATTAA | GGG-3′. The gRNA for the Idua gene was selected following visual inspection of the sequence at the site of the G→A W392X mutation which identified an AGG PAM and protospacer in which the target adenine is at position 6. Specifically, the Idua targeting protospacer and PAM was 5′-ACTCTAGGCAGAGGTCTCAA | AGG-3′.

**Generation of AAV vectors.** AAV2/9 CMV-GFP and AAV2/9 CMV-mCherry were purchased from Vector Biolabs (Malvern, PA). AAV2/9 serotype vectors containing SpCas9 and the loxP targeting gRNA (for mTmG studies) or ABEmax and the Idua targeting gRNA (for MPS-IH studies) were generated by Vector Biolabs (Malvern, PA). Due to the size of the ABE and SpCas9 and the limited packaging capacity of AAV, a split AAV intein-mediated approach was used such that half of the ABE or SpCas9, the gRNA, and intein transgenes were delivered in one AAV and a second AAV was used to deliver the corresponding intein and other half of the ABE or SpCas9 transgenes[39,69] (Supplementary Fig. 5a, b).

Sequences for the framework of the split-intein SpCas9 were obtained from Truong et al.[69]. Briefly, the CAG promoter driving the SpCas9 to Glu573 fused to the N-terminal Nostoc punctiforme (Npu) DnaE intein was replaced with the CMV chimeric intron promoter from pCI (Promega, Madison, WI) using standard molecular cloning techniques. Following the polyadenylation signal, the gRNA cassette—a kind gift from Kiran Musunuru (Addgene Plasmid #64711)—containing the loxp targeting protospacer sequence (5′- ATTATACGAAGTTATATTAA-3′) was inserted with the plasmid designated as pAAV-CMV-SpCas9-N-sgloxp. The corresponding C-terminal Npu intein fused to the remaining SpCas9 sequences under the CAG promoter was inserted in the pZac backbone—a kind gift from the Gene Therapy Program, University of Pennsylvania—and was designated pAAV-CAG-SpCas9-C.

The pCMV_ABEmax plasmid (Addgene plasmid #112095), a kind gift from David Liu, served as the template for the split-intein ABE. Using Q5 High Fidelity DNA Polymerase (NEB, Ipswich, MA) amplified N- and C-termini of the ABEmax were ligated at the SpCas9 Glu573 and Cys574 to the codon optimized N- and C-termini of the Npu intein, respectively. To simplify ligation, codons for Leu564 and Lys565 were altered to an AflII site (CTT | AAG). Those sequences were inserted between AAV ITRs in a plasmid backbone containing the CBh promoter[70] and WPRE3-bGH polyadenylation signal[71] and designated pAAV-CBh-ABEmax-N and pAAV-CBh-ABEmax-C. The U6 cassette containing the mouse Idua protospacer sequence (5′-ACTCTAGGCAGAGGTCTCAA-3′) was inserted with the plasmid now designated pAAV-CBh-ABEmax-C-sgmIDUA (Supplementary Fig. 5c, d).

Prior to submission of transfer plasmids to Vector Biolabs for production and purification of high titer serotype 2/9 AAVs, all plasmids were sequenced by the Children's Hospital of Philadelphia Nucleic Acid Core or University of Pennsylvania Sequencing Facility.

**Animals.** Balb/c (stock #000651), C57BL/6 J (called B6; stock #000664), B6.129 (Cg)-Gt(ROSA)26Sor^tm4(ACTB-tdTomato,-EGFP)Luo/J (called $R26^{mTmG/+}$; stock #007676), and B6.126S-Idua^tm1.1Kmke/J (called Idua-W392X, stock #017681) mice were purchased from The Jackson Laboratory (Bar Harbor, ME). Animals were housed in the Laboratory Animal Facility of the Colket Translational Research Building at The Children's Hospital of Philadelphia (CHOP). Standard conditions included a 12/12-h light/dark cycle, 20 °C ambient temperature, 50% humidity, and ad libitum access to food and water. The experimental protocols were approved by the Institutional Animal Care and Use Committee at CHOP and followed all relevant ethical regulations for animal testing and research and guidelines set forth in the National Institutes of Health's Guide for the Care and Use of Laboratory Animals.

**Genotyping.** Idua-W392X mice were genotyped to confirm homozygosity for the G→A (W392X) mutation in the Idua gene and homozygotes were subsequently

bred for colony maintenance/expansion and time dated experiments for in utero injections. About 2-mm tail snips were digested using Lucigen Quick Extract DNA Solution (Lucigen, Middleton, WI) as per kit instructions. Extracted DNA was amplified using primers Idua-F (5′-TGCTAGGTATGAGAGAGCCA-3′) and Idua-R (5′-AGTGTAGATGAGGACTGTGGT-3′) and the PCR product was analysed by Sanger sequencing to confirm homozygosity at the mutation site (Supplementary Table 3).

**In utero and postnatal mouse injections**. Intravenous in utero injections were performed in a similar fashion as previously published and as follows[34]. Specifically, fetuses of time-dated $R26^{mTmG/mTmG}$ × Balb/c (to generate $R26^{mTmG/+}$ fetuses) and Idua-W392X mice were injected on gestational day (E) 15.5. Under isoflurane anesthesia and after providing local anesthetic (0.25% bupivacaine subcutaneously), a midline laparotomy was performed, and the uterine horns were exposed. The vitelline vein—which runs along the uterine wall and enters the portal circulation resulting in first-pass effect to liver and systemic delivery via the ductus venosus—was identified under a dissecting microscope and 15 μL of phosphate buffered saline (PBS) or total virus (7.5 μL of each split-intein AAV vector) was injected per fetus using a 100-μm beveled glass micropipette. Based on the viral titers (Supplementary Table 2), this resulted in the injection of $6.5 \times 10^{10}$ total genome copies for Idua-W392X fetuses injected with AAV.ABE.Idua and $2 \times 10^9$ genome copies for $R26^{mTmG/+}$ fetuses injected with AAV.SpCas9.mTmG. C57BL/6 injections of AAV9 GFP and mCherry were conducted similarly with the delivery of 15 μL of total virus at a 1:1 ratio with a total injection of $2.5 \times 10^{10}$ total genome copies. A successful injection was confirmed by temporary clearance of blood from the vein and absence of injectate extravasation. The uterus was returned to the abdominal cavity and the laparotomy incision was closed in two layers with 4-0 Vicryl suture.

Viral injections into adult Idua-W392X mice were performed at 10 weeks of age via the retroorbital vein under isoflurane anesthesia. A total volume of 300 μL of virus (150 μL of each split-intein AAV vector) was injected such that ~$1.3 \times 10^{12}$ genome copies were injected per mouse. Based on the average weight of 10-week-old Idua-W392X mice (20 g) and E15.5 fetuses (1 g), both adult and fetal Idua-W392X mice received ~$6.5 \times 10^{10}$ total genome copies/gram.

**$R26^{mTmG/+}$ studies**. $R26^{mTmG/+}$ fetuses were injected via the vitelline vein at E15.5 with AAV.SpCas9.mTmG. Injected mice were sacrificed at DOL1 and 7 at which time organs, including the brain, heart, and liver were assessed for GFP expression indicative of on-target editing by flow cytometry and IHC. Uninjected age-matched $R26^{mTmG/+}$ served as controls.

**Idua-W392X studies**
*Prenatal experiments*. Idua-W392X homozygous fetuses were injected via the vitelline vein at E15.5 with AAV.ABE.Idua. In an initial screening experiment, two injected fetuses were sacrificed at 1 month of age at which time DNA from brain, heart, lung, liver, kidney, spleen, and gonads was assessed by Sanger sequencing and NGS for on-target editing to correct the G→A (W392X) mutation. A separate cohort of mice was set up for long-term genetic, metabolic, and phenotypic studies. Specifically, E15.5 Idua-W392X homozygous fetuses were injected at E15.5 with AAV.ABE.Idua or PBS. The experimental group was maintained until 6 months of age, the designated study endpoint. Uninjected age- and sex-matched Idua-W392X mice and wild-type B6 mice served as positive and negative controls, respectively. Urine was collected monthly starting at 1 month of age for GAG analysis. Serum was collected at 1 month of age for immunologic studies (see below) and at 6 months of age for IDUA activity analysis. Echocardiography was performed at 4 months of age and just prior to sacrifice (6 months of age). Additionally, at 6 months of age, a microCT (μCT) of the entire mouse body was performed and mice were subjected to a repetitive open field test and a grip strength test. Following sacrifice at 6 months of age, DNA from brain, heart, lung, liver, kidney, spleen, muscle, bone, and ovaries or sperm-containing epididymis was assessed by Sanger sequencing and NGS for off-target editing and on-target editing to correct the G→A (W392X) mutation. IHC was also performed. Finally, organs were assessed for GAG content and IDUA activity.

*Postnatal experiments*. For adult Idua-W392X studies, mice were injected via the retroorbital vein at 10 weeks of age with AAV.ABE.Idua. Urine was collected for GAG analysis prior to injection and at 4 and 5 months of age. Serum was collected for immunologic studies at 1-month post injection. Echocardiography was performed at 4 months of age as were 10 mm² liver biopsies from which genomic DNA was assessed by NGS for on-target editing to correct the G→A (W392X) mutation and from which liver IDUA activity and GAGs were evaluated. Prior to sacrifice at 6 months, echocardiography was repeated, after which heart and liver DNA were assessed for gene editing, IDUA enzyme activity, and GAG level.

**GFP/mCherry studies**. C57BL/6 fetuses were injected via the vitelline vein at E15.5 with AAV.GFP.mCherry. Seven days after injection, brains, hearts, and livers from pups and brains, hearts, livers, muscles, and uteri from dams were harvested. A portion of each organ was processed for flow cytometry. Double-stranded DNA was extracted from the indicated tissue using the Qiagen DNEasy Blood and Tissue

Kit according to the manufacturer's instructions (Qiagen, Hilden, Germany). Quantitative PCR of pup organs for the presence of AAV2 ITRs was conducted according to published protocols[72] using PowerUp SYBR Green reagent with a plasmid standard curve (Thermo Fisher Scientific, Waltham, MA). Supplementary Table 3 lists the PCR primers used for amplification.

**On-target and off-target sequence analysis**. On-target editing of the Idua gene was assessed by Sanger sequencing and NGS. Genomic DNA was extracted from the indicated tissue using the Qiagen DNEasy Blood and Tissue Kit according to the manufacturer's instructions (Qiagen, Hilden, Germany). Q5 polymerase was used to amplify the genomic region encompassing the W392X mutation with an annealing temperature of 66 °C. PCR products were assessed using a 1% agarose gel and then purified using the Qiagen PCR Purification Kit according to the manufacturer's recommendations. The top ten off-target sites were predicted using CRISPOR (http://crispor.tefor.net), ranked using the Cutting Frequency Determination (CFD) off-target score, amplified using Platinum SuperFi II Hi-Fidelity DNA Polymerase (Thermo Fisher Scientific, Waltham, MA) with a universal annealing temperature of 60 °C, and similarly purified and evaluated by NGS for the target genomic regions. Sanger and NGS were conducted by GeneWiz, Plainfield, NJ and data from NGS was analysed using CRISPResso2[73]. For NGS studies, at least 50,000 reads for each PCR amplicon at each target site for each sample was obtained. Supplementary Table 3 lists the PCR primers used for amplification and Supplementary Table 4 lists the predicted Idua off-target sites.

**Transthoracic echocardiography**. Mice were anaesthetized with 2% isoflurane and restrained on an imaging table with electrocardiogram (EKG) sensors. Transthoracic echocardiograms were performed using the Vevo 3100 Imaging System with a linear array MX550D transducer (FUJIFILM VisualSonics, Toronto, CA). An experienced echocardiographer acquired and analysed the images using Vevo LAB analysis software. The parasternal long-axis views and short-axis views were used to assess LV function and dimensions. A modified suprasternal view was used to measure the aortic arch and the aortic annulus. Measurements were performed to evaluate aortic dimensions, wall thickness, and LV dimensions during systole and diastole of the cardiac cycle. The Vevo LAB LV analysis tool, LV trace, was used to calculate cardiac output, fractional shortening, ejection fraction, stroke volume, heart rate, LV mass, dimensions, and volumes. Aortic annulus dimensions were taken during systole at maximal separation of the aortic cusps. Aortic arch measurements were taken in the ascending aorta during systole at the widest segment proximal to the innominate artery. Both imaging and analysis took place with the operator blinded to the treatment that each subject received.

**Micro computed tomographic scan (μCT)**. μCTs were conducted using a Siemens Inveon Multi-modality microPET/SPECT/CT platform (Siemens, Munich, Germany). Mice were anaesthetized with 2% isoflurane and placed in the μCT chamber. Slices (34 μm) were obtained with an integration time of 100 ms and reconstructed using Inveon Acquisition Workplace. DICOM images were analysed using Dragonfly v4.1 (Object Research Systems, Montreal, CA). Morphometric parameters were obtained as follows: snout angle (midsagittal angle between the nasal bone and the hard palate); skull width (maximum width at zygomatic arch); skull length (maximum caudal-rostral distance); zygomatic arch width (maximum thickness in the xy plane). Bone parameters endorsed by the American Society for Bone and Mineral Research were obtained using semiautomated bone analysis. Regions of interest were defined manually. Bone was segmented using the Otsu method. Cortical and trabecular bone were further segmented using the Buie algorithm at a trabecular thickness of 150 μm[74]. At sacrifice, handheld calipers were used to measure the midshaft femoral width and length.

**Open field test (OFT)**. After ensuring at least 72 h free of any research or anaesthetic exposure, OFTs were conducted in a neutrally lit quiet environment in a polycarbonate arena measuring 44 cm x 44 cm x 9 cm. After a 15-min acclimatization period, mice were placed into the center of the arena for 5 min at a time for a total of three trials. Video footage was acquired, cropped, and then converted to raw AVI format using FFMPEG V4.2. Videos were then analysed using the MouBeAT V1 plugin for NIH ImageJ V1.53c using a minimum detection threshold of 100 sq. pixels[75]. Color thresholding was set at 42. Rearing activity was scored manually by a blinded observer.

**Grip strength test**. After ensuring at least 72 h free of any research or anaesthetic exposure, grip strength was assessed using a digital force meter. Mice were removed from their cages and lifted by the tail to induce the forelimbs to grasp a 1.5 mm diameter metal bar attached to the grip strength meter. After engagement with the bar, the mouse was drawn away in the plane of the meter until its grip was broken. The maximum isometric muscular contraction was recorded. Three trials were obtained for each mouse and averaged.

**IDUA activity**. α-L-iduronidase tissue and serum assays were performed using a protocol described extensively by Ou et al.[76]. After organ harvest, 20 mg tissue samples were homogenized with 0.1% Triton X-100 lysis buffer using a TissueLyser

LT (Qiagen, Hilden, Germany). Of note, proximal femurs included acetabula were crushed using a mortar and pestle prior to homogenization. α-L-iduronidase activity was induced in specimens (25 µL of tissue homogenate or serum) using 4-methyl-umbelliferyl-α-L-iduronide (Glycosynth, Warrington, UK) in 0.4 M sodium formate buffer. After incubation for 30 min, reactions were arrested using glycine carbonate buffer and fluorescence measured at excitation 360 nm and emission 460 nm using a fluorescent plate reader (BioTek, Winooski, VT). A standard calibration curve was generated using 4-methylumbelliferone (Sigma-Aldrich, St. Louis, MO) with arrestant buffer at a detection sensitivity of 80. Enzyme activity was normalized to tissue lysate protein content using the Pierce BCA Protein Assay Kit according to the manufacturer's instructions (Thermo Fisher Scientific, Waltham, MA).

**Glycosaminoglycan (GAG) measurements**. After obtaining IDUA enzyme activity, 0.5 mL of papain solution was added to homogenized tissue lysates and incubated at 65 °C for 3 h and then centrifuged to clarify supernatant (Sigma-Aldrich). The Blyscan Glycosaminoglycan Assay (Biocolour, Carrickfergus, UK) was then used according to the manufacturer instructions to develop a calibration curve and measure sample absorbances. Tissue GAG content was normalized to tissue lysate protein content using the Pierce BCA Protein Assay Kit. Urine GAGs were quantified using the same kit and normalized to urine creatinine using the Mouse Creatinine Assay Kit according to the manufacturer instructions (Crystal Chem, Elk Grove Village, IL).

**SpCas9 antibody analysis**. Serum levels of anti-SpCas9 antibodies were assessed in Idua-W392X mice that were prenatal and postnatal recipients of AAV.ABE.Idua as well as uninjected 4-month-old *Idua*-W392X mice[37]. Serum was harvested 1-month post injection and antibody levels determined by ELISA. Ninety-six well Nunc MaxiSorp Plates (Thermo Fisher Scientific) were coated with SpCas9 protein (PNA Bio #CP01, Newbury Park, CA) at 0.5 µg/well in 1× coating buffer diluted from Coating Solution Concentrate Kit (SeraCare, Milford, MA) and placed at 4 °C overnight. Plates were washed with 1× Wash buffer and blocked with 1% BSA Blocking Solution (SeraCare) at room temperature for 1 h. Experimental and control sera were diluted 1000-fold with 1% BSA Diluent Solution (SeraCare) and added to wells for 1 h at room temperature with shaking. The mouse monoclonal anti-SpCas9 antibody (Clone 7A9, #A-9000-100, Epigentek, Farmingdale, NY) was serially diluted in 1% BSA Diluent Solution and used as a standard to quantify anti-SpCas9 IgG1 levels. After the 1-h incubation, wells were washed, and 100 µL of HRP-labeled mouse IgGκ binding protein (#SC-516102, Santa Cruz Biotechnology, Santa Cruz, CA) was added to each well for an additional 1 h at room temperature. Wells were subsequently washed four times and incubated with 100 µL of ABTS ELISA HRP Substrate (SeraCare). The SpectraMax M5 plate reader (Molecular Devices, San Jose, CA) with SoftMax Pro 6.3 software was used to measure Optical density at 410 nm.

**Histology**. For standard histology, tissues were fixed at 4 °C in 4% paraformaldehyde overnight and then manually dehydrated to 70% ethanol. Embedding, sectioning, and staining for Hematoxylin Eosin, Alcian Blue, Toluidine Blue, and Trichrome were conducted by IHC World (Ellicott City, Maryland). Slides and specimens were imaged using a Leica DMi8 (Leica Biosystems). Quantification of Alcian Blue was conducted using built-in ImageJ algorithms for stain-specific color deconvolution. Stain thresholding was set using the Otsu method and measured in each high-power field. Nuclei count was obtained using Renyi thresholding on binary hematoxylin channel images and then using ImageJ particle analysis with a nuclear size 100–900. Toluidine Blue was used specifically to assess vacuolization and cell size in the hippocampus which was graded (0-++++) in a blinded fashion[18]. Vacuolization scores were assessed as follows: 0, no cytoplasmic vacuoles; +, rare vacuolated cell (<1%); ++, cytoplasmic vacuoles in 0–10% of cells; +++, cytoplasmic vacuoles in 10–25% of cells; ++++, cytoplasmic vacuoles in >25% of cells. Cell size was assessed compared to C57BL/6 mice as follows: 0, no different; +, <25% larger; ++, 25–50% larger; +++, 50–75% larger; ++++, >75% larger.

For whole mount histologic analysis, antibody staining on thick tissue sections was performed[77]. Tissue was fixed in 2% paraformaldehyde overnight at 4 °C and washed the next day in cold PBS. The tissue was embedded in agarose, and precision cut tissue slices at 150 µm were obtained using a Compresstome vibrating microtome (Precisionary Instruments, Natick, MA). Tissue slices were prepared for whole mount IHC first by blocking overnight in a staining buffer that included 1% bovine serum albumin in PBS with 1% Triton X-100 (PBST). Incubation of primary antibodies including rabbit anti-IDUA-c-terminal (IDUA; 1:50), rat anti-leucine-rich repeat-containing G-protein receptor 5 (LGR5; 1:50) for liver tissue sections, and rabbit anti-cardiac troponin (CTNT; 1:50) for cardiac tissue sections was performed in staining buffer for 48 h. The sections were washed six times over 1 h in PBST. Sections were then incubated with secondary antibodies (donkey anti-rat or anti-rabbit Alexa Fluors) in staining buffer for 48 h. Sections were again washed for 6 h with the last wash including DAPI. The tissue section was cleared overnight in Scale SQ5 at 37 °C and then mounted in Scale S4 for imaging with a Leica Thunder Imaging system[78]. See Supplementary Table 5 for the list of antibodies used for histology.

**Cardiomyocyte isolation**. Cardiomyocytes, cardiac fibroblasts, and cardiac endothelial cells were isolated from in utero AAV.ABE.Idua injected *Idua*-W392X mice to assess for on-target gene editing in these cell populations. For cardiomyocyte isolation, freshly dissected hearts were transferred immediately to a petri dish containing ice cold calcium-free Hanks Balanced Salt Solution (HBSS). A 10 mm$^2$ section of LV was excised sharply and quartered. The Pierce Primary Cardiomyocyte Isolation Kit (Thermo Fisher Scientific) was used per the manufacturer's instructions except as follows. Enzyme supernatant and washings were reserved and combined to isolate non-cardiomyocyte cell fractions. After removal of supernatant, the remaining digestion products contained cardiomyocytes and were homogenized using 1 mL pipette tips cut to 3–5 mm diameters and then filtered through a 250 µm tissue strainer to avoid shear damage. Serial gravity filtration on ice was then employed for up to three cycles of 20 min to enrich cardiomyocytes. Cardiomyocyte enrichment was verified via light microscopy with confirmation of a majority of sarcomere containing cells and visible contraction. Genomic DNA from the enriched cardiomyocyte population was isolated using the Quick Extract DNA Solution as per the manufacturer's instructions. The supernatants containing the non-cardiomyocyte cell fractions were subjected to flow cytometry sorting to isolate cardiac fibroblasts and endothelial cells (see below).

**Flow cytometry**
*Sorting liver and cardiac cell populations*. Flow cytometry was used to sort liver and cardiac cell populations from 6-month-old AAV.ABE.Idua in utero injected *Idua*-W392X mice. For heart, after isolation of the enriched cardiomyocyte population (described above), the remaining supernatant was centrifuged at 300x*g* for 5 min. Pellets were resuspended in FACS staining buffer and then stained with anti-CD45-APC (1:160), anti-CD90.2-PE-Cyanine7 (1:333), and anti-CD31-Brilliant Violet 421(1:50). BD FACS Aria (BD Biosciences, Franklin Lakes, NJ) was utilized to sort fibroblasts (CD45$^-$ CD90.2$^+$), endothelial cells (CD45$^-$ CD31$^+$), and bone marrow-derived cells (CD45$^+$). For liver cells, freshly dissected livers were transferred immediately to a petri dish containing ice cold PBS, manually homogenized to create a single cell suspension, and resuspended in FACS staining buffer. Liver cells were subsequently stained with anti-CD45-APC and anti-LGR5-PE (1:50) and then sorted into CD45$^+$LGR5$^-$ (hematopoietic cells) and CD45$^-$LGR5$^+$ (liver progenitor cells). All sorted cell populations were then lysed using QuickExtract DNA Extraction Solution to isolate genomic DNA which was then analysed by NGS for on-target Idua editing (as described above). Please see Supplementary Table 5 for the list of antibodies used and Supplementary Fig. 6 for representative flow cytometry gating for sorting liver LGR5$^+$ cells and cardiac fibroblasts and endothelial cells.

*mTmG studies*. E15.5 $R26^{mTmG/+}$ fetuses were injected with AAV.SpCas9.mTmG, and hearts and livers were harvested at DOL1 and 1 week of age. Brains, hearts, and livers were processed as described above to obtain single cell suspensions. Cells were stained with DAPI to exclude dead cells and with anti-CD45-PerCP-Cyanine5.5 (1:50) to exclude lymphohematopoietic cells. The expression of endogenous GFP upon excision of the *mT* gene via targeting the *loxP* sites was assessed as an indication of successful on-target editing. Uninjected $R26^{mTmG/+}$ mice served as a negative control for gate placement. Please see Supplementary Table 5 for the list of antibodies used.

*GFP/mCherry studies*. E15.5 C57BL/6 fetuses were injected with AAV9.GFP.mCherry. Pup brains, hearts, and livers and dam brains, hearts, livers, muscles, and uteri were harvested at 7 days post injection. Uteri were digested in PBS solution containing Type II collagenase, dispase, and DNAse. Muscles were digested in PBS solution containing Type I collagenase, dispase, and DNAse. Hearts and livers were processed as described above to obtain single cell suspensions, muscle and uteri were triturated to a single cell suspension, and brains were homogenized to obtain a single cell suspension. Single cell suspensions were placed in FACS staining buffer. The presence of CD45-GFP+, CD45-mCherry+, and CD45-GFP + mCherry+ populations was assessed by flow cytometry. Uninjected fetuses and dams served as negative controls for gating.

**Statistics and reproducibility**. All quantitative data was analysed in JMP 14.3 (SAS Institute, Inc., Cary, NC). Outlier analysis and normality assumptions were tested using normal quantile plots and fitting a normal distribution to the data. The Brown–Forsythe test was used to assess unequal variance between comparison groups. Two-sided Student's *t*-test was used for direct comparisons between two groups. Groups with unequal variance were compared using the Wilcoxon test for multiple groups. Survival analysis was evaluated using Mantel–Cox log-rank comparison of survival curves. Regressions for comparisons between transduction and viral titers and 95% prediction confidence intervals were produced using a quadratic fit and appropriate transformations per residual plots. All comparison tests were conducted with α = 0.05. Representative immunohistochemical, histologic, µCT, photographic, and microscopic images were obtained for three biologic replicates per sample unless otherwise noted for quantification purposes. Graphical output was generated using GraphPad Prism version 8.0.0 (GraphPad Software, San Diego, CA).

**Reporting summary**. Further information on research design is available in the Nature Research Reporting Summary linked to this article.

## Data availability

DNA sequencing data has been deposited on the NCBI Sequence Read Archive; BioProject ID: PRJNA725910 (http://www.ncbi.nlm.nih.gov/bioproject/725910). Source data are provided with this paper. Additional data that support the findings of this study are available from the corresponding author upon reasonable request.

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

## Acknowledgements

This work was supported by grants DP2HL152427 and R01DK123049 (W.H.P.) from the United States National Institute of Health (NIH) and generous family gifts to The Children's Hospital of Philadelphia (CHOP) (W.H.P. and D.B.F.). We thank A. Weilerstein for his help with animal care, A. Radu for her assistance with histology, the Imaging facility at CHOP, C. Pascua for her echocardiography expertize, and Dr. S. Magnitsky for his assistance with CT scans. Figure 1b was created by the authors with BioRender.com.

## Author contributions

S.K.B., B.M.W., M.V.K., A.D., and H.A.H. performed experiments and acquired and analysed the data. S.K.B. and B.M.W. were the lead individuals on the MPS1 experiments. S.K.B., B.M.W., and H.A.H. were the lead individuals on the mTmG experiments. S.K.B., M.V.K., and H.A.H. performed the in utero injections. S.K.B. and B.M.W. performed the postnatal injections and led all data acquisition. H.L. provided technical help and performed ELISA and flow cytometry analyses. P.Z. provided technical help and designed plasmids. K.S. performed preliminary in vitro studies. S.K.B., B.M.W., M.V.K., F.R.D.B., and T.W. obtained and analysed CT scans and echocardiography. S.K.B., B.M.W., and S.T. obtained and analysed open field tests. S.K.B., B.M.W., F.R.D.B., P.C., S.J., and D.B.F. performed immunofluorescent and histologic analyses. V.S. and P.M. provided critical technical help. R.J. and K.M. provided expert technical guidance. W.H.P. designed and oversaw the experiments and wrote the paper with S.K.B. and B.M.W.

## Competing interests

The authors declare no competing interests.

## Additional information

**Peer Review information** *Nature Communications* thanks the anonymous reviewer(s) for their contribution to the peer review of this work. Peer reviewer reports are available.

