## [Peer Review File · Nature Communications]

Reviewer comments, first round

Reviewer #1 (Remarks to the Author):

In this novel and impactful study, Bose and White et al. demonstrate proof-of-concept feasibility of using split-AAVs to deliver adenine base editor to the developing murine fetus in a model of Mucopolycassharidosis Type I (MPS-IH, Hurler Syndrome). This technique is used to elicit high levels of base editing in the liver and lung, with some in utero treated animals displaying editing in the brain. Importantly, this work also illustrates durable biochemical and physiologic improvements in disease phenotype as evidenced by significant increases in serum, heart, and liver IDUA; a significant decrease in GAGs in the urine, heart, and liver; improved cardiac function; improved musculoskeletal strength; and improved survival; among other evidence, when compared to untreated controls. Additionally, the authors demonstrate a degree of genomic precision, with insignificant off-target mutagenesis in ten predicted sites. This work builds upon previous work of this group and others who have demonstrated the therapeutic potential of in utero gene editing in the treatment of genetic disorders affecting neonates and children and is likely to be of interest to Nature Communications' broad readership.

I have the following questions about the work presented:

1. In the introduction the authors state that children typically present with disease pathology at six months of age. Although in utero diagnosis may be technically possible (there are reports of measuring GAGs and culturing amniotic fluid cells to obtain a diagnosis), are there any instances of Hurler Syndrome being diagnosed in utero using noninvasive prenatal testing?
2. Additionally, in the introduction, the authors suggest that in utero base editing offers the possibility of treating MPS-IH prior to the onset of irreversible pathology. Is it known when the onset of irreversible pathology occurs in human fetuses and/or the Idua-W392X MPS-IH mouse model?
3. The authors demonstrate an improved survival of a cohort of in utero treated mice when compared to untreated MPS-IH mice (Figure 4t). From a procedural standpoint, what is the survival of fetuses to birth after vitelline vein injection of AAVs? And is there any impact on survival when compared to vitelline vein injection of a control such as saline?
4. For the R26mTmG/+ studies, GFP expression was evaluated on days of life one and seven. There appears to be an increase in GFP expression when evaluated at one week compared to one day of life; were any longer time points assessed? GFP expression in the brain or nervous system should also be evaluated.
5. When measuring the % base edited alleles in the gonads, what proportion of the mice were male or female? Were spermatocytes and oocytes isolated? Given the potential to access gametes in utero, it would be important to understand if this technique led to germline modification.
6. With any fetal intervention, one concern is the risk undertaken by the mother. Are AAVs or evidence of base editing (either GFP expression or sequence modification) detectable in maternal tissues?
7. A significant improvement in grip strength was shown at 6 months (Figure 5m). Did the authors look at editing in axial musculature and is there a correlation with increased grip strength when compared to untreated animals?

8. In figure 4, the authors demonstrate that significant improvement in cardiac function is largely observed at six months post-in utero treatment, when compared to the four-month time point. It is unclear why the authors then elected to compare the postnatally treated animals to both control groups and the in utero treated cohort at four months when the majority of the cardiac parameters assayed by echocardiography do not exhibit significant differences.

9. It appears that the x-axis of Figure 8 is mislabeled. The text states, "as in the previous study, anti-SpCas9 antibodies were noted in the serum of adult but not fetal AAV.ABE.Idua recipients 1 month following injection (Figure 8)." The current labeling of the figure, however, indicates that there is a significant increase in the concentration of antibodies in the prenatally treated animals.

10. The authors argue that an advantage of in utero base editing is the lack of immune response in utero following AAV delivery. It is appreciated that the possibility to administer a postnatal booster treatment could afford an appreciable therapeutic advantage. Do the authors have any evidence that the presence of antibodies negatively impacts either the percentage of corrected alleles or the difference in biochemical and ultrasonographical disease improvement? Or, perhaps, could the differences observed when editing pre- versus postnatally be attributed to another aspect of development, such as stem cell biology, or the increased length of time the postnatally edited mice were exposed to reduced IDUA activity as posited by the authors in the discussion?

11. The Sequencing Read Archive acquisition number should be included.

Reviewer #2 (Remarks to the Author):

This study employed AAV9 to deliver an intein split ABEmax and gRNA to edit the W392X (G->A) mutation in a mouse model of Mucopolysaccharidosis Type I (MPS-IH), a lethal lysosomal storage disease caused by mutations in IDUA. In utero AAV9 delivery resulted in editing of W392X mutation in both liver and heart cells with an efficiency of ~13%, which improved the pathology at 6 months of age. Retro-orbital injection into adult animals resulted in partial improvement of the MPS-IH phenotype. This study highlights the promise of in vivo base editing as a therapeutic approach for the rare genetic disease such as MPS-IH. While the study is interesting and has very thorough phenotypical characterization, there is limited conceptual and/or technical advancement.

1. Fig 1h-m, The authors should provide quantitative analysis of what percentage of brain, heart and liver cells were restored to express IDUA, with large images covering the entire cross-section area.

2. The overall editing efficiency, even in the animals treated in utero, was only about 13% in the liver and heart. The authors should determine if the transduction efficiency is about similar to editing efficiency observed. This could help to understand if the editing efficiency could be improved by simply increasing the dose of AAV9.

3. In the postnatally injected animals, the authors should provide data on the editing efficiency in the heart, with IHC staining to quantify the percentage of cells which show IDUA expression.

4. As stated in the discussion that in utero AAV9 injection needs to consider the risk to mother, did the authors examine whether the in utero injection transduced any tissues in the mother?

5. How many MPS-IH patients may be treated with adenine base editing?

Reviewer #3 (Remarks to the Author):

The authors describe "In utero split AAV9 adenine base editing corrects the multi-organ pathology

in a lethal lysosomal storage diseaseI.”

In utero base editing has the potential to correct disease-causing mutations before the onset of irreversible pathology. MPS-IH is a LSD affecting multiple organs, often leading to early postnatal cardiopulmonary demise. The authors assessed in utero AAV9 delivery of an adenine base editor (ABE) targeting the Idua W392X mutation in the MPS-IH mouse, corresponding to the common IDUA W402X mutation in MPS-IH patients. They show efficient long-term W392X correction in hepatocytes and cardiomyocytes and low-level editing in the brain. In utero editing was associated with improved survival and amelioration of metabolic, musculoskeletal, neurologic, and cardiac disease. This proof-of-concept study demonstrates the possibility of efficiently performing therapeutic base editing in multiple organs before birth via a clinically relevant delivery mechanism, highlighting the potential of this approach for MPS-IH and other genetic diseases. The manuscript is written concisely, but there are several critical comments.

1. Provide the vector maps in details since the proposed study suggests the complicated strategy.
2. In utero AAV gene therapy; it remains unclear how this strategy can apply to human patients since newborn screening for MPS I is now popular to identify the patients. How the authors identify the patients in fetus?
3. Postnatal experiments; why injected via the retroorbital vein at 10 weeks-of-age? Any reason to use the retroorbital vein? Why at 10 weeks old?
4. Glycosaminoglycan (GAG) measurements; Why did the authors use total GAG assay by the dye method? This is semi-quantitative. Use LC-MS/MS method to measure individual GAGs (HS, DS, and KS) in urine, plasma (or serum) and tissues.
5. Histology; it is a standard to use toluidine blue staining with 0.5 um to see vacuoles. Specify the extent of vacuoles or the size of the cells, especially in neurons and chondrocytes.
6. Specify the bone pathology in growth plate region as well as brain hippocampus area.
7. Why were the enzyme activities were increased only in liver and cardiac muscle? Any reason? Why enzyme activity in brain is trivial? How about the enzyme activity in bone?
8. References; cite the following articles
A Highly Efficacious PS Gene Editing System Corrects Metabolic and Neurological Complications of Mucopolysaccharidosis Type I.
Ou L, Przybilla MJ, Ahlat O, Kim S, Overn P, Jarnes J, O'Sullivan MG, Whitley CB, Ou L, et al. Mol Ther. 2020 Jun 3;28(6):1442-1454.
Human genome-edited hematopoietic stem cells phenotypically correct Mucopolysaccharidosis type I.
Gomez-Ospina N, Scharenberg SG, Mostrel N, Bak RO, Mantri S, Quadros RM, Gurumurthy CB, Lee C, Bao G, Suarez CJ, Khan S, Sawamoto K, Tomatsu S, Raj N, Attardi LD, Aurelian L, Porteus MH. Gomez-Ospina N, et al. Among authors: tomatsu s. Nat Commun. 2019 Sep 6;10(1):4045
9. Describe the successful gene therapy by lenti-virus vector on MPS I patients.

Overall, the manuscript should be revised according to the comments.

Reviewer #1:

In this novel and impactful study, Bose and White et al. demonstrate proof-of-concept feasibility of using split-AAVs to deliver adenine base editor to the developing murine fetus in a model of Mucopolysaccharidosis Type I (MPS-IH, Hurler Syndrome). This technique is used to elicit high levels of base editing in the liver and heart, with some in utero treated animals displaying editing in the brain. Importantly, this work also illustrates durable biochemical and physiologic improvements in disease phenotype as evidenced by significant increases in serum, heart, and liver IDUA; a significant decrease in GAGs in the urine, heart, and liver; improved cardiac function; improved musculoskeletal strength; and improved survival; among other evidence, when compared to untreated controls. Additionally, the authors demonstrate a degree of genomic precision, with insignificant off-target mutagenesis in ten predicted sites. This work builds upon previous work of this group and others who have demonstrated the therapeutic potential of in utero gene editing in the treatment of genetic disorders affecting neonates and children and is likely to be of interest to Nature Communications' broad readership.

Thank you for your comment highlighting the novelty and impactful nature of the study as well as its likely interest to Nature Communications' broad readership.

I have the following questions about the work presented:

1. In the introduction the authors state that children typically present with disease pathology at six months of age. Although *in utero* diagnosis may be technically possible (there are reports of measuring GAGs and culturing amniotic fluid cells to obtain a diagnosis), are there any instances of Hurler Syndrome being diagnosed *in utero* using noninvasive prenatal testing?

Thank you for this important question. The eventual clinical success of prenatal gene editing will be reliant on reliable and accurate prenatal testing.

As you astutely point out, amniotic fluid sampling for GAG and enzyme levels can serve as markers for mucopolysaccharidoses and lead to sequencing-based diagnosis. Current options for sampling include chorionic villus sampling and amniocentesis. For example, in one study involving amniotic fluid sampling between 16 and 22 weeks gestation in eight women with a family history of MPS-IH, amniotic fluid GAG and enzyme levels confirmed the diagnosis¹. In a separate study 14 pregnant women with already affected children underwent either amniocentesis

or chorionic villus sampling during a second pregnancy. Diagnosis was made based on 2-dimensional electrophoresis or enzyme assay. There were no false positives or negatives in this study². Finally, amniocentesis in five pregnant women in at-risk families led to prenatal diagnosis in six fetuses on the basis of direct sequencing of the IDUA gene³. These studies demonstrate that prenatal diagnosis of MPS-IH including genetic diagnosis of the affected gene, particularly in already affected families is currently feasible. Accordingly, we would imagine the initial clinical application of *in utero* gene editing to involve a fetus in which a previous family member had the disease. Although amniocentesis is an invasive diagnostic test, it is now routinely performed (greater than 200,000 amniocenteses are performed annually in the United States⁴) and carries with it a very low complication rate as indicated in a recent meta-analysis demonstrating the risk of pregnancy loss following amniocentesis and CVS at less than 24 weeks gestation to be 0.11% and 0.22% respectively⁵.

Regarding the current state of technology, two different carrier screens that include MPS-IH as part of their comprehensive standard panel are available to parents to establish overall fetal risk^{6,7}. In families with a positive carrier screen or known affected children, new technology allows for the potential of direct non-invasive diagnosis. The recently available commercial test Resura from Progenity isolates cell-free fetal DNA, uses digital droplet PCR amplification, and next generation sequencing (NGS) to calculate fetal status with 99.9% sensitivity and 99.9% specificity⁸. Although there are no documented cases of MPS-IH diagnoses with this technology, it is indicated for monogenic diseases for which there are known single nucleotide variants as in the human W402X MPS-IH mutation. We anticipate that this and other non-invasive prenatal diagnostic technology will be increasingly accessible in the timeline during which prenatal therapeutics will be available for clinical use.

We have amended the seventh paragraph of the Discussion to reflect the above as follows:

“In addition to demonstrating treatment efficacy, key elements for clinical translation of prenatal gene editing for MPS-IH include feasible prenatal diagnosis and ensuring maternal-fetal safety. In families with no known history of MPS-IH, commercial carrier screens can identify pregnancies at risk for MPS-IH (e.g. Progenity Preparent Carrier Test and Integrated Genetics 500 Plus Panel). Alternatively, in families at high-risk of MPS-IH or with known affected offspring, minimally invasive ultrasound guided amniocentesis and chorionic villus sampling (CVS) have been used to prenatally diagnose MPS-IH affected fetuses with high specificity and sensitivity based on elevated GAGs and reduced IDUA enzyme activity^{1,2}. Moreover, amniocentesis in five pregnant women led to prenatal diagnosis in six fetuses on the basis of direct sequencing of the IDUA gene⁹. These studies demonstrate that prenatal diagnosis of MPS-IH, including the genetic diagnosis of the causative mutation, is currently feasible. Although invasive, these procedures are routinely performed (greater than 200,000/year in the United States⁴) and are associated with very low maternal and fetal complication rates (amniocentesis, 0.11% fetal loss; CVS, 0.22% fetal loss⁵). Recently, advances in noninvasive prenatal testing technology support the possibility that MPS-IH could be diagnosed via a noninvasive test. The optimization of techniques for collecting cell-free fetal DNA from maternal serum has allowed targeted sequencing of the fetal genome for disease-causing alleles and highlights the possibility of

a noninvasive prenatal genetic test for MPS-IH⁸. This technique and other non-invasive prenatal genomic diagnostic technologies are likely to be increasingly accessible in the timeline during which prenatal therapies may be available for clinical use.”

2. Additionally, in the introduction, the authors suggest that *in utero* base editing offers the possibility of treating MPS-IH prior to the onset of irreversible pathology. Is it known when the onset of irreversible pathology occurs in human fetuses and/or the *IDUA-W392X* MPS-IH mouse model?

Thank you for this important clarification. The irreversibility of skeletal and cardiovascular components of the MPS-IH pathology is widely discussed and referenced in the literature¹⁰⁻¹². Furthermore, the prenatal onset of MPS-IH pathology has also been demonstrated in the literature. Specifically, studies of aborted human fetuses have demonstrated evidence of tissue pathology as early as 18 weeks gestation including vacuolization in the liver, kidney, heart, bone, and the nervous system and GAG deposition in the nervous system and bone^{13,14}. Furthermore, studies of the most severe cases suggest that “nearly all patients (98%) show signs of disease during the first 6 months of life” and that early deposition of GAGs leads to irreversible neurologic and bone disease¹⁵.

As prenatal pathology is only characterized on aborted MPS-IH fetuses in which the ability to treat pathology is not possible, it is challenging to ascertain the irreversibility of MPS-IH pathology. However, it is known that GAG deposition begins prior to birth and that delayed treatment after birth with either enzyme replacement therapy or hematopoietic stem cell transplants fails to reverse cardiac and skeletal pathologies associated with MPS-IH. This is nicely exemplified by case reports from sibling studies wherein late enzyme replacement demonstrated the inability to reverse preexisting pathology, particularly in the bone^{14,16}. Thus, it appears that the earlier treatment is initiated, potentially even before birth, the higher the likelihood of a beneficial therapeutic effect. This possibility is also supported by the recently FDA-approved clinical trial (NCT04532047) for the use of prenatal enzyme replacement therapy for MPS-IH and other lysosomal storage diseases to treat prenatally diagnosed fetuses¹⁷.

To address the Reviewer’s comment and better align our discussion with published reports, we have amended the manuscript and removed the word “irreversible” in our descriptions of MPS-IH pathology highlighting only that the pathology begins prior to birth and that the earlier treatment is initiated the better the outcome.

We amended the second and third paragraphs of the Introduction to reflect this logic:

“Although MPS-IH typically presents with symptoms by 6 months of age, it can be prenatally diagnosed via biochemical and genetic assays and associated pathology begins before birth¹⁸⁻²². On histopathologic examination, mid-gestation MPS-IH fetuses have demonstrated evidence of disease in multiple organs including the liver, heart, and brain^{15,19-21}. Studies of severe MPS-IH cases demonstrate tissue deposition of GAGs leading to neurologic and bone pathology as early as 18 weeks gestation^{14,23}. Finally,

prenatal cardiac dysfunction has led to myocardial hypertrophy and early postnatal death in MPS-IH²².

“Current postnatal treatments include costly, lifelong, immunogenic enzyme replacement therapy (ERT) and hematopoietic stem cell transplantation (HSCT), which is limited by donor availability, graft failure, graft-versus-host disease, and complications of myeloablation/immunosuppression²⁴. Both human and mouse studies demonstrated improved outcomes following early initiation of ERT or HSCT compared to late treatment^{14,16,17,25}. Importantly, neither treatment resolves preexisting musculoskeletal and cardiac pathologies^{16,24,26}, which significantly contribute to MPS-IH clinical grade²⁷. These findings suggest that there is both a benefit to early diagnosis and treatment in MPS-IH, potentially even before birth. Moreover, current therapies have limited ability to correct the global disease phenotype, especially with delayed initiation.”

3. The authors demonstrate an improved survival of a cohort of *in utero* treated mice when compared to untreated MPS-IH mice (Figure 4t). From a procedural standpoint, what is the survival of fetuses to birth after vitelline vein injection of AAVs? And is there any impact on survival when compared to vitelline vein injection of a control such as saline?

We appreciate the Reviewer’s point regarding the procedural and vector risk posed by our treatment approach.

We now present our data reflecting survival to birth of MPS-IH fetuses injected with AAV.ABE.Idua which was 60.9% (14/23 fetus injected survived to birth). To address the Reviewer’s second question, we performed additional experiments to evaluate survival following the vitelline vein injection of E15.5 *IDUA-W392X* fetuses with equivalent volumes of phosphate buffered saline (PBS). Survival of PBS-injected pups to birth was 60.0% (6/10 fetuses injected), not statistically significantly different compared to the AAV.ABE.Idua treatment group. Of note, this survival to birth is comparable to a previously published study from our group in which we document a survival-to-birth rate of ~40% following vitelline vein injection of 5×10^6 bone marrow cells in E14 C57BL/6 fetuses²⁸. The variability in survival among these studies is likely related to what is being injected, who is doing the injection, the strain of mouse injected, and the volume of the injectate. Thus, as the reviewer aptly suggests, it is critical to do an internal control for each study as we have now done with the PBS-injected cohort.

Additionally, it is important to note that that the technical limitations associated with a prenatal injection in the fetal mouse model would not be expected in large animal models or in clinical translation. Specifically, clinical translation would involve the intravascular delivery of gene editing technology via the umbilical vein. This would be a minimally invasive procedure performed under ultrasound guidance and local anesthesia without the requirement for a maternal laparotomy. The procedural risk for umbilical vein injections would be similar to that documented for umbilical vein transfusions which are routinely performed—often multiple times during pregnancy—for fetuses diagnosed with fetal anaemia. In a series of 937 fetal blood transfusions for fetal anaemia (2001-2015) the complication rate per procedure was 1.2% and the fetal loss rate

per procedure was 0.6%²⁹. Notably these procedures were performed in anaemic fetuses whereas the risk of complication may be even lower in non-anaemic fetuses. Finally, the recently approved clinical trial (NCT02986698), “In utero hematopoietic stem cell transplantation for alpha-thalassemia major” in which the transplant is performed via ultrasound guided umbilical vein injection supports this procedural approach for *in utero* delivery of gene therapy technology in the future once additional small and large animal studies are complete.

We have amended the final paragraph of the second section of the Results section as follows to include survival data as follows:

“Survival to birth after *in utero* injection of AAV.ABE.Idua (60.9%) was comparable to survival after *in utero* injection of phosphate-buffered saline (PBS) in *IDUA-W392X* fetuses (60.0%).”

We also have amended the Discussion as follows:

“For all *in utero* procedures, the safety of both the fetus and mother is of utmost importance for potential clinical translation. Major risks include procedural morbidity, vector toxicity, and unintended mutagenesis. A limitation of prenatal intervention in the mouse is procedural risk related to injection. We demonstrated no maternal loss and a ~60% survival-to-birth for *IDUA-W392X* fetuses injected with either PBS or AAV.ABE.Idua which is comparable to previously published survival rates following injection of bone marrow cells in the mouse model²⁸. The parity between the two groups in the current study suggests that fetal mortality is related to technical aspects of the procedure and not directly associated with AAV at the dose used. Similar technical challenges are not expected for clinical translation. Specifically, clinical translation would involve intravascular delivery of gene editing technology via the umbilical vein. This would be a minimally invasive procedure performed under ultrasound guidance and local anesthesia without a maternal laparotomy. The procedural risks for an umbilical vein injection would be similar to those documented for umbilical vein transfusions which are performed routinely, often multiple times during pregnancy, for select fetuses diagnosed with anaemia. In a series of 937 fetal blood transfusions for fetal anaemia, a 1.2% maternal/fetal complication rate (e.g. emergency caesarean section, fetal bradycardia, preterm prelabour rupture of membranes, infection) and a 0.6% fetal loss rate were noted²⁹. Importantly, these procedures were performed in anaemic fetuses and complications may be lower in nonanaemic fetuses.”

We have also amended the Methods section to reflect the new survival-to-birth study including the control as follows:

“The vitelline vein—which runs along the uterine wall and enters the portal circulation resulting in first-pass effect to liver and systemic delivery via the ductus venosus—was identified under a dissecting microscope and 15 μ L of phosphate buffered saline (PBS) or total virus (7.5 μ L of each split-intein AAV vector) was injected per fetus using a 100- μ m beveled glass micropipette.”

4. For the R26mTmG/+ studies, GFP expression was evaluated on days of life one and seven. There appears to be an increase in GFP expression when evaluated at one week compared to one day of life; were any longer time points assessed? GFP expression in the brain or nervous system should also be evaluated.

Thank you for this recommendation. We assessed GFP expression in the brain at day of life 1 and at 1 week and found very low levels of expression consistent with very low levels of editing in the brain in the R26^{mTmG/+} model. This is also consistent with the low levels of base editing in the brain that we demonstrate in the MPS-IH mouse model.

We have now included the flow cytometry data on the brain in an updated Figure S1 demonstrating these findings.

Updated Figure S1: *In utero* CRISPR-mediated nonhomologous end joining (NHEJ) following split-intein AAV9 delivery in the R26^{mTmG/+} mouse model. E15.5 R26^{mTmG/+} fetuses were injected with split-intein AAV9s containing the SpCas9 transgene and gRNA targeting the *loxP* sites flanking the *mT* cassette (AAV.SpCas9.mTmG). Successful excision of the mT cassette and repair

via NHEJ results in expression of green fluorescence. **(a-p)** The hearts and livers of prenatally injected mice and uninjected $R26^{mTmG/+}$ controls were analysed at 1 week-of-age by stereomicroscopy (experimental heart, a-d; control heart, e-h; experimental liver, i-l; control liver, m-p) for GFP expression. (a-p) Scale bar=1mm. **(q-s)** Brains, hearts, and livers of day-of-life 1 and 1 week old $R26^{mTmG/+}$ mice prenatally injected with AAV.SpCas9.mTmG were assessed by flow cytometry for GFP expression. **(t-aa)** Whole mount IHC of heart with staining for troponin (yellow, t-w) and the liver with staining for LGR5 (white, x-aa) in 1 week old $R26^{mTmG/+}$ mice prenatally injected with AAV.SpCas9.mTmG (u,w,y,aa) and uninjected $R26^{mTmG/+}$ mice (t,v,x,z). (t-w) Scale bar=50 μ m. Red-filled arrowheads identify cardiomyocytes. (x-aa) Scale bar=25 μ m. White arrowheads identify LGR5+ cells. EGFP, green; TdTomato, red; cTnT, cardiac troponin, yellow; LGR5, leucine-rich repeat-containing G-protein receptor 5, white.

We also noted these findings in the Results section:

“Wide-field microscopy and flow cytometry in the heart and liver on days of life 1 and 7 demonstrated GFP expression consistent with editing, whereas minimal GFP expression was noted in the brain (Figure S1a-s).”

We also updated the Methods section to reflect the addition of the brain analysis.

The $R26^{mTmG/+}$ studies were performed as short-term screening studies to demonstrate the ability, in our hands, to achieve CRISPR-mediated gene editing using a dual AAV system. Since the primary focus of our study was long-term correction of the *Idua* mutation and mouse MPS-IH phenotype using prenatal base editing we did not generate longer term data in the $R26^{mTmG/+}$ model.

5. When measuring the % base edited alleles in the gonads, what proportion of the mice were male or female? Were spermatocytes and oocytes isolated? Given the potential to access gametes *in utero*, it would be important to understand if this technique led to germline modification.

Thank you for this clarifying and important question. We agree that attention to germline modification is very important to consider for any type of editing including *in utero* editing. Our experimental group of prenatally injected mice harvested at 6 months of age were 30% male (3/10). Notably, control and postnatally injected groups were sex-matched to this proportion. We harvested sperm-containing epididymis for all three males for subsequent NGS. However, it was not possible to obtain oocytes without stimulation in these reproductively poor mice. As such, the whole ovary was digested for sequencing in all 7 females. Our results indicated editing efficiencies of 0-0.55% which was not different from that found in uninjected control gonads (0.16-0.34%). Given the sensitivity of this sequencing technique, we felt confident that germline mutations would be detected if present. Of note this is also consistent with our previously published study in which adenovirus was used to deliver the cytosine base editor, BE3, *in utero* in the mouse model. In this study, NGS of DNA isolated from gametes demonstrated editing efficiencies of 0.2-0.8% which was similar to negative controls (0.03-0.6%)³⁰. Furthermore, in unpublished work, we have mated

mice that were edited prenatally. NGS of genomic DNA from off-spring of these mice have not demonstrated any evidence of editing. Thus, in the mouse model in which either adenovirus or AAV is used to deliver a base editor, we have yet to demonstrate evidence of germline modification.

We have amended the fourth paragraph of the Results section as follows:

No editing was noted in genomic DNA from the ovaries (N=7 mice; NGS: 0-0.55%) or sperm-containing epididymis (N=3 mice; NGS: 0-0.39%) in experimental mice (control gonad; N=2; NGS: 0.16-0.34%) (Figure 1c).

We also amended the Methods paragraph describing DNA extraction as follows:

“Following sacrifice at 6 months of age, DNA from brain, heart, lung, liver, kidney, spleen, muscle, bone, and ovaries or sperm-containing epididymis was assessed by Sanger sequencing and NGS for off-target editing and on-target editing to correct the G→A (W392X) mutation.”

6. With any fetal intervention, one concern is the risk undertaken by the mother. Are AAVs or evidence of base editing (either GFP expression or sequence modification) detectable in maternal tissues?

As the Reviewer insightfully points out, risk to the mother is a critical consideration in any prenatal intervention. We thank them for the opportunity to highlight this in the manuscript.

To address this question, we performed two additional sets of experiments/analyses. We injected E15.5 C57BL/6 fetuses of 2 dams with AAV9 containing the GFP transgene and AAV9 containing the mCherry transgene (note injection of AAVs expressing two different fluorescent transgenes was done to address an additional Reviewer’s question below). At 7 days post injection, we confirmed GFP and mCherry expression in the fetal recipients and then assessed the brains, hearts, muscles, livers and uteri of the dams by flow cytometry for GFP and/or mCherry expression. This analysis did not demonstrate any GFP or mCherry expression in the maternal organs. Additionally, we assessed evidence of base editing in tissues from two MPS-IH mothers in whom fetuses underwent editing following *in utero* injection. In addition to evaluating genomic DNA from the uterus, we isolated DNA from the heart and liver, organs with strong AAV9 tropism and in which we saw evidence of editing in *in utero* injected AAV.ABE.Idua fetuses. NGS editing efficiencies in livers (0.07%, 0.21%), hearts (0.04%, 0.01%) and uteri (0.02%, 0.02%) from dams that underwent *in utero* surgery were no higher than those in uninjected control livers and hearts (0.19-0.49%). Given this finding, we believe *in utero* injection of AAV.ABE.Idua poses minimal editing risk to the mother in the mouse model. Of note this is also consistent with our previously published study in which adenovirus was used to deliver the cytosine base editor, BE3, *in utero* in the mouse

model. In this study, NGS of DNA isolated from maternal organs demonstrated no significant editing above background³⁰.

We have amended the Results section as follows:

“Finally, the brains, hearts, muscles, livers and uteri of 2 mothers of injected fetuses were evaluated by flow cytometry 7 days post injection and demonstrated no evidence of GFP or mCherry expression suggesting that no maternal transduction occurred.”

“Given the significant heart and liver editing, the importance of MPS-IH cardiac pathology, and the rationale that *in utero* editing can target progenitor cells, NGS editing efficiencies were evaluated in liver LGR5⁺ progenitor cells (~12.8%) and cardiac cell subpopulations including myocytes (~12.6%), endothelial cells (~3.0%), and fibroblasts (~2.3%) (Figure 1d-e). Finally, genomic DNA from the uterus, heart and liver of two dams in whom fetuses underwent editing following *in utero* AAV.ABE.Idua injection was assessed by NGS. Editing efficiencies in experimental dams (heart: 0.04%, 0.01%; liver: 0.07%, 0.21%; uteri: 0.02%, 0.02%) were comparable to uninjected control livers and hearts (0.19-0.49%) suggesting no maternal editing occurred.”

We have also amended the eighth paragraph of the Discussion as follows:

“For all *in utero* procedures, the safety of both the fetus and mother is of utmost importance for potential clinical translation. Major risks include procedural morbidity, vector toxicity, and unintended mutagenesis. A limitation of prenatal intervention in the mouse is procedural risk related to injection. We demonstrated no maternal loss and a ~60% survival-to-birth for *IDUA-W392X* fetuses injected with either PBS or AAV.ABE.Idua which is comparable to previously published survival rates following injection of bone marrow cells in the mouse model²⁸. The parity between the two groups in the current study suggests that fetal mortality is related to technical aspects of the procedure and not directly associated with AAV at the dose used. Similar technical challenges are not expected for clinical translation. Specifically, clinical translation would involve intravascular delivery of gene editing technology via either an umbilical vein or intracardiac injection. This would be a minimally invasive procedure performed under ultrasound guidance and local anesthesia without a maternal laparotomy. The procedural risks for an umbilical vein injection would be similar to that documented for umbilical vein transfusions which are performed routinely, often multiple times during pregnancy, for select fetuses diagnosed with anaemia. In a series of 937 fetal blood transfusions for fetal anaemia, a 1.2% maternal/fetal complication rate (e.g. emergency caesarean section, fetal bradycardia, preterm prelabour rupture of membranes, infection) and a 0.6% fetal loss rate were noted²⁹. Importantly, these procedures were performed in anaemic fetuses and complications may be lower in nonanaemic fetuses. In addition to procedure-related risks, maternal editing,

and off-target mutagenesis in the fetus are critical safety considerations. As such, we found no evidence of fluorescent expression in mothers of fetuses injected with AAV.GFP.mCherry and no evidence of base editing in the uteri, hearts, or livers of mothers of fetuses injected with AAV.ABE.Idua. These organs reflect both those in direct proximity to the fetal injection as well as those that align with AAV9 viral tropism. Finally, an assessment of off-target mutagenesis at the top 10 predicted sites in genomic DNA from the fetal liver revealed negligible unintended base edits and indels. These findings suggest that fetal injection of AAV.ABE.Idua portends limited procedural, genetic, and vector risks to the mother and fetuses in the mouse model.”

We have also modified the limitations paragraph of the Discussion as follows:

“In addition, the *in utero* mouse model does not fully recapitulate the corresponding techniques that would be employed in human prenatal interventions. This limits the extrapolation of model-specific safety data to clinical practice. As such, additional large animal studies that better approximate human anatomy and procedural technique are merited to assess maternal and fetal risk comprehensively.”

Finally, the Methods section was updated to reflect the new experiments.

“AAV2/9 CMV-GFP and AAV2/9 CMV-mCherry were purchased from Vector Biolabs (Malvern, PA).”

“C57BL/6 fetuses were injected via the vitelline vein at E15.5 with AAV.GFP.mCherry. Seven days after injection, brains, hearts, and livers from pups and brains, hearts, livers, muscles, and uteri from dams were harvested. A portion of each organ was processed for flow cytometry.”

“*GFP/mCherry studies:* E15.5 C57BL/6 fetuses were injected with AAV9.GFP.mCherry. Pup brains, hearts, and livers and dam brains, hearts, livers, muscles, and uteri were harvested at 7 days post injection. Uteri were digested in PBS solution containing Type II collagenase, dispase, and DNase. Muscles were digested in PBS solution containing Type I collagenase, dispase, and DNase. Hearts and livers were processed as described above to obtain single cell suspensions, muscle and uteri were triturated to a single cell suspension, and brains were homogenized to obtain a single cell suspension. Single cell suspensions were placed in FACS staining buffer. The presence of CD45-GFP+, CD45-mCherry+, and CD45-GFP+mCherry+ populations was assessed by flow cytometry. Uninjected fetuses and dams served as negative controls for gating.”

7. A significant improvement in grip strength was shown at 6 months (Figure 5m). Did the authors look at editing in axial musculature and is there a correlation with increased grip strength when compared to untreated animals?

To address this question, we have performed additional experiments to assess editing efficiencies and IDUA enzyme activity in biceps, a muscle involved in forelimb grip strength testing. In mice that underwent *in utero* injection of AAV.ABE.Idua, on-target base editing was ~1.65% and IDUA enzyme activity was ~3.2% of that seen in wild-type B6 control mice (~0.31 vs. ~9.4 ng/mg protein/hour). These samples were also assessed for GAG levels. GAG levels in mice that underwent *in utero* injection of AAV.ABE.Idua (~4.3 µg/mg protein) were significantly improved compared to W392X controls (~6.6 µg/mg protein; $p = 0.0016$) but not equivalent to the levels seen in B6 controls (~1.7 µg/mg protein; $p < 0.0001$).

Although the observed muscle IDUA activity is lower than that seen in the heart and liver, it is above the cited 1% of normal threshold to achieve phenotypic improvement^{31,32}. Finally, circulating IDUA as detected in the serum likely also provides benefit to the muscle. In sum, these findings suggest a correlation between bicep genomic editing, bicep biochemical expression, and phenotypic outcome as measured by grip strength.

We have amended the paragraph in the Results describing grip strength findings as follows:

“Finally, grip strength demonstrated significant improvement following *in utero* base editing (Figure 5m). Given this finding, bicep muscles were assessed for on-target editing efficiency in genomic DNA, IDUA enzyme activity, and GAG levels. NGS demonstrated an on-target correction rate of ~1.65% (Figure 1c) which was associated with detectable bicep IDUA enzyme activity (~3.2% of the normal IDUA activity in bicep muscles in B6 controls) compared to undetectable levels in W392X controls. This was consistent with a significant reduction in bicep GAG levels in prenatally AAV.ABE.Idua injected mice (~4.3 µg/mg protein) compared to W392X controls (6.6 µg/mg protein; $p = 0.0016$); however, the levels in treated mice were not equivalent to those in B6 controls (~1.7 µg/mg protein; $p < 0.0001$).”

We have also modified the manuscript to include data on bicep base editing (Figure 1c), GAG level (Figure 2c), and enzyme activity (Figure 2f) as follows:

Updated Figure 1C: NGS in organs from mice prenatally injected with AAV.ABE.Idua at 6 months

Updated Figure 2C: Tissue GAG levels in organs from mice prenatally injected with AAV.ABE.Idua at 6 months

Updated Figure 2F: Tissue IDUA levels in organs from mice prenatally injected with AAV.ABE.Idua at 6 months

Finally, the figure legend for Figure 2 was updated as follows:

Figure 2 Durable improvement in biochemical parameters in *IDUA-W392X* mice following *in utero* base editing. **(a)** Urine GAGs were measured monthly in B6 (N=14), *IDUA-W392X* mice prenatally injected with AAV.ABE.Idua (N=10 except for month 1, N=7), and uninjected *IDUA-W392X* mice (N=14). **(b-c)** Tissue GAGs were measured at 6 months of age in the heart and liver (b) and other indicated organs (c) in B6 (N=14, except for muscle/femur N=10), prenatally injected mice (N=10), and uninjected mice (N=14, except eye, N=13; and muscle/femur N=10). **(d)** Serum IDUA activity was measured at 6 months of age in B6 (N=10), prenatally injected mice (N=10), and uninjected mice (N=10). **(e-f)** Tissue IDUA activity was measured at 6 months of age in the heart and liver (e) and other indicated organs (f) in B6 (N=14, except for muscle/femur N=10), prenatally injected mice (N=10), and uninjected mice (N=14, except for muscle/femur N=10). ^, $p < 0.0001$; #, $p < 0.001$; *, $p < 0.05$. Wilcoxon test for multiple comparisons used to assess urine GAG months 1-3, liver GAG, 6-month serum IDUA activity, and heart and liver IDUA activity. All remaining statistical analyses used Student's t-test with $\alpha = 0.05$. GAG, glycosaminoglycans; IDUA, α -L-iduronidase.

8. In figure 4, the authors demonstrate that significant improvement in cardiac function is largely observed at six months post-*in utero* treatment, when compared to the four-month time point. It is unclear why the authors then elected to compare the postnatally treated animals to both control groups and the *in utero* treated cohort at four months when the majority of the cardiac parameters assayed by echocardiography do not exhibit significant differences.

We appreciate this careful review of the presented data.

As we observed that untreated control animals with the most severe cardiac phenotype died between 4 and 6 months, we opted to conduct 4-month echocardiography in postnatally treated mice to ensure adequate power for comparison. At the time of submission, we did not have 6-month data, but we are now pleased to present it.

As presented in the manuscript, although editing improved some cardiac outcome variables at 4 months, some functional characteristics such as ejection fraction, fractional shortening, and LV systolic diameter were similar between postnatally treated mice and *IDUA-W392X* controls. We now find postnatally treated mice did not have statistically significant worsening in these variables between 4 and 6 months. In addition, at 6 months, these variables are not statistically different than either uninjected *IDUA-W392X* controls or mice prenatally injected with AAV.ABE.Idua (the parameters are between the two means). This non-significant trend towards normalization without statistical significance compared to disease controls may be explained by the death of the sickest controls prior to 6 months which diminishes the detected effect size thus reducing the power of comparison. Nonetheless, we believe that the attenuation of the phenotype by postnatal editing is encouraging and represents a potential benefit of intervention.

To address these findings, we have now amended the text in the paragraph presenting postnatal data to reflect these 6-month findings as follows. As the findings are not statistically significantly different between 4 and 6 months, we have elected to leave the associated figure unchanged to maximize the presented number of controls. If the Reviewer or Editor feels it important to include the 6-month postnatally injected mice in the panels, we are happy to do so.

“Echocardiography at 4 months of age in *IDUA-W392X* mice postnatally injected with AAV.ABE.Idua revealed reduced ascending aorta and aortic valve diameters but similar left ventricle size, ejection fraction, and fractional shortening compared to untreated *IDUA-W392X* mice (Figure 7e-i). Between 4 and 6 months of age, ascending aorta diameter (4-to-6 month differences of means $\Delta\bar{x} = 0.47$, $p = 0.23$), aortic valve diameter ($\Delta\bar{x} = -0.03$, $p = 0.65$), left ventricle size ($\Delta\bar{x} = 0.14$, $p = 0.63$), ejection fraction ($\Delta\bar{x} = -2.90$, $p = 0.63$), and fractional shortening ($\Delta\bar{x} = -1.76$, $p = 0.62$) did not worsen and at 6 months of age were not significantly different than either prenatally treated mice or untreated *IDUA-W392X* controls. Notably, the death of the most diseased *IDUA-W392X* controls between 4 and 6 months diminishes the power of comparison at 6 months. Consequently, the relative improvement of postnatally treated mice is likely statistically underestimated at 6 months. In sum, these findings suggest an attenuation of disease progression in mice that underwent postnatal injection of AAV.ABE.Idua.”

We also amended the text in sixth paragraph of the Discussion as follows:

“Although ontological advantages for prenatal gene editing may exist, some patients may not be diagnosed before birth and the clinical safety of base editing technology initially needs to be demonstrated in a postnatal recipient. As such, we evaluated the efficiency of base editing and a limited phenotype assessment in postnatal recipients in the context of our prenatal experiments. Postnatally treated mice demonstrated efficient on-target editing in the heart and liver which was associated with improvement in cardiac parameters. Notably, although disease controls demonstrated rapid cardiac decline between 4 and 6 months even resulting in mortality, postnatally treated mice demonstrated cardiac disease at 4 months with attenuated progression between 4 and 6 months of age and no death. This difference at 4 months of age between prenatally and postnatally treated mice is likely due to prolonged IDUA exposure from the time of birth in those undergoing *in utero* base editing. This suggests that postnatal treatment can have encouraging phenotypic benefits; though, maximal effect may be realized with earlier therapy.”

We have also updated the Methods to reflect the addition of six-month echocardiography.

“Prior to sacrifice at six months, echocardiography was repeated, after which heart and liver DNA were assessed for gene editing, IDUA enzyme activity, and GAG level.”

9. It appears that the x-axis of Figure 8 is mislabeled. The text states, “as in the previous study, anti-SpCas9 antibodies were noted in the serum of adult but not fetal AAV.ABE.Idua recipients 1 month following injection (Figure 8).” The current labeling of the figure, however, indicates that there is a significant increase in the concentration of antibodies in the prenatally treated animals.

Thank you for pointing out this error.

The figure has been appropriately corrected to match the text (Updated Figure 8).

Updated Figure 8: Serum SpCas9 antibodies in W392X controls, mice prenatally injected with AAV.ABE.W392, and mice postnatally injected with AAV.ABE.W392X

10. The authors argue that an advantage of *in utero* base editing is the lack of immune response *in utero* following AAV delivery. It is appreciated that the possibility to administer a postnatal booster treatment could afford an appreciable therapeutic advantage. Do the authors have any evidence that the presence of antibodies negatively impacts either the percentage of corrected alleles or the difference in biochemical and ultrasonographical disease improvement? Or, perhaps, could the differences observed when editing pre- versus postnatally be attributed to another aspect of development, such as stem cell biology, or the increased length of time the postnatally edited mice were exposed to reduced IUDA activity as posited by the authors in the discussion?

This is an important point and we thank the Reviewer for raising these questions. Although we found that postnatally edited mice possessed anti-Cas9 antibodies, we do not have evidence that the presence of antibodies negatively impacts the percentage of corrected alleles or biochemical and ultrasonographical disease improvement. However, literature precedent demonstrates that the presence of anti-Cas9 antibodies can induce a cytotoxic CD8+ T cell response characterized by loss of recombinant AAV genomes and elimination of gene-edited cells³³. Importantly, as the Reviewer points out, the differences in editing observed between pre- and postnatally treated mice is likely multifactorial including but not limited to immune response, stem cell proliferation, and timing of treatment. Although we plan to pursue studies to further elucidate some of these mechanisms, we respectfully feel a robust characterization is outside the scope of this manuscript.

We have amended the fifth paragraph of the Discussion to acknowledge the Reviewer’s comments as follows:

“Importantly our study does not address the direct causality between the presence of SpCas9 antibodies and the differential editing levels seen in postnatally versus prenatally

treated mice. This difference is likely multifactorial due to, but not limited to, immune response, stem cell biology, and timing of treatment. Nonetheless, the potential benefit of a naïve fetal immune system may still be an important consideration for diseases in which a high level of correction and therefore additional booster treatments are necessary to rescue a phenotype.”

11. The Sequencing Read Archive acquisition number should be included.

We will include the Sequencing Read Archive number at time of publication.

Reviewer #2:

This study employed AAV9 to deliver an intein split ABEmax and gRNA to edit the W392X (G>A) mutation in a mouse model of Mucopolysaccharidosis Type I (MPS-IH), a lethal lysosomal storage disease caused by mutations in IDUA. *In utero* AAV9 delivery resulted in editing of W392X mutation in both liver and heart cells with an efficiency of ~13%, which improved the pathology at 6 months of age. Retro-orbital injection into adult animals resulted in partial improvement of the MPS-IH phenotype. This study highlights the promise of *in vivo* base editing as a therapeutic approach for the rare genetic disease such as MPS-IH. While the study is interesting and has very thorough phenotypical characterization, there is limited conceptual and/or technical advancement.

We thank the Reviewer for these comments. We believe demonstrating *in utero* editing of multiple target organs using a clinically relevant delivery approach (AAV) in a disease that could benefit from prenatal therapy represents a critical step towards future clinical translation.

1. Fig 1h-m, The authors should provide quantitative analysis of what percentage of brain, heart and liver cells were restored to express IDUA, with large images covering the entire cross-section area.

Thank you for this suggestion. The literature regarding the use of immunohistochemistry in the W392X mouse model (as opposed to the knockout model) of MPS-I is very limited. We used an antibody targeting the C-terminus of IDUA in order to avoid background contamination from the truncated version of the protein that is present in this disease model in contrast to the knockout model. We included these images to qualitatively demonstrate the presence of IDUA to further support our findings of enzyme activity. Given the multiple other objective and quantitative molecular and phenotypic outcome measures evaluated in this study, we did not pursue wide-field quantitative analysis at the time.

In an effort to address the Reviewer’s suggestions, we reexamined the previously stained slides (now over 6 months old) which had been preserved at -20°C in a light-protected environment. However, these slides have undergone significant degradation and are not quantifiable at this point (Reviewer Figure 1). Upon discovering this, we re-sectioned and re-stained the original paraffin-

embedded tissues. Unfortunately, the specific C-terminus IDUA antibody that was used for staining is no longer available from the manufacturer. In communication with the manufacturer there is no plan to reintroduce the product. We were able to identify an alternate available C-terminus antibody (Aviva Systems Biology ARP63635_P050) which unfortunately was derived from a different source than the original antibody used in our manuscript. On attempted staining across multiple preparation conditions including immunoperoxidase, we were unable to detect IDUA even in C57BL/6 control mice. As such, these figures are not currently reproducible. If the Reviewer and/or Editor feels it appropriate, we are happy to redact them from the manuscript.

Reviewer Figure 1. IHC staining of C-terminus IDUA in a B6 liver. Note degradation of both DAPI and IDUA staining.

2. The overall editing efficiency, even in the animals treated *in utero*, was only about 13% in the liver and heart. The authors should determine if the transduction efficiency is about similar to editing efficiency observed. This could help to understand if the editing efficiency could be improved by simply increasing the dose of AAV9.

Thank you for raising this question. Indeed, understanding the transduction efficiency in the prenatal setting is critical to assessing the dose-delivery relationship. To address the Reviewer's comment, we performed additional studies to assess the relationship between transduction efficiency and editing efficiency.

We first conducted a fluorescent titering experiment to assess the relationship between AAV9 delivery and observed transduction efficiency. To do so, we injected E15.5 time-dated C57BL/6 fetuses with 2.5×10^{10} total genome copies of AAV2/9 carrying CMV-GFP and CMV-mCherry (AAV.GFP.mCherry) in a 1:1 ratio to approximate the dosing of dual AAV delivery of AAV.ABE.Idua. Mice were harvested at 7 days post-injection to achieve maximal detection of fluorescent expression which approximates transduction. Livers, brains, and hearts were assessed by flow cytometry to determine CD45-GFP+, CD45-mCherry+, and CD45-GFP+mCherry+ populations in each specimen. Total transduction percentage is the sum of the three populations while dual transduction percentage refers only to the double positive cells (CD45-GFP+mCherry+). In parallel, quantitative PCR was used to amplify AAV-2 inverted terminal repeats (ITRs) to quantify the presence of the viral genome in DNA extracted from the brains,

hearts, and livers (n = 25) of the *in utero* AAV.GFP.mCherry injected mice. In so doing, we derived a regression model describing the relationship between the AAV9 titre and transduction efficiency (**New Figure S2**). Although the titres were found to appropriately vary in different organs, all data was fit collectively to generate a statistical relationship that would address a range of tissue viral titres.

New Figure S2. (a) Correlation between total GFP and/or mCherry or dual transduction versus vector titres in heart, liver, and brain at 7 days post injection. (b) Correlation between dual GFP/mCherry transduction versus vector titres in heart, liver, and brain at 7 days post injection. Dotted lines: 95% prediction confidence intervals.

The figure legend was updated to reflect this new figure.

“The relationship between tissue viral titres and expression of fluorescent reporters following *in utero* dual AAV9 delivery in C57BL/6 mice. E15.5 fetuses were injected with a 1:1 ratio of AAV9 CMV-GFP and AAV9 CMV-mCherry. Brain, heart, and liver were harvested at 7 days post injection and evaluated using flow cytometry for total fluorescent expression (CD45-GFP+, CD45-mCherry+, and/or CD45-GFP+mCherry+) or dual fluorescent expression (CD45-GFP+mCherry+). Tissue AAV2 inverted terminal repeats indicating the presence of viral DNA were assessed by quantitative PCR. (a, b) Quadratic predictive equations describing the relationship between transduction and viral titres were generated for total fluorescent transduction (a) and dual AAV transduction (b). Y axes depict transduction percent as determined via flow cytometry and X axes depict $\log_{10}(\text{vector genome copies per } \mu\text{g double-stranded DNA})$. Dotted lines represent 95% prediction confidence intervals. The statistical significance of regression parameters was assessed at the $\alpha=0.05$ level.

We then assessed the relationship between predicted transduction efficiency in AAV.ABE.Idua samples and editing efficiency. To do so, we conducted quantitative PCR to amplify AAV-2 ITRs in livers and hearts derived from *in utero* AAV.ABE.Idua injected mice at 30 days. Using these values, we applied the relationship assessed in the prior fluorescent titering experiment to predict the transduction efficiency in AAV.ABE.Idua samples. The average editing efficiencies and predicted transduction efficiencies at 30 days are demonstrated in Reviewer Table 1.

	% Edit	Predicted total transduction %	95% CI lower	95% CI upper	Predicted dual transduction %	95% CI lower	95% CI upper
Heart	13.9	3.9	2.1	6.1	0.7	0.3	1.2
Liver	26.8	2.3	1.4	3.4	0.4	0.2	0.7

Reviewer Table 1. Actual average 30 day on-target editing efficiency in AAV.ABE.Idua samples (n = 2) compared to predicted total and dual transduction efficiencies.

Given the delivery of similar titres of AAV.GFP.mCherry and AAV.ABE.Idua, we believe that the predicted transduction efficiency in prenatal AAV.ABE.Idua injected mice is much lower than the observed editing efficiency. To further substantiate this effect, we used the same methodology to evaluate the predicted dual transduction level in livers from postnatally injected mice 1 month after injection and found it to be 0.9% (95% CI 0.32, 1.87).

One explanation for the discrepancy between editing and predicted transduction efficiency is a potential competitive advantage for gene edited cells and the consequent clonal expansion of cells harboring corrected alleles. Nonetheless, these observations suggest that increasing the transduction efficiency may be a mechanism to increase editing efficiency and one way to improve transduction may be by increasing the viral dose.

We have modified the third paragraph of the Results as follows:

“We next sought to understand the relationship between dual AAV transduction and editing levels. AAV9s containing GFP or mCherry transgenes (AAV.GFP.mCherry) were delivered to C57BL/6 (B6) mice at a 1:1 ratio via the vitelline vein at E15.5. Flow cytometry at 7 days post injection demonstrated GFP+mCherry+ populations in the brain, heart, and liver which were compared to the concentration of AAV genomes in the same tissues yielding a significant relationship between dual viral delivery and fluorescent expression ($R^2 = 0.18$, $p = 0.034$) (Figure S2). This predictive regression model was then used to estimate the transduction efficiency of AAV.ABE.Idua based on AAV genomes and correlate transduction with editing efficiency at thirty days. The predicted dual transduction efficiency was 0.7% (95% CI 0.3, 1.2) in the heart and 0.4% (95% CI 0.2, 0.7) in the liver, suggesting substantially higher editing levels than viral transgene expression. Finally, the brains, hearts, muscles, livers and uteri of 2 mothers of injected fetuses were evaluated by flow cytometry 7 days post injection and demonstrated no evidence of GFP or mCherry expression suggesting that no maternal transduction occurred.”

We have also modified the Results paragraph on postnatal treatment as follows:

“At 4 months of age, NGS demonstrated editing of ~10.8% in liver genomic DNA (Figure 7a) which was associated with a predicted dual AAV transduction efficiency of 0.9% (95% CI 0.32, 1.87). In addition, liver IDUA and GAG levels were significantly improved compared to untreated *IDUA-W392X* mice (Figure 7c, d) and by 5 months of age, urine GAGs were reduced compared to untreated *IDUA-W392X* mice (Figure 7b).”

We also modified the third paragraph of the Discussion as follows:

“In this study, we predicted low but comparable levels of dual AAV transduction in both pre- and postnatally injected mice. Notably, our approximation of transduction is based on the delivery of fluorescent reporters in which kinetics may be different than that of AAV.ABE.Idua. In addition, despite similar viral dose and predicted transduction levels, we found that the level of liver editing in *in utero* injected mice was substantially higher than that in postnatally injected mice 1-month post injection. Given the difference in editing between pre- and postnatally injected mice not explained by viral delivery or predicted transduction, there is likely a potentiating mechanism for editing. Possible explanations include a competitive advantage for prenatally gene-edited cells, the accessibility of progenitor cells which are increasingly quiescent with age, and a higher rate of cellular replication and thus expansion of edited cells earlier in life. Collectively, these observations suggest that strategies to enhance transduction such as increasing the dose of AAV9 may improve editing efficiency. This, however, must be considered in the context of toxicity associated with high dose AAV wherein systemic delivery of greater than 1E14 vg/kg may result in dose limiting toxicities in humans^{34,35}. Importantly, the increased editing levels in prenatally treated mice were achieved using ~8E13 vg/kg and only 5% of the volume of viral vector delivered in adults. Given the engineering and financial challenges related to gene therapy dosing, *in utero* gene editing may thus be an effective means to maximize dosing value.”

We also modified the Methods to reflect the analyses involving AAV9.GFP.mCherry as follows:

“C57BL/6 injections of AAV9 GFP and mCherry were conducted similarly with the delivery of 15µL of total virus at a 1:1 ratio with a total injection of 2.5×10^{10} total genome copies.”

In utero and postnatal injections:

“GFP/mCherry studies: C57BL/6 fetuses were injected via the vitelline vein at E15.5 with AAV.GFP.mCherry. Seven days after injection, brains, hearts, and livers from pups and brains, hearts, livers, muscles, and uteri from dams were harvested. A portion of each organ was processed for flow cytometry. Double-stranded DNA was extracted from the indicated tissue using the Qiagen DNEasy Blood and Tissue Kit according to the manufacturer’s instructions (Qiagen, Hilden, Germany). Quantitative PCR of pup organs for the presence of AAV2 ITRs was conducted according to published protocols³⁶ using PowerUp SYBR Green reagent with a plasmid standard curve (Thermo Fisher Scientific, Waltham, MA). Supplementary Table 2 lists the PCR primers used for amplification.”

Flow cytometry:

“GFP/mCherry studies: E15.5 C57BL/6 fetuses were injected with AAV9.GFP.mCherry. Pup brains, hearts, and livers and dam brains, hearts, livers, muscles, and uteri were harvested at 7 days post injection. Uteri were digested in PBS solution containing Type II collagenase, dispase, and DNase. Muscles were digested in PBS solution containing Type I collagenase, dispase, and DNase. Hearts and livers were processed as described above to obtain single cell suspensions, muscle and uteri were triturated to a single cell

suspension, and brains were homogenized to obtain a single cell suspension. Single cell suspensions were placed in FACS staining buffer. The presence of CD45-GFP+, CD45-mCherry+, and CD45-GFP+mCherry+ populations was assessed by flow cytometry. Uninjected fetuses and dams served as negative controls for gating.”

Statistics:

“Regressions for comparisons between transduction and viral titres and 95% prediction confidence intervals were produced using a quadratic fit and appropriate transformations per residual plots.”

We also modified Supplementary Tables 1 and 2 to reflect the titres of the AAV9.GFP and AAV9.mCherry and appropriate primers.

Dual AAV 2/9	Titre (GC/mL)
AAV.ABE.Idua C-terminus	2.8x10 ¹²
AAV.ABE.Idua N-terminus	5.8x10 ¹²
AAV.mTmG C-terminus	1.5x10 ¹¹
AAV.mTmG N-terminus	1.2x10 ¹¹
AAV.GFP	1.0x10 ¹³
AAV.mCherry	1.0x10 ¹³

Table S1 Viral vector titres

Target	Forward Primer	Reverse Primer
On-target		
Idua	TGCTAGGTATGAGAGAGCCA	AGTGTAGATGAGGACTGTGGT
Off-target		
Intron:1700010I14Rik	GGGATTGCTCTGCTCTGTCT	TGTGTAAGAGTGGGCCATGT
Intron:Wnt11	CAGGCTTGAACACACACACA	AAAATCCCGTTGAGACCCCA
Intergenic:PapI-Fbxo27	CAACATTTGGAAGTCTGAGGC	TGCTGGGGTTACAAGGGTG
Intergenic:Fgf9-Gm25614	ACTGCAGGAATGGAAACTCC	CTCTAGAGACCCTGTGCTGG
Intergenic:Gm12106-Stc2	AGGCCTTCGATCAGACATCA	CAACAACATGGCTGCTCAGG
Intergenic:Gm26190-B3gat1	CCTTCACTCTCTTGGGCCTT	CAGTGTGAGCAAAGGGAAGC
Intergenic:Ccdc85c-Hhip1	ACAAGGAGGGGTGTGTGTAC	CTGCTGAGAGGTCTGGAG
Intron:Osbpl1a	GCCCACTTAATAACCCTGTGT	GCAGGAGGGGTCATTGATCT
Intron:Blnk	ACAGCACTGAGAAGGGACAA	CGGGAGGGATCGTAAAGTGA
Intron:Rhoj	TTGGCTAGTCTCCGTGTGAA	GGGGTCTAGAGGTCTTTGGG
qPCR		
AAV2 ITR	GGAACCCCTAGTGATGGAGTT	CGGCCTCAGTGAGCGA

Table S2 Primers used for Sanger sequencing, NGS in on- and off-target analysis, and quantitative PCR.

3. In the postnatally injected animals, the authors should provide data on the editing efficiency in the heart, with IHC staining to quantify the percentage of cells which show IDUA expression.

Thank you for this comment. Given the focus of the manuscript on the prenatal aspects of editing, we had not sacrificed the postnatally edited mice at the time of initial submission and thus did not have the requested heart data.

To address the Reviewer's question, we sacrificed the postnatally edited mice at 6 months of age and performed a limited phenotypic assessment in the context of our prenatal studies to determine cardiac editing, IDUA enzyme activity and GAG levels. Specifically, at six months of age, NGS of genomic DNA demonstrated editing in the liver (mean % \pm SD; 10.5 ± 5.2), the heart ($5.0\% \pm 4.0$), and in cardiac cell subpopulations including myocytes ($4.0\% \pm 3.0$), endothelial cells ($2.1\% \pm 2.8$), and fibroblasts ($0.54\% \pm 1.1$). Heart samples were also assessed for IDUA enzyme activity (3.9 ± 3.5 ng/mg/hr) and GAG levels (3.3 ± 1.2 μ g/mg protein).

As discussed above, we are unable to provide IHC staining to quantify the percentage of cells which show IDUA expression but believe that the cardiac IDUA activity and GAG levels combined with the phenotype assessment by echocardiography provide a detailed assessment of the effects of postnatal editing in *IDUA-W392X* mice.

To address the Reviewer's request, we have amended the section of the Results focusing on postnatal editing to include editing efficiencies in 6-month postnatally injected mice:

“At six months of age, NGS of genomic DNA demonstrated editing in the liver (mean % \pm SD; $10.5\% \pm 5.2$), the heart ($5.0\% \pm 4.0$), and in cardiac cell subpopulations including myocytes ($4.0\% \pm 3.0$), endothelial cells ($2.1\% \pm 2.8$), and fibroblasts ($0.54\% \pm 1.1$). Heart samples were also assessed for IDUA enzyme activity (3.9 ng/mg/hr ± 3.5) and GAG levels (3.3 μ g/mg protein ± 1.2). The observed enzyme activity in postnatally injected mice were significantly improved compared to *IDUA-W392X* controls ($p < 0.0001$), equivalent compared to *in utero* injected AAV.ABE.Idua mice (5.6 ng/mg/hr ± 3.0 , $p = 0.36$), and not equivalent to B6 controls (11.6 ng/mg/hr ± 7.4 , $p = 0.014$). Similarly, heart GAG levels were significantly improved compared to *IDUA-W392X* controls (7.2 μ g/mg protein ± 3.9 , $p = 0.0063$), equivalent compared to *in utero* injected AAV.ABE.Idua mice (4.1 μ g/mg protein ± 1.8 , $p = 0.24$), and not equivalent to B6 controls (1.1 μ g/mg protein ± 1.0 , $p = 0.0047$).”

4. As stated in the discussion that *in utero* AAV9 injection needs to consider the risk to mother, did the authors examine whether the *in utero* injection transduced any tissues in the mother?

As the Reviewer insightfully points out, risk to the mother is a critical consideration in any prenatal intervention.

To address this question, we assessed the presence of fluorescent expression in tissues from two C57BL/6 mothers in whom fetuses underwent *in utero* injection of AAV.GFP.mCherry. Seven days post injection, no fluorescent expression was detected on flow cytometry in brains, hearts, muscles, livers, or uteri suggesting no maternal transduction occurred.

We also assessed evidence of base editing in tissues from two *IDUA-W392X* mothers in whom fetuses underwent editing following *in utero* injection. In addition to evaluating genomic DNA from the uteri, we isolated DNA from the hearts and livers, organs with strong AAV9 tropism and in which we saw evidence of editing in *in utero* injected AAV.ABE.Idua fetuses. NGS editing efficiencies were evaluated in livers (0.07%, 0.21%), hearts (0.04%, 0.01%) and uteri (0.02%, 0.02%) from dams that underwent *in utero* surgery, and were no higher than that in uninjected control livers and hearts (0.19-0.49%). Given this finding, we believe *in utero* injection of AAV.ABE.Idua poses minimal editing risk to the mother. Of note this is also consistent with our previously published study in which adenovirus was used to deliver the cytosine base editor, BE3, *in utero* in the mouse model. In this study, NGS of DNA isolated from maternal organs demonstrated no significant editing above background³⁰.

Please see our prior response (Reviewer 1, Question 6) with respect to additions and modifications to the manuscript.

5. How many MPS-IH patients may be treated with adenine base editing?

The overall incidence of MPS-IH is 1:100,000 in Western society. 42.8% of patients harbor the W402X mutation of interest. This suggests, given the 2018 birth rate, approximately 16 new US babies per year could benefit from intervention. Given an average lifespan of 10 years for the most severe form of disease, the prevalence of disease between age one and ten is likely 160 children in the United States.

We have modified the first paragraph of the manuscript as follows:

“Lysosomal storage disorders affect multiple organs, have limited treatments, and have pathology that begins before birth¹⁸. In MPS-IH, *IDUA* gene mutations cause α -L-iduronidase (IDUA) deficiency and lysosomal accumulation of glycosaminoglycans (GAGs). The incidence of MPS-IH is 1:100,000 in Western society and one of the most common mutations (G→A; tryptophan→stop; W402X) accounts for over 40% of patients, results in undetectable IDUA in the homozygous state, and has a strong genotype-phenotype correlation³⁷.”

Reviewer #3:

The authors describe “*In utero* split AAV9 adenine base editing corrects the multi-organ pathology in a lethal lysosomal storage disease.” *In utero* base editing has the potential to correct disease-causing mutations before the onset of irreversible pathology. MPS-IH is a LSD affecting multiple organs, often leading to early postnatal cardiopulmonary demise. The authors assessed *in utero* AAV9 delivery of an adenine base editor (ABE) targeting the Idua W392X mutation in the MPS-IH mouse, corresponding to the common IDUA W402X mutation in MPS-IH patients. They show efficient long-term W392X correction in hepatocytes and cardiomyocytes and low-level editing in

the brain. *In utero* editing was associated with improved survival and amelioration of metabolic, musculoskeletal, neurologic, and cardiac disease. This proof-of-concept study demonstrates the possibility of efficiently performing therapeutic base editing in multiple organs before birth via a clinically relevant delivery mechanism, highlighting the potential of this approach for MPS-IH and other genetic diseases. The manuscript is written concisely, but there are several critical comments.

1. Provide the vector maps in details since the proposed study suggests the complicated strategy.

We have provided vector maps for AAV9.SpCas9.mTmG and AAV.ABE.Idua as a new supplemental figure. Additionally, we are happy to provide sequence files on request.

New Figure S4 depicts vector maps for split AAVs employed in the study as follows:

New Figure S4: Vector maps depicting N- and C-termini of split AAV constructs employed as AAV.SpCas9.mTmG and AAV.ABE.Idua.

2. *In utero* AAV gene therapy; it remains unclear how this strategy can apply to human patients since newborn screening for MPS I is now popular to identify the patients. How the authors identify the patients in fetus?

Thank you for this important question. The success of prenatal gene editing will depend on reliable and accurate prenatal testing as well as rigorous studies in small and large animal models demonstrating its safety and efficacy. Please see our response to Reviewer 1, Question 1 in which we detail the feasibility of prenatal diagnosis of MPS-IH and the rationale for the likely scenario that initial human application would involve families in which a sibling was previously diagnosed with MPS-IH. Furthermore, as described above, and as supported by a recent manuscript detailing prenatal enzyme replacement therapy for MPS VII³⁸, it is believed that the pathology for many lysosomal storage diseases, including MPS-IH, begins prior to birth and would benefit from prenatal treatment as is currently being investigated in a clinical trial (NCT04532047; In Utero Enzyme Replacement Therapy for Lysosomal Storage Diseases). Together with the literature cited in our response to Reviewer 1, we respectfully believe this trial highlights the ability to identify fetuses with MPS-IH and the merit of investigating novel prenatal treatments that could improve the lives of these patients and their families.

Please see our prior response (Reviewer 1, Question 1) with respect to additions and modifications to the manuscript.

3. Postnatal experiments; why injected via the retroorbital vein at 10 weeks-of-age? Any reason to use the retroorbital vein? Why at 10 weeks old?

Thank you for this clarifying question.

Retroorbital vein injection is a route of intravascular systemic injection in adult mouse models and one that we and others have used routinely. In our previous publication, we demonstrated efficient base editing at 1-month post injection in an adult mouse model following retroorbital vein injection of an adenovirus carrying a cytosine base editor³⁰. Ten weeks was chosen to ensure that the adult mice were immunologically mature.

4. Glycosaminoglycan (GAG) measurements; Why did the authors use total GAG assay by the dye method? This is semi-quantitative. Use LC-MS/MS method to measure individual GAGs (HS, DS, and KS) in urine, plasma (or serum) and tissues.

We thank the Reviewer for this specialized insight. We utilized the Blyscan GAG assay as we found that it was well-established and well-characterized in the literature with respect to assessing the MPS-IH phenotype³⁹⁻⁴³. As we do not have LC-MS/MS capabilities or expertise in our lab, this was not a technique we could readily employ for current studies. In addition, we believe and have found that the dye method offers adequate resolution with which to assess differences in total GAGs between experimental and control animals and presents results that are consistent with those realized via the other robust biochemical and phenotypic assays described in our manuscript.

However, to address the Reviewer's request, we have explored the feasibility of utilizing LC-MS/MS to measure individual GAGs in urine, serum, and tissue samples derived from the mice described in our experiment. To do so, we contacted three cores at our institutions. First, two of three cores are either not experienced in the specific published techniques for GAG determination or unsure of the feasibility of set up. Second given delays related to COVID-19 and limitations on research personnel on campus, the common estimate to establish a protocol for GAG determination and completion of the study is > 8 months. Third, the costs associated with LC-MS/MS are projected at \$1500 for set up and \$100 per sample. For the approximately 660 samples assessed in this manuscript, the total estimated cost for LC-MS/MS is \$67500 which is unfortunately outside of our budget. Given the multiple other phenotypic and biochemical outcome assays that were performed for the current study (IDUA activity, CT scan, cardiac echocardiography, grip strength, histology), we believe that the total GAG assay is adequate for the hypotheses we are addressing especially given the substantial costs and time that would be required for LC-MS/MS GAG determination.

5. Histology; it is a standard to use toluidine blue staining with 0.5 um to see vacuoles. Specify the extent of vacuoles or the size of the cells, especially in neurons and chondrocytes.

Thank you again for this suggestion. We elected to utilize Alcian blue staining in this manuscript given the substantial literature precedent for the use of Alcian blue to stain glycosaminoglycans and our own experience with this stain⁴⁴⁻⁴⁸. In contrast, the literature characterization of bone pathology in the MPS-IH mouse is largely based on a different knockout mouse. As we did not seek to conduct a primary assessment of cellular bone pathology in the *IDUA-W392X* model, we did not initially pursue a histologic assessment of chondrocytes.

To address the comments of the Reviewer, we have re-sectioned paraffin-embedded brains from mice included in our manuscript and restained with Toluidine Blue to evaluate vacuoles in neurons and the hippocampus. We assessed the extent of vacuoles and the size of cells in high-powered fields of the hippocampus from prenatally AAV.ABE.Idua injected *IDUA-W392X* mice, uninjected C57BL/6 controls, and uninjected *IDUA-W392X* mice at 6 months of age. This was graded in a blinded fashion and we did not appreciate a significant improvement in these characteristics in AAV.ABE.Idua injected mice compared to *IDUA-W392X* controls which is consistent with our finding of limited brain editing and no IDUA activity in treated mice.

We have modified the Results section as follows:

“The hippocampi of six-month-old prenatal AAV.ABE.Idua injected *IDUA-W392X* mice were evaluated by Toluidine Blue histopathology and demonstrated nominal improvement in cellular vacuolization and no difference in cell size compared to uninjected *IDUA-W392X* mice (Figure S3). However, Alcian Blue histology of the occipital cortex demonstrated decreased overall GAG staining (Figure 3a-b, g-h, m-n, s).”

We have also included a new Supplementary figure (Figure S3) to detail these findings as follows:

New Figure S3: Toluidine blue staining of hippocampal regions of C57BL/6, prenatal AAV.ABE.Idua injected *IDUA-W392X*, and *IDUA-W392X* mice highlighting vacuolated cells.

The figure legend was updated to reflect this new figure as follows:

“Hippocampal cellular pathology following *in utero* base editing. (a-f) Representative histology with Toluidine staining of the hippocampal region in 6-month-old C57BL/6 mice (a, b), prenatal AAV.ABE.Idua injected *IDUA-W392X* mice (c, d), and uninjected *IDUA-W392X* mice (e, f). Blinded assessments of cellular vacuolization and cell size were conducted in prenatal AAV.ABE.Idua injected mice (n=10), C57BL/6 mice (n=2), and *IDUA-W392X* mice (n=2). Vacuolization scores were assessed as follows: 0, no cytoplasmic vacuoles; +, rare vacuolated cell (<1%); ++, cytoplasmic vacuoles in 0–10% of cells; +++, cytoplasmic vacuoles in 10–25% of cells; +++++, cytoplasmic vacuoles in >25% of cells. Cell size was assessed compared to C57BL/6 mice as follows: 0, equivalent; +, 0-25% larger; ++, 25-50% larger; +++, 50-75% larger; +++++, >75% larger. Mean pathologic scores within groups are displayed in inset panels. (a,c,e) Scale bar=100µm. (b,d,f) Scale bar=25µm.”

We have also modified the fourth paragraph of the Discussion as follows:

“We demonstrated variable low-level editing in the brain following prenatal base editing (highest level of brain editing = 9%), with a concomitant reduction in GAG staining on Alcian blue histology as well as improvement in some components of the open-field test. However, the degree of hippocampal cellular vacuolization on Toluidine staining was not different between treated and untreated animals. Although low-level brain editing may contribute to some phenotypic improvement, we did not detect significant tissue IDUA activity in perfused brains. The nominal pathologic improvements therefore likely result from secretion of IDUA into serum, which may affect the brain and other low or unedited organs.”

The histology section of the Methods section has also been updated to reflect these changes:

“Toluidine Blue was used specifically to assess vacuolization and cell size in the hippocampus which was graded (0-++++) in a blinded fashion as per previously published methods⁴⁹. Vacuolization scores were assessed as follows: 0, no cytoplasmic vacuoles; +, rare vacuolated cell (<1%); ++, cytoplasmic vacuoles in 0–10% of cells; +++, cytoplasmic vacuoles in 10–25% of cells; +++++, cytoplasmic vacuoles in >25% of cells. Cell size was assessed compared to C57BL/6 mice as follows: 0, no different; +, <25% larger; ++, 25–50% larger; +++, 50–75% larger; +++++, >75% larger.”

To further address the Reviewer’s comment regarding chondrocyte staining, we attempted to evaluate bone pathology in the growth plate region, particularly with respect to chondrocytes. Although we did not have any fixed bones available for analysis, we thawed and processed femurs and knee joints from mice involved in the study to evaluate chondrocytes and the growth plate region (stored at -80° for 6 months). Unfortunately, due to the degree of freezing artifact (Reviewer Figure 2), it is difficult to draw conclusions about differences in chondrocytes and the growth plate in control and experimental animals. Therefore, we have not included these analyses.

Reviewer Figure 2: Toluidine staining of femoral growth plates in B6, prenatally injected AAV.ABE.Idua, and *IDUA-W392X* mice.

6. Specify the bone pathology in growth plate region as well as brain hippocampus area.

Please see the response above to Question 5.

7. Why were the enzyme activities were increased only in liver and cardiac muscle? Any reason? Why enzyme activity in brain is trivial? How about the enzyme activity in bone?

We found that enzyme activity was correlated with the degree of editing. Accordingly, tissues with robust editing demonstrated increased IDUA activity. Given the tropism of AAV9 and the systemic route of injection, in which the initial organs that encounter the injected viral vector are the liver and then the heart, increased editing and thus increased enzyme activity in these organs is expected. In addition, enzyme function in the cerebellum in prenatal AAV.ABE.Idua injected *IDUA-W392X* mice was not statistically significantly higher than that in untreated *IDUA-W392X* controls. This finding is in line with the low-level editing in the brain observed on NGS.

In order to respond to the Reviewer's observations and confirm our previous findings that brains of prenatally treated *IDUA-W392X* mice had undetectable IDUA activity, we reanalyzed frozen brain samples (stored at -80°C) to assess IDUA activity in different brain regions. Whereas we had previously sampled the cerebellum and occipital cortex, we resampled middle cortex and forebrain but did not find evidence of IDUA activity in these tissues.

With respect to the bone, to our knowledge, there is no published reliable processing approach to assess IDUA activity in bone. However, in a good faith effort, we use a mortar and pestle to manually crush and then homogenize the acetabula and proximal femurs (stored at -80°C for 6 months) from experimental and control mice to assess IDUA activity. We sampled this location to minimize contamination from the bone marrow in which editing of the lymphohematopoietic compartment may confound the results. We detected IDUA activity in prenatally AAV.ABE.Idua injected *IDUA-W392X* mice that was 1.2% of that seen in B6 mice. In addition, we found 0.29% on-target editing by NGS of genomic DNA isolated from this region. This compares to background levels of 0.19% in uninjected controls and thus does not likely represent editing. Thus, the presence of active enzyme in the bone likely reflects secreted IDUA, that nonetheless, may be adequate to ameliorate pathology.

To account for these findings, we have updated Figures 1c, 2c, and 2f to present the bone findings. Please see our prior response (Reviewer 1, Question 7) with respect to graphical additions and modifications to the manuscript.

In addition, we have modified the fifth paragraph of the Discussion to include the following:

“Although low-level brain editing may contribute to some phenotypic improvement, we did not detect significant tissue IDUA activity in the brain following PBS perfusion. The nominal pathologic improvements therefore likely result from secretion of IDUA into serum, which may affect the brain and other low or unedited organs.”

Finally, we have updated the Methods to reflect this new analysis:

“After organ harvest, 20mg tissue samples were homogenized with 0.1% Triton X-100 lysis buffer using a TissueLyser LT (Qiagen, Hilden, Germany). Of note, proximal femurs included acetabula were crushed using a mortar and pestle prior to homogenization”

8. References; cite the following articles

A Highly Efficacious PS Gene Editing System Corrects Metabolic and Neurological Complications of Mucopolysaccharidosis Type I.

Ou L, Przybilla MJ, Ahlat O, Kim S, Overn P, James J, O'Sullivan MG, Whitley CB, Ou L, et al. Mol Ther. 2020 Jun 3;28(6):1442-1454.

Human genome-edited hematopoietic stem cells phenotypically correct Mucopolysaccharidosis type I.

Gomez-Ospina N, Scharenberg SG, Mostrel N, Bak RO, Mantri S, Quadros RM, Gurumurthy CB, Lee C, Bao G, Suarez CJ, Khan S, Sawamoto K, Tomatsu S, Raj N, Attardi LD, Aurelian L, Porteus MH, Gomez-Ospina N, et al. Among authors: tomatsu s. Nat Commun. 2019 Sep 6;10(1):4045

We have included these citations in the list of references.

9. Describe the successful gene therapy by lenti-virus vector on MPS I patients.

We thank the Reviewer for bringing this to our attention.

At the time of submission, we were not aware of the ongoing clinical trial which investigates lenti-viral mediated IDUA replacement in autologous CD34+ hematopoietic stem/progenitor cells (HSPCs) for subsequent retransplantation⁵⁰. Although there are numerous published studies involving this approach for other diseases, we are unable to find a peer-reviewed manuscript regarding this treatment approach in MPS-IH patients. However, an abstract was presented by Orchard Therapeutics at the American Society of Gene and Cell Therapy (ASGCT) 22nd Annual Meeting in 2019 and detailed phase I/II data in which two patients were treated with autologous HSPCs transduced with a lentivirus carrying the *IDUA* transgene following conditioning with fludarabine/busulfan. They demonstrated normalization of blood and CSF IDUA and a decrease in GAGs^{51,52}.

Early data from this trial is certainly encouraging. However, we note that there are some potential limitations to this approach including a potential risk of insertional mutagenesis and aberrant splicing associated with lentiviruses^{53,54}, the need for peri-transplant conditioning regimens, and the likelihood that this approach will have the same phenotypic correction seen following allogeneic HSCT. Finally, in light of the potential benefit of prenatal/neonatal therapy for MPS-IH, the requisite HSC harvest and conditioning prior to therapy present a barrier to early intervention.

In response to the Reviewer's query, we have highlighted the encouraging early results from this study in the Discussion:

“MPS-IH and other lysosomal storage diseases represent attractive targets for prenatal base editing as the multi-organ disease pathology is progressive and begins prior to birth. Furthermore, the bone and cardiac pathologies are not responsive to the limited current postnatal treatments including enzyme replacement and HSCT^{24,55-57}. As an alternative, an ongoing trial of autologous transplantation of hematopoietic stem/progenitor cells treated with a lentivirus containing the *IDUA* transgene has had early encouraging results^{50,52}. This approach, however, is limited to postnatal treatment and is likely to have the same phenotypic correction seen following allogenic HSCT. In our study, prenatal base editing was shown to correct the disease-causing mutation in multiple disease-affected organs in the *IDUA-W392X* MPS-IH mouse model, with high-level correction in the liver and cardiomyocytes. Efficient prenatal base editing was associated with amelioration of the cardiac and musculoskeletal phenotypes at six months of age.”

Reviewer References

1. Akella, R. R. D. & Kadali, S. Amniotic fluid glycosaminoglycans in the prenatal diagnosis of mucopolysaccharidoses - A useful biomarker. *Clinica Chimica Acta* **460**, 63–66 (2016).
2. A, A. N. & E, F. Prenatal diagnosis of mucopolysaccharidoses (MPS): the first Egyptian experience. *Bratisl Lek Listy* **105**, 310–314 (2004).
3. Mucopolysaccharidosis I mutations in Chinese patients: identification of 27 novel mutations and 6 cases involving prenatal diagnosis - Wang - 2012 - Clinical Genetics - Wiley Online Library. https://onlinelibrary-wiley-com.proxy.library.upenn.edu/doi/full/10.1111/j.1399-0004.2011.01680.x?casa_token=VKzLnCAg5PMAAAAAA%3A8jqER1BnGDX93_vZyo6-n3bdFFFA3N_k8pbzJlShirSUfehrc0efkUtxy5K6jM265c5gqNp4k1WSfA.
4. Meaney, F. J., Riggle, S. M., Cunningham, G. C., Stern, K. S. & Davis, J. G. Prenatal genetic services: toward a national data base. *Clin Obstet Gynecol* **36**, 510–520 (1993).
5. Akolekar, R., Beta, J., Picciarelli, G., Ogilvie, C. & D'Antonio, F. Procedure-related risk of miscarriage following amniocentesis and chorionic villus sampling: a systematic review and meta-analysis. *Ultrasound in Obstetrics & Gynecology* **45**, 16–26 (2015).
6. Preparent® Carrier Test | Progenity. <https://www.progenity.com/products/preparent>.
7. 500 Plus Panel | Integrated Genetics. <https://www.integratedgenetics.com/patients/pre-pregnancy/inheritest/500-plus>.
8. Zhang, J. *et al.* Non-invasive prenatal sequencing for multiple Mendelian monogenic disorders using circulating cell-free fetal DNA. *Nature Medicine* **25**, 439–447 (2019).
9. Wang, X. *et al.* Mucopolysaccharidosis I mutations in Chinese patients: identification of 27 novel mutations and 6 cases involving prenatal diagnosis. *Clinical Genetics* **81**, 443–452 (2012).

10. Osborn, M. J., McElmurry, R. T., Peacock, B., Tolar, J. & Blazar, B. R. Targeting of the CNS in MPS-IH Using a Nonviral Transferrin- α -l-iduronidase Fusion Gene Product. *Molecular Therapy* **16**, 1459–1466 (2008).
11. Lin, S.-P. *et al.* A pilot newborn screening program for Mucopolysaccharidosis type I in Taiwan. *Orphanet J Rare Dis* **8**, 147 (2013).
12. Miebach, E. Enzyme replacement therapy in mucopolysaccharidosis type I. *Acta Paediatrica* **94**, 58–60 (2005).
13. Prenatal pathology in mucopolysaccharidoses: a comparison with postnatal cases. - Abstract - Europe PMC. <https://europepmc.org/article/med/6226467>.
14. Muenzer, J. Early initiation of enzyme replacement therapy for the mucopolysaccharidoses. *Molecular Genetics and Metabolism* **111**, 63–72 (2014).
15. Kiely, B. T., Kohler, J. L., Coletti, H. Y., Poe, M. D. & Escolar, M. L. Early disease progression of Hurler syndrome. *Orphanet J Rare Dis* **12**, 32 (2017).
16. Gabrielli, O., Clarke, L. A., Bruni, S. & Coppa, G. V. Enzyme-Replacement Therapy in a 5-Month-Old Boy With Attenuated Presymptomatic MPS I: 5-Year Follow-up. *Pediatrics* **125**, e183–e187 (2010).
17. In Utero Enzyme Replacement Therapy for Lysosomal Storage Diseases - Full Text View - ClinicalTrials.gov. <https://clinicaltrials.gov/ct2/show/NCT04532047>.
18. Ferreira, C. R. & Gahl, W. A. Lysosomal storage diseases. *Translational Science of Rare Diseases* **2**, 1–71 (2017).
19. Crawford, M. d'A *et al.* Early Prenatal Diagnosis of Hurler's Syndrome with Termination of Pregnancy and Confirmatory Findings on the Fetus. *Journal of Medical Genetics* **10**, 144–153 (1973).

20. Crow, J., Gibbs, D. A., Cozens, W., Spellacy, E. & Watts, R. W. Biochemical and histopathological studies on patients with mucopolysaccharidoses, two of whom had been treated by fibroblast transplantation. *Journal of Clinical Pathology* **36**, 415–430 (1983).
21. Ikeno, T. *et al.* Prenatal diagnosis of Hurler’s syndrome—Biochemical studies on the affected fetus. *Hum Genet* **59**, 353–359 (1981).
22. Weber, R. *et al.* Spectrum and Outcome of Primary Cardiomyopathies Diagnosed During Fetal Life. *J Am Coll Cardiol HF* **2**, 403–411 (2014).
23. Jj, M. & C, C. Prenatal pathology in mucopolysaccharidoses: a comparison with postnatal cases. *Clin Neuropathol* **2**, 122–127 (1983).
24. Tolar, J. & Orchard, P. J. α -L-iduronidase therapy for mucopolysaccharidosis type I. *Biologics* **2**, 743–751 (2008).
25. Baldo, G. *et al.* Enzyme replacement therapy started at birth improves outcome in difficult-to-treat organs in mucopolysaccharidosis I mice. *Molecular Genetics and Metabolism* **109**, 33–40 (2013).
26. Braunlin, E. A. *et al.* Cardiac disease in patients with mucopolysaccharidosis: presentation, diagnosis and management. *J Inherit Metab Dis* **34**, 1183–1197 (2011).
27. Ahmed, A. *et al.* Mucopolysaccharidosis (MPS) Physical Symptom Score: Development, Reliability, and Validity. in *JIMD Reports, Volume 26* (eds. Morava, E. *et al.*) 61–68 (Springer, 2016). doi:10.1007/8904_2015_485.
28. Boelig, M. M. *et al.* The Intravenous Route of Injection Optimizes Engraftment and Survival in the Murine Model of In Utero Hematopoietic Cell Transplantation. *Biology of Blood and Marrow Transplantation* **22**, 991–999 (2016).

29. Zwiers, C. *et al.* Complications of intrauterine intravascular blood transfusion: lessons learned after 1678 procedures. *Ultrasound in Obstetrics & Gynecology* **50**, 180–186 (2017).
30. Rossidis, A. C. *et al.* In utero CRISPR-mediated therapeutic editing of metabolic genes. *Nature Medicine* **24**, 1513–1518 (2018).
31. Hopwood, J. J. & Muller, V. Biochemical discrimination of hurler and scheie syndromes. *Pathology* **11**, 327 (1979).
32. Bunge, S. *et al.* Genotype–phenotype correlations in mucopolysaccharidosis type I using enzyme kinetics, immunoquantification and in vitro turnover studies. *Biochimica et Biophysica Acta (BBA) - Molecular Basis of Disease* **1407**, 249–256 (1998).
33. Li, A. *et al.* AAV-CRISPR Gene Editing Is Negated by Pre-existing Immunity to Cas9. *Molecular Therapy* **28**, 1432–1441 (2020).
34. Subacute Liver Failure Following Gene Replacement Therapy for Spinal Muscular Atrophy Type 1 - The Journal of Pediatrics. [https://www.jpeds.com/article/S0022-3476\(20\)30682-X/abstract](https://www.jpeds.com/article/S0022-3476(20)30682-X/abstract).
35. Shieh, P. B. *et al.* Re: “Moving Forward After Two Deaths in a Gene Therapy Trial of Myotubular Myopathy” by Wilson and Flotte. *Human Gene Therapy* **31**, 787–787 (2020).
36. Addgene: AAV Titration by qPCR. <https://www.addgene.org/protocols/aav-titration-qpcr-using-sybr-green-technology/>.
37. Bunge, S. *et al.* Mucopolysaccharidosis type I: identification of 8 novel mutations and determination of the frequency of the two common α -L-iduronidase mutations (W402X and Q70X) among European patients. *Hum Mol Genet* **3**, 861–866 (1994).

38. Nguyen, Q.-H. *et al.* Tolerance induction and microglial engraftment after fetal therapy without conditioning in mice with mucopolysaccharidosis type VII. *Science Translational Medicine* **12**, (2020).
39. Wang, D. *et al.* Cas9-mediated allelic exchange repairs compound heterozygous recessive mutations in mice. *Nature Biotechnology* **36**, 839–842 (2018).
40. Ou, L., Przybilla, M. J., Koniar, B. L. & Whitley, C. B. Elements of lentiviral vector design toward gene therapy for treating mucopolysaccharidosis I. *Molecular Genetics and Metabolism Reports* **8**, 87–93 (2016).
41. Gunn, G. *et al.* Long-term nonsense suppression therapy moderates MPS I-H disease progression. *Molecular Genetics and Metabolism* **111**, 374–381 (2014).
42. Keeling, K. M. *et al.* Attenuation of Nonsense-Mediated mRNA Decay Enhances In Vivo Nonsense Suppression. *PLOS ONE* **8**, e60478 (2013).
43. Ou, L. *et al.* A Highly Efficacious PS Gene Editing System Corrects Metabolic and Neurological Complications of Mucopolysaccharidosis Type I. *Molecular Therapy* **28**, 1442–1454 (2020).
44. Hinderer, C. *et al.* Neonatal tolerance induction enables accurate evaluation of gene therapy for MPS I in a canine model. *Molecular Genetics and Metabolism* **119**, 124–130 (2016).
45. Hinderer, C. *et al.* Neonatal Systemic AAV Induces Tolerance to CNS Gene Therapy in MPS I Dogs and Nonhuman Primates. *Molecular Therapy* **23**, 1298–1307 (2015).
46. Mahalingam, K., Janani, S., Priya, S., Elango, E. M. & Sundari, R. M. Diagnosis of mucopolysaccharidoses: How to avoid false positives and false negatives. *Indian J Pediatr* **71**, 29–32 (2004).

47. Kobayashi, H. *et al.* Neonatal Gene Therapy of MPS I Mice by Intravenous Injection of a Lentiviral Vector. *Molecular Therapy* **11**, 776–789 (2005).
48. Miyadera, K. *et al.* Intrastromal Gene Therapy Prevents and Reverses Advanced Corneal Clouding in a Canine Model of Mucopolysaccharidosis I. *Molecular Therapy* **28**, 1455–1463 (2020).
49. Hartung, S. D. *et al.* Correction of metabolic, craniofacial, and neurologic abnormalities in MPS I mice treated at birth with adeno-associated virus vector transducing the human α -l-iduronidase gene. *Molecular Therapy* **9**, 866–875 (2004).
50. Gene Therapy With Modified Autologous Hematopoietic Stem Cells for the Treatment of Patients With Mucopolysaccharidosis Type I, Hurler Variant - Full Text View - ClinicalTrials.gov. <https://clinicaltrials.gov/ct2/show/NCT03488394>.
51. Orchard Therapeutics Licenses MPS-I Gene Therapy from SR-TIGET. *GEN - Genetic Engineering and Biotechnology News* <https://www.genengnews.com/news/orchard-therapeutics-licenses-mps-i-gene-therapy-from-sr-tiget/> (2019).
52. Gentner, B. *et al.* Ex-Vivo Gene Therapy for Hurler Disease: Initial Results from a Phase I/II Clinical Study. in *Clinical Trials Spotlight* vol. 27 1–2 (Molecular Therapy, 2019).
53. Rothe, M., Modlich, U. & Schambach, A. Biosafety Challenges for Use of Lentiviral Vectors in Gene Therapy. *Current Gene Therapy* **13**, 453–468 (2013).
54. Rothe, M., Schambach, A. & Biasco, L. Safety of gene therapy: new insights to a puzzling case. *Curr Gene Ther* **14**, 429–436 (2014).
55. Weisstein, J. S., Delgado, E., Steinbach, L. S., Hart, K. & Packman, S. Musculoskeletal Manifestations of Hurler Syndrome: Long-Term Follow-Up After Bone Marrow Transplantation. *Journal of Pediatric Orthopaedics* **24**, 97–101 (2004).

56. Taylor, C. *et al.* Mobility in Hurler Syndrome. *Journal of Pediatric Orthopaedics* **28**, 163–168 (2008).
57. Gomez-Ospina, N. *et al.* Human genome-edited hematopoietic stem cells phenotypically correct Mucopolysaccharidosis type I. *Nat Commun* **10**, 4045 (2019).

Reviewer comments, second round –

Reviewer #1 (Remarks to the Author):

All of my questions and concerns have been sufficiently addressed. The updates to the manuscript are thorough and appropriate. I congratulate the authors on their work and believe this manuscript to be suitable for publication in Nature Communications.

Reviewer #2 (Remarks to the Author):

The revised manuscript has been improved, but further work is required to substantiate the work.

1. Besides potential off-target DNA editing activities, the ABEmax has been shown to also induce transcriptome-wide RNA off-target activities, which should be examined by RNAseq.
2. It is noted that higher dosage of AAV greater than 1E14 vg/kg may result in toxicities in humans via systemic delivery, however, it is not clear if such toxicities would similarly be induced in embryos with in utero delivery when the immune system is not fully developed/mature. It's thus important to test different dosages of AAV9-ABE.Idua via in utero delivery to determine the editing efficiency vs toxicities.
3. The expression of full-length ABE and the potential N and C-terminal intein split fragments was not examined following AAV9-ABE.Idua delivery.
4. It is not clear why the dosage of AAV9-GFP/mCherry injected to estimate the transduction efficiency was ~2-3 times lower than AAV9-ABE.Idua. It would be better to use the same dosage as AAV9-ABE.Idua or to directly examine the relationship between the viral titres in the tissues vs the expression of full-length ABE.
5. A more appropriate control in the treatments is missing in the treatments – vectors with non-targeting guide RNA.

Reviewer #3 (Remarks to the Author):

The authors responded to the most comments by the reviewers.

1. Bone pathology; specify the data of evaluation of growth plate by light microscopy. Bone should be decalcified and toluidine blue-stained sections 0.5 µm thick should be examined. How about micro CT? Any data? This is the most valuable method to characterize the structure in bone.

Cite the following manuscripts related to bone pathology in MPS I;

Pievani A, Azario I, Antolini L, Shimada T, Patel P, Remoli C, Rambaldi B, Valsecchi MG, Riminucci M, Biondi A, Tomatsu S, Serafini M. Neonatal bone marrow transplantation prevents bone pathology in a mouse model of mucopolysaccharidosis type I. *Blood*. 2015 Mar 5;125(10):1662-71.

Azario I, Pievani A, Del Priore F, Antolini L, Santi L, Corsi A, Cardinale L, Sawamoto K, Kubaski F, Gentner B, Bernardo ME, Valsecchi MG, Riminucci M, Tomatsu S, Aiuti A, Biondi A, Serafini M. Neonatal umbilical cord blood transplantation halts skeletal disease progression in the murine model of MPS-I. *Sci Rep*. 2017; 7(1):9473.

2. GAG measurements; The reviewer feels very strange that the authors do not use LC-MS/MS method to measure specific GAGs (HS, DS, and KS) in urine, plasma (or serum), CSF, and tissues. The authors can collaborate with other institutes to measure the specific GAGs. Any problem to collaborate with other groups?
HS is related to CNS impairment while KS is related to bone disease. Therefore, this is critical to measure them now as a biomarker.

Overall, the manuscript should be revised according to the comments.

Dear Editorial team and Reviewers,

Thank you for the thorough and thoughtful review of our Article entitled “*In utero split AAV9 adenine base editing corrects the multi-organ pathology in a lethal lysosomal storage disease*”. We have provided additional data and made modifications in the manuscript indicated by the use of highlighting and/or provided clarifications to address the comments. We believe these changes have significantly improved the work. Below, please find a “point-by-point” response to the Reviewers’ comments.

Reviewer #1

All of my questions and concerns have been sufficiently addressed. The updates to the manuscript are thorough and appropriate. I congratulate the authors on their work and believe this manuscript to be suitable for publication in Nature Communications.

We thank Reviewer 1 for their comments and appreciation of the significant revisions that have been made to improve the quality of the work.

Reviewer #2

The revised manuscript has been improved, but further work is required to substantiate the work.

1. Besides potential off-target DNA editing activities, the ABEmax has been shown to also induce transcriptome-wide RNA off-target activities, which should be examined by RNAseq.

We thank the Reviewer for the insightful comment about the RNA off-target profile of ABEmax. Importantly, other recent influential papers (published recently in *Nature Biomedical Engineering*) evaluating adenine base editing for genetic diseases in mouse models have limited their analyses of off-target activities to DNA editing and not RNA¹⁻⁴. In addition, the current off-target limitations of ABEmax are likely to be overcome with continued evolution of adenine base editors prior to clinical translation, and we do not suggest that proceeding with ABEmax into human trials is warranted without additional data. Finally, as we consider clinical translation, the guide RNA targeting the mouse IDUA gene is different than that which would be used to target the human mutation. We believe that RNA off-target analyses are of significantly more importance and are essential in studies in which editing is assessed in the context of the human genome.

We have amended the eighth paragraph of the Discussion to highlight the proof-of-concept nature of our study and the need for these future studies in human specimens and prior to clinical translation:

“Although we did not appreciate any off-target DNA editing activity in the current proof-of-concept study, ABEmax has also been shown to induce transcriptome-wide RNA off-target activities⁵. This potential limitation may be addressed with continued evolution of base editing enzymes, however, future studies in large animal models and human samples assessing additional off-target analyses including transcriptome-wide RNA off-target activities are essential prior to clinical translation.”

2. It is noted that higher dosage of AAV greater than 1E14 vg/kg may result in toxicities in humans

via systemic delivery, however, it is not clear if such toxicities would similarly be induced in embryos with in utero delivery when the immune system is not fully developed/mature. It's thus important to test different dosages of AAV9-ABE.Idua via in utero delivery to determine the editing efficiency vs toxicities.

Thank you for this comment. We elected to administer $6.5E13$ vg/kg, less than the currently known toxic dose of AAV, and did not note any overt toxicities as indicated by the survival data. Notably, the toxicity of AAV is believed to be related to an immune/inflammatory response to the vector. Thus, as the Reviewer points out, the immature fetal immune system would potentially facilitate higher doses with less toxicity than is appreciated with postnatal delivery. We appreciate this consideration and believe it is an important question for *in utero* gene therapy/editing using AAV vectors in general. However, the purpose of the current proof-of-concept study was to demonstrate the feasibility of adenine base editing, and specifically the application of adenine base editing before birth, as a potential new therapy for MPS-IH. We agree that dose-toxicity studies of *in utero* AAV administration (independent of the transgene) should be further explored in large animal preclinical studies but believe such studies are outside the scope of the current work.

We have revised the third paragraph of the Discussion to note the importance of dose-toxicity studies in future preclinical work prior to clinical translation as follows:

“Importantly, the toxic dose range of AAV has not been thoroughly evaluated in the prenatal setting. However, given the immaturity of the fetal immune system, it is conceivable that the maximal tolerated dose may be different than that observed in postnatal studies. Therefore, a thorough characterization of the in utero AAV dose-toxicity relationship should be conducted in large animal studies prior to clinical translation.”

3. The expression of full-length ABE and the potential N and C-terminal intein split fragments was not examined following AAV9-ABE.Idua delivery.

Intein-mediated split AAV delivery, including delivery of ABEmax via this approach, is well described in the literature with prior papers characterizing the particular inteins used in our construct^{2,6,7}. Although we did not demonstrate the presence of the full-length ABE, the presence of efficient editing in the heart and liver is indicative that the functional, full length ABE was formed. Similarly, the primary endpoints of the current study are editing and organ phenotypes, and the expression level of ABE would not meaningfully inform the study, as it would not change the interpretation of the primary endpoints. For these reasons, we do not believe the use of an intein strategy necessitates further interrogation in our work.

To address the Reviewer's comment, we have referenced the prior publication noting the use of an intein-mediated dual AAV delivery system for base editing and added the following sentence to the first section of the Results:

“Due to the limited AAV9 packaging capacity, we sought to deliver the ABE-guide RNA (gRNA) transgene in split AAVs with reconstitution of the ABE protein *in vivo* via inteins². This approach has previously demonstrated the ability to produce a functioning adenine base editor *in vivo*.”

4. It is not clear why the dosage of AAV9-GFP/mCherry injected to estimate the transduction efficiency was ~2-3 times lower than AAV9-ABE.Idua. It would be better to use the same dosage as AAV9-ABE.Idua or to directly examine the relationship between the viral titres in the tissues vs the expression of full-length ABE.

The objective of the GFP/mCherry experiment was to address the Reviewer’s question of whether increasing the delivered dose of AAV.ABE.IDUA might increase base editing.

Although both viral doses were in the 10^{10} range, the dosage of AAV.GFP.mCherry (~ 2.5×10^{10}) was not perfectly titre matched to AAV.ABE.IDUA (~ 5×10^{10}) due to the practicalities of available virus and number of fetuses in the operative litters. However, the relationship between AAV9 dose and tissue titres has previously been demonstrated to be linear in the dose range we employed (Reviewer Figure 1 from Prasad et al.)⁸.

Reviewer Figure 1 (Figure 4b from Prasad et al.⁸): “Graph illustrating the linear correlation between vector dose and mean vector genome copy number per µg of genomic DNA for each of the five capsid serotypes.”

As such, the starting dose of AAV.GFP.mCherry is unlikely to affect the described correlation between transduction percentage and tissue viral titres and as such does not alter the conclusions of the experiment which suggest that enhancing transduction has the potential to increase gene editing.

With respect to using mice injected with AAV.ABE.Idua, these mice were 6 months old at sacrifice. Therefore, neither the presence of virus in the tissue nor the expression of full-length ABE are reliable estimates of initial viral transduction and base editing. To better address the Reviewer’s concern, we would seek to inject new litters of MPS-IH fetuses with AAV.ABE.Idua, sacrifice them at 1 week of age, and correlate ABE protein expression by western blot to viral titres in the tissue by qPCR. Notably, MPS-IH mice are very poor breeders and the time required to generate additional time-dated pregnancies and the analyses required for these proposed studies would be substantial. Finally, and most importantly, we do not believe this line of experimentation would lead to a different conclusion than already reached, namely that higher editing may be possible with higher transduction efficiency.

To address the Reviewer's comment, we have added the following text to the second to last paragraph of the Discussion in which we discuss caveats/limitations of the study:

“Second, although our estimation of transduction efficiency as it relates to editing efficiency was determined by delivering 10^{10} vector copies of AAV.GFP.mCherry or AAV.ABE.Idua, the delivered dose did differ by $\sim 2\times$ ($2.5E10$ vs. $5E10$) due to experimental conditions. Notably, as prior work has demonstrated that AAV9 dose and tissue titres are linear in the 10^{10} dose range⁸, the described correlations between transduction percentage and tissue viral titres and the conclusion that enhancing transduction has the potential to increase gene editing are unlikely to be altered by these differences. However, studies of the dose-editing relationship are merited in future investigations.”

5. A more appropriate control in the treatments is missing in the treatments – vectors with non-targeting guide RNA.

Thank you for this comment.

Homozygous MPS-IH mice necessary for *in utero* experiments are not prolific breeders. Furthermore, to obtain accurate time-dated pregnancies, we only place male and female mice together for a 12-hour period. Thus, it is very difficult to obtain time-dated pregnancies *for in utero* injections which also has a procedural risk of mortality independent of mouse strain. Thus, to maximize the use of time-dated pregnancies, all fetuses of time-dated pregnancies were used to inject the experimental vector. Alternatively, non-time-dated pregnant MPS-IH mice can be obtained by leaving males and females together for longer periods of time. Although this increases the incidence of pregnancy, we cannot determine the exact gestational age of a fetus but only can note the day it was born. We therefore used these age-matched uninjected MPS-IH mice as controls. Of note, uninjected MPS-IH mice have previously been used as controls including studies evaluating gene editing technology in MPS-IH mice^{9,10}.

Notably, our study is a proof-of-concept to demonstrate the potential of on-target correction using adenine base editing and the high ($\sim 23\%$ in liver) levels of observed allelic correction are exceedingly unlikely due to chance. In addition, the guide RNA used in the mouse is different than the potential guide RNAs that might be used in the human. For these reasons, we believe the addition of a non-targeting control offers marginal value in the context of the overall objective of this mouse study but will be important in the context of preclinical large animal MPS studies.

As a point of clarification, for the survival studies requested by Reviewer 1, we employed a saline control as this purely addresses the effect of procedure vs. viral injectate on mortality.

To acknowledge the limitation described by the Reviewer, we have added the following text to the second to last paragraph of the Discussion in which we highlight caveats/limitations of the study:

“Third, our experiments utilized uninjected and PBS-injected disease mice as controls due to limited availability of time-dated pregnancies. Future studies, including those in large animals, may benefit from the addition of a vector control with non-targeting gRNA in order to more thoroughly interrogate the on- and off-target efficiencies and safety of the selected base editor.”

Reviewer #3

The authors responded to the most comments by the reviewers.

1. Bone pathology; specify the data of evaluation of growth plate by light microscopy. Bone should be decalcified and toluidine blue-stained sections 0.5 μm thick should be examined. How about micro CT? Any data? This is the most valuable method to characterize the structure in bone.

In response to the Reviewer's initial comment, we decalcified femurs from all mice described in the study and performed toluidine blue staining as recommended. However, we noted significant freezing artifact as the bone had been stored for 6 months at -80C. As such, we do not believe our specimens are appropriate for additional studies to address bone pathology.

The paper currently notes significant improvements in femur cortical area and thickness in treated mice. However, we also have data to suggest additional improvements in treated mice compared to disease controls including decreased femur bone volume, decreased endocortical perimeter/surface, decreased average marrow area, and decreased periosteal perimeter/surface. Unfortunately, the resolution of our CT scans is not adequate to reliably quantify growth plate differences or trabecular features with a high degree of certainty.

To address the Reviewer's commentary, we have amended the manuscript to include additional data from our microCT scans which supports phenotypic improvement in edited compared to unedited control mice. This data can be seen in new Figure S3.

Figure S3 *In utero* base editing improves the femoral bone phenotype in *IDUA-W392X* mice. (a-e) CT femur parameters were measured in 6-month-old B6 (N=10), 6-month-old *IDUA-W392X* mice prenatally injected with AAV.ABE.*Idua* (N=10), and surviving uninjected *IDUA-W392X* mice (N=6). Wilcoxon test for multiple comparisons was used to assess parameters given differing variances; $\alpha=0.05$. CT, computed tomography scan. AAV.ABE indicates AAV.ABE.*Idua*.

2. Cite the following manuscripts related to bone pathology in MPS I; Pievani A, Azario I, Antolini L, Shimada T, Patel P, Remoli C, Rambaldi B, Valsecchi MG, Riminucci M, Biondi A, Tomatsu S, Serafini M. Neonatal bone marrow transplantation prevents bone pathology in a mouse model of mucopolysaccharidosis type I. *Blood*. 2015 Mar 5;125(10):1662-71.

Azario I, Pievani A, Del Priore F, Antolini L, Santi L, Corsi A, Cardinale L, Sawamoto K, Kubaski F, Gentner B, Bernardo ME, Valsecchi MG, Riminucci M, Tomatsu S, Aiuti A, Biondi A, Serafini M. Neonatal umbilical cord blood transplantation halts skeletal disease progression in the murine model of MPS-I. *Sci Rep*. 2017; 7(1):9473.

Thank you for providing these references. We have added the references as requested by the Reviewer.

3. GAG measurements; The reviewer feels very strange that the authors do not use LC-MS/MS method to measure specific GAGs (HS, DS, and KS) in urine, plasma (or serum), CSF, and tissues.

The authors can collaborate with other institutes to measure the specific GAGs. Any problem to collaborate with other groups?

HS is related to CNS impairment while KS is related to bone disease. Therefore, this is critical to measure them now as a biomarker.

Thank you for this comment. We again highlight the prohibitive nature in time and resources of this request and note that the analysis of specific GAGs is peripheral to the focus of the paper, which is corrective editing and improvement in organ phenotype.

We believe that when considered along with deep sequencing and IDUA enzyme activity, the presented GAG measurements sufficiently support the conclusions stated in the paper. For example, determining the quantity of HS in the brain versus overall GAGs does not add substantial insight to the argument that overall brain correction was limited. Moreover, the colorimetric assay that we employed is a standard approach to determine GAGs in the MPS-I mouse literature^{9,11-13}.

To address the Reviewer's comments, we have communicated with our and neighboring institutions. Currently, each of the facilities we spoke with will require 3-6 months to develop a protocol for determination of GAG subtypes, a timeline that is exacerbated due to the COVID-19 backlog. Moreover, as previously mentioned, the cost to pursue de novo LC-MS will be significant (>\$40,000).

Nonetheless, we appreciate the Reviewer's expertise and have added the following sentence to the second to last paragraph of the Discussion stating:

"In addition, although the objective of our study was to assess gross phenotypic correction including changes in total GAG levels, future translational work would benefit from GAG subtype analysis utilizing liquid chromatography/mass spectrometry to identify organ-specific saccharide changes following editing since heparan sulfate and keratan sulfate levels have been shown to be related to CNS and bone pathology respectively^{14,15}."

References

1. Suh, S. *et al.* Restoration of visual function in adult mice with an inherited retinal disease via adenine base editing. *Nature Biomedical Engineering* 1–10 (2020) doi:10.1038/s41551-020-00632-6.
2. Levy, J. M. *et al.* Cytosine and adenine base editing of the brain, liver, retina, heart and skeletal muscle of mice via adeno-associated viruses. *Nature Biomedical Engineering* **4**, 97–110 (2020).
3. Song, C.-Q. *et al.* Adenine base editing in an adult mouse model of tyrosinaemia. *Nature Biomedical Engineering* **4**, 125–130 (2020).
4. Wang, X. *et al.* Efficient Gene Silencing by Adenine Base Editor-Mediated Start Codon Mutation. *Molecular Therapy* **28**, 431–440 (2020).
5. Richter, M. F. *et al.* Phage-assisted evolution of an adenine base editor with improved Cas domain compatibility and activity. *Nature Biotechnology* **38**, 883–891 (2020).
6. Villiger, L. *et al.* Treatment of a metabolic liver disease by in vivo genome base editing in adult mice. *Nature Medicine* **24**, 1519–1525 (2018).
7. Truong, D.-J. J. *et al.* Development of an intein-mediated split-Cas9 system for gene therapy. *Nucleic Acids Res* **43**, 6450–6458 (2015).
8. Prasad, K.-M. R., Xu, Y., Yang, Z., Acton, S. T. & French, B. A. Robust cardiomyocyte-specific gene expression following systemic injection of AAV: in vivo gene delivery follows a Poisson distribution. *Gene Therapy* **18**, 43–52 (2011).
9. Ou, L. *et al.* A Highly Efficacious PS Gene Editing System Corrects Metabolic and Neurological Complications of Mucopolysaccharidosis Type I. *Molecular Therapy* **28**, 1442–1454 (2020).

10. Ou, L. *et al.* ZFN-Mediated In Vivo Genome Editing Corrects Murine Hurler Syndrome. *Molecular Therapy* **27**, 178–187 (2019).
11. Ou, L., Przybilla, M. J., Koniar, B. L. & Whitley, C. B. Elements of lentiviral vector design toward gene therapy for treating mucopolysaccharidosis I. *Molecular Genetics and Metabolism Reports* **8**, 87–93 (2016).
12. Gunn, G. *et al.* Long-term nonsense suppression therapy moderates MPS I-H disease progression. *Molecular Genetics and Metabolism* **111**, 374–381 (2014).
13. Keeling, K. M. *et al.* Attenuation of Nonsense-Mediated mRNA Decay Enhances In Vivo Nonsense Suppression. *PLOS ONE* **8**, e60478 (2013).
14. Pievani, A. *et al.* Neonatal bone marrow transplantation prevents bone pathology in a mouse model of mucopolysaccharidosis type I. *Blood* **125**, 1662–1671 (2015).
15. Tomatsu, S. *et al.* Dermatan sulfate and heparan sulfate as a biomarker for mucopolysaccharidosis I. *J Inherit Metab Dis* **33**, 141–150 (2010).

Reviewer comments, third round –

Reviewer #3 (Remarks to the Author):

The authors responded to the comments in many points. However, there are some questions.

1. Figure S3; why treated mice had lower BV, perimeter, TC/MA, Av marrow area than WT mice? Is that adverse effect beyond therapeutic effect? Need to clarify it.

2. GAG assay; the rev

9. GAG levels;

Considering urinary levels of GAGs do not seem to have a clear direct correlation with the clinical improvement. It is already well known that urine GAG does not correlate with any clinical improvement in MPS. Urine GAGs are pharmacodynamic marker but not for therapeutic efficacy. Is there any reason why urine GAG is used for monitoring purpose in this experiment?

It is better to measure specific GAGs (HS, DS, and KS) in blood and CSF.

HS in CSF correlates with CNS involvement while KS in blood is biomarker for the bone.

There are several articles which question about the usefulness of urine GAG in therapeutic effect since the origin of urine, blood, and tissue GAGs are different. Sulfation levels in urine and others are different. No correlation is present between urinary GAG reduction and clinical improvement. Please cite the following articles that the limitation is present to use the urine GAG as monitoring or therapeutic efficacy purpose.

Erickson RP et al. Lack of relationship between blood and urine levels of glycosaminoglycans and lysosomal enzymes. *Biochemical medicine*. 1975; 12:331–339.

Saville JT et al. Glycosaminoglycan fragments as a measure of disease burden in the mucopolysaccharidosis type I mouse. *Mol Genet Metab*. 2018;123(2):112-117.

Fujitsuka H et al. Biomarkers in patients with mucopolysaccharidosis type II and IV. *Mol Genet Metab Rep*. 2019;19:100455.

Khan SA et al. Glycosaminoglycans analysis in blood and urine of mucopolysaccharidoses by tandem mass spectrometry. *Mol Genet Metab*. 2018;125(1-2):44-52

Wang J et al. High-Throughput Liquid Chromatography–Tandem Mass Spectrometry Quantification of Glycosaminoglycans as Biomarkers of Mucopolysaccharidosis II. *Int. J. Mol. Sci*. 2020, 21, 5449; doi:10.3390/ijms21155449.

Viana GM et al. Brain Pathology in Mucopolysaccharidoses (MPS) Patients with Neurological Forms. *Journal of Clinical Medicine J Clin Med*. 2020;9(2):396.

The reviewer does not believe that it costs >\$40,000. If the authors ask the collaboration to assay GAGs, it should be a much lower cost. Since the paper is sought for high quality given the high impact with the journal, it is better to ask the collaboration to measure the specific GAGs.

Please find below a point-by-point response to Rev3's comments and clarifications we have made to the manuscript.

Reviewer #3 (Remarks to the Author):

The authors responded to the comments in many points. However, there are some questions.

1. Figure S3; why treated mice had lower BV, perimeter, TC/MA, Av marrow area than WT mice? Is that adverse effect beyond therapeutic effect? Need to clarify it.

The reviewer notes that, in some measured bone parameters, values of treated mice are less than those of the B6 wild type controls and asks for clarification. Gross evaluation of the treated and B6 control mice found them to have comparable mobility and function. Furthermore, the difference in these parameters between the B6 and treated mice is much smaller than the difference in these parameters between the B6 and untreated disease mice as well as between the treated and untreated disease mice. Thus, we do not believe that this represents an adverse effect of the treatment especially given the improvement in these and other skeletal parameters in treated compared to untreated disease mice (a comparison with potentially more relevance). As such, we do not believe the findings of these three parameters in comparing treated to wild-type change the overall message and conclusions of the manuscript. However, it is important to note the need for continued studies in more sophisticated models, which are beyond the scope of the current work, in the context of these findings. We have added the following sentences to the discussion:

“Furthermore, although treated and B6 control mice had grossly comparable mobility and function, some bone parameters were reduced in treated compared to B6 mice. Whether these findings represent a clinically meaningful effect of the treatment requires investigation in additional models.”

2. GAG assay; the rev

9. GAG levels;

Considering urinary levels of GAGs do not seem to have a clear direct correlation with the clinical improvement. It is already well known that urine GAG does not correlate with any clinical improvement in MPS. Urine GAGs are pharmacodynamic marker but not for therapeutic efficacy. Is there any reason why urine GAG is used for monitoring purpose in this experiment?

It is better to measure specific GAGs (HS, DS, and KS) in blood and CSF.

HS in CSF correlates with CNS involvement while KS in blood is biomarker for the bone.

There are several articles which question about the usefulness of urine GAG in therapeutic effect since the origin of urine, blood, and tissue GAGs are different. Sulfation levels in urine and others are different. No correlation is present between urinary GAG reduction and clinical improvement.

Please cite the following articles that the limitation is present to use the urine GAG as monitoring or therapeutic efficacy purpose.

Erickson RP et al. Lack of relationship between blood and urine levels of glycosaminoglycans and lysosomal enzymes. *Biochemical medicine*. 1975; 12:331–339.

Saville JT et al. Glycosaminoglycan fragments as a measure of disease burden in the mucopolysaccharidosis type I mouse. *Mol Genet Metab*. 2018;123(2):112-117.

Fujitsuka H et al. Biomarkers in patients with mucopolysaccharidosis type II and IV. *Mol Genet Metab Rep*. 2019;19:100455.

Khan SA et al. Glycosaminoglycans analysis in blood and urine of mucopolysaccharidoses by tandem mass spectrometry. *Mol Genet Metab*. 2018;125(1-2):44-52

Wang J et al. High-Throughput Liquid Chromatography–Tandem Mass Spectrometry Quantification of Glycosaminoglycans as Biomarkers of Mucopolysaccharidosis II. *Int. J. Mol. Sci*. 2020, 21, 5449; doi:10.3390/ijms21155449.

Viana GM et al. Brain Pathology in Mucopolysaccharidoses (MPS) Patients with Neurological Forms. *Journal of Clinical Medicine J Clin Med*. 2020;9(2):396.

Serial measurement of urine GAG levels provided a noninvasive monitoring approach that we then correlated with a large number of biochemical and phenotypical parameters. This was felt to be most appropriate and the best utilization of resources in this mouse model in which pregnancies are difficult to generate. In this context, it was thought that generating a large enough animal number to allow for sacrifice at multiple time points to obtain organ GAG levels at serial time points was not feasible. Additionally, urine GAG levels have been used clinically to monitor MPS-IH disease and have been an outcome measurement in previously published mouse studies^{1–6}.

We have amended the discussion with the following phrase and added the requested references to address the reviewer's comment:

“In addition, although the objective of our study was to assess gross phenotypic correction including changes in total GAG levels, future translational work would benefit from GAG subtype analysis utilizing liquid chromatography/mass spectrometry to identify organ-specific saccharide changes following editing. Specifically, heparan sulfate and keratan sulfate levels have been shown to be related to CNS and bone pathology respectively and this analysis will add value to urinary GAG levels for clinical monitoring^{15,60–66}.”

Reviewer Response References

1. Kakkis, E. & Marsden, D. Urinary glycosaminoglycans as a potential biomarker for evaluating treatment efficacy in subjects with mucopolysaccharidoses. *Molecular Genetics and Metabolism* **130**, 7–15 (2020).
2. Sifuentes, M. *et al.* A follow-up study of MPS I patients treated with laronidase enzyme replacement therapy for 6 years. *Molecular Genetics and Metabolism* **90**, 171–180 (2007).
3. Guffon, N. *et al.* Long term disease burden post-transplantation: three decades of observations in 25 Hurler patients successfully treated with hematopoietic stem cell transplantation (HSCT). *Orphanet Journal of Rare Diseases* **16**, NA-NA (2021).
4. Wang, D. *et al.* The designer aminoglycoside NB84 significantly reduces glycosaminoglycan accumulation associated with MPS I-H in the Idua-W392X mouse. *Molecular Genetics and Metabolism* **105**, 116–125 (2012).
5. Giugliani, R. *et al.* Neurocognitive and somatic stabilization in pediatric patients with severe Mucopolysaccharidosis Type I after 52 weeks of intravenous brain-penetrating insulin receptor antibody-iduronidase fusion protein (valanafusp alpha): an open label phase 1-2 trial. *Orphanet J Rare Dis* **13**, 110 (2018).
6. Belur, L. R. *et al.* Intravenous delivery for treatment of mucopolysaccharidosis type I: A comparison of AAV serotypes 9 and rh10. *Molecular Genetics and Metabolism Reports* **24**, 100604 (2020).